# Latent Refinement via Flow Matching for Training-free Linear Inverse Problem Solving

**Hossein Askari**[1]    **Yadan Luo**[1]    **Hongfu Sun**[1]    **Fred Roosta**[1,2]
[1]The University of Queensland
[2]ARC Training Centre for Information Resilience (CIRES)
{h.askari, yadan.luo, hongfu.sun, fred.roosta}@uq.edu.au

## Abstract

Recent advances in *inverse problem* solving have increasingly adopted flow *priors* over diffusion models due to their ability to construct straight probability paths from noise to data, thereby enhancing efficiency in both training and inference. However, current flow-based inverse solvers face two primary limitations: (i) they operate directly in pixel space, which demands heavy computational resources for training and restricts scalability to high-resolution images, and (ii) they employ guidance strategies with *prior*-agnostic posterior covariances, which can weaken alignment with the generative trajectory and degrade posterior coverage. In this paper, we propose **LFlow** (**L**atent Refinement via **Flow**s), a *training-free* framework for solving linear inverse problems via pretrained latent flow priors. LFlow leverages the efficiency of flow matching to perform ODE sampling in latent space along an optimal path. This latent formulation further allows us to introduce a theoretically grounded posterior covariance, derived from the optimal vector field, enabling effective flow guidance. Experimental results demonstrate that our proposed method outperforms state-of-the-art latent diffusion solvers in reconstruction quality across most tasks. The code will be publicly available at GitHub.

## 1   Introduction

Linear *inverse problems* are fundamental to a variety of significant image processing tasks, such as super-resolution [1], inpainting [2], deblurring [3], and denoising [4]. Solving such problems involves inferring an unknown image $\mathbf{x}_0 \in \mathbb{R}^n$, which is assumed to follow an unknown prior distribution $q(\mathbf{x}_0)$, from incomplete and noisy observations $\mathbf{y} \in \mathbb{R}^m$, commonly modeled as:

$$\mathbf{y} = \mathcal{A}\mathbf{x}_0 + \mathbf{n}, \quad \mathbf{n} \sim \mathcal{N}(\mathbf{0}, \sigma_{\mathbf{y}}^2 \boldsymbol{I}), \tag{1}$$

where $\mathcal{A} \in \mathbb{R}^{m \times n}$ represents a known linear operator and $\mathbf{n}$ denotes additive i.i.d. Gaussian noise. When the operator $\mathcal{A}$ is singular (e.g., if $m < n$), the inverse problem becomes ill-posed [5], hindering the unique or stable recovery of $\mathbf{x}_0$ from $\mathbf{y}$. Consequently, accurate and plausible inferences demand strong *priors* that effectively integrate domain-specific knowledge to constrain the solution space.

Deep generative models that perform progressive *refinement* via stochastic differential equations (SDEs), particularly diffusion models [6–8], have solidified their role as powerful *priors* for solving a broad spectrum of inverse problems [9, 10]. Specifically, these models have proven effective for *zero-shot* inference of images from partially acquired and noisy measurements, with extensive research focusing on the design of *guidance* mechanisms to inject data consistency into the generative process [11–32]. Building on these advancements, diffusion-based inverse solvers have been further extended to operate in latent spaces [33–39] rather than raw pixel spaces, aiming to reduce the computational cost of training and improve generalization [40]. However, these approaches often neglect posterior variability by assuming zero covariance in likelihood-based guidance, which can lead to unstable sampling and reduced coverage of the posterior distribution [21, 30].

39th Conference on Neural Information Processing Systems (NeurIPS 2025).

Recently, *flow matching* [41, 42] has gained prominence as a compelling alternative for generative modeling. By parameterizing transformation dynamics with ordinary differential equations (ODEs), these models can generate arbitrary probability paths, including those grounded in optimal transport (OT) principles [41]. This flexibility enables the design of straight-line generative trajectories, leading to more efficient training and sampling compared to diffusion-based approaches [43, 44]. Motivated by these capabilities, several recent works have explored the use of flow-based priors for inverse problems, achieving faster and higher-quality solutions across diverse tasks [45–50]. Nevertheless, existing methods still suffer from two key drawbacks: (1) they operate in pixel space, which restricts scalability to high-dimensional data and limits generalizability across different types of inverse problems; and (2) they adopt guidance techniques originally developed for diffusion models, which estimate posterior covariances independently of the learned prior. This disconnect may steer the sampling trajectory away from high-probability regions, leading to degraded sample quality and reduced fidelity, with slower convergence often observed when *adaptive* ODE solvers are employed.

To address these limitations, we propose **LFlow** (**L**atent Refinement via **Flow**s), a framework that utilizes latent flow matching to solve linear inverse problems without additional training. By applying flow matching in latent space, LFlow achieves enhanced computational efficiency and enables more scalable and effective inverse solutions in reduced-dimensional domains. Additionally, we introduce a well-founded, time-dependent variance for the latent identity posterior covariance, formulated using Tweedie's covariance formula and the optimal vector field under the assumption of a *Gaussian latent representation*. This posterior covariance is explicitly informed by the pretrained optimal vector field, ensuring that guidance remains consistent with the generative dynamics. Our empirical evaluations demonstrate that images inferred via latent ODE sampling along conditional OT paths exhibit superior perceptual quality compared to those generated through latent diffusion-based probability paths.

**Our primary contributions are as follows:**

- **Methodological:** We propose a training-free framework based on latent flow matching and posterior-guided ODE sampling for solving linear inverse problems, significantly outperforming latent diffusion-based approaches in both efficiency and reconstruction quality.

- **Analytical:** We derive a principled correction to the pretrained latent flow using the measurement likelihood gradient and introduce an analytically justified, time-dependent posterior covariance to improve sampling accuracy and convergence speed.

- **Empirical:** We validate the performance of LFlow through extensive experiments on image reconstruction tasks, including deblurring, super-resolution, and inpainting, achieving state-of-the-art results *without* requiring substantial problem-specific hyperparameter tuning.

## 2 Overview of Related Work

*Training-free* inverse problem solvers that exploit diffusion or flow priors can be generally categorized into four methodological classes: **(1)** *Variable splitting* methods decompose inference into two alternating steps: one enforces data fidelity and the other imposes regularization [51, 36, 20]; **(2)** *Variational Bayesian* methods introduce a parameterized surrogate posterior distribution, typically Gaussian, whose parameters are optimized using a variational objective [27, 52, 53]; **(3)** *Asymptotically exact* methods combine generative priors with classical samplers—such as MCMC, SMC, or Gibbs sampling—to approximate the true posterior with convergence guarantees as the sample size grows [54–57]; and **(4)** *Guidance-based* methods correct the generative trajectory using an approximate likelihood gradient to steer samples toward the posterior [12, 21–23]. Our work centers on the fourth category and further elaborates on related methods built on various types of *priors*.

**Diffusion Guidance Approximation** refers to estimating the likelihood score $\nabla_{\mathbf{x}_t} \log p(\mathbf{y}|\mathbf{x}_t)$ during the reverse-time diffusion process governed by an SDE of the form:

$$d\mathbf{x}_t \approx \left[\mathbf{f}(\mathbf{x}_t, t) - \mathbf{g}(t)^2 \left(\mathbf{s}_{\boldsymbol{\theta}}(\mathbf{x}_t, t) + \nabla_{\mathbf{x}_t} \log p(\mathbf{y}|\mathbf{x}_t)\right)\right] dt + \mathbf{g}(t) \, d\mathbf{w}_t, \tag{2}$$

where $\mathbf{f}(\cdot, \cdot)$ is the drift, $\mathbf{g}(\cdot)$ the diffusion coefficient, $\mathbf{w}_t$ standard Brownian motion, and $\mathbf{s}_{\boldsymbol{\theta}}(\cdot, \cdot)$ a pretrained score network. Notable methods such as DPS [21], ΠGDM [22], TMPD [23], OPC [29], and MMPS [30] define the likelihood score as $\nabla_{\mathbf{x}_t} \log p(\mathbf{y}|\mathbf{x}_t) = \nabla_{\mathbf{x}_t} \log \int p(\mathbf{y}|\mathbf{x}_0)p(\mathbf{x}_0|\mathbf{x}_t) \, d\mathbf{x}_0$. The main challenge lies in computing the expectation over all possible denoised states $\mathbf{x}_0$ given $\mathbf{x}_t$, which requires sampling from $p(\mathbf{x}_0|\mathbf{x}_t)$ at every reverse step—posing significant computational demands. A

common remedy is a local Gaussian approximation $p(\mathbf{x}_0|\mathbf{x}_t) \approx \mathcal{N}\big(\mathbb{E}[\mathbf{x}_0|\mathbf{x}_t], \mathbb{C}\text{ov}[\mathbf{x}_0|\mathbf{x}_t]\big)$, reducing the problem to estimating posterior moments. These approaches differ mainly in how they specify $\mathbb{C}\text{ov}[\mathbf{x}_0|\mathbf{x}_t]$. For instance, DPS sets it to zero, and ΠGDM obtains a *prior-agnostic*, time-scaled identity matrix derived from the *forward process* under a *Gaussian data assumption*. OPC performs a *post hoc* constant variance optimization for each step. TMPD approximates the covariance by replacing the Jacobian in the Tweedie relation $\mathbb{C}\text{ov}[\mathbf{x}_0|\mathbf{x}_t] = \frac{\sigma(t)^2}{\alpha(t)}\nabla_{\mathbf{x}_t}^\top \mathbb{E}[\mathbf{x}_0|\mathbf{x}_t]$ with a diagonal row-sum surrogate, whereas MMPS evaluates the full Jacobian via automatic differentiation and solves the induced linear system with Conjugate Gradients [58]. Estimating the full covariance, however, remains computationally expensive in high dimensions, with prohibitive memory and runtime costs.

**Inference in Latent Space** has become feasible with latent diffusion models (LDMs) [59], allowing inverse solvers to reduce training costs and improve scalability [33, 36–38]. As an initial attempt, PSLD [33] augments DPS with a "gluing" objective to enforce posterior mean consistency under the autoencoder mapping, aiming to mitigate encoder–decoder nonlinearity. However, this constraint remains empirically ineffective under noisy measurements, and reconstruction artifacts persist [34]. Other approaches avoid explicit nonlinearity correction: In particular, Resample [36] constructs a Gaussian posterior by fusing a supposedly Gaussian prior on the unconditional *reverse* sample with a Gaussian pseudo-likelihood centered at a *forward-projected* measurement-consistent posterior mean. Similarly, DAPS [37] enhances posterior sampling by decoupling consecutive diffusion steps through a two-step procedure: first drawing $\mathbf{z}_0 \sim p(\mathbf{z}_0|\mathbf{z}_t, \mathbf{y})$, then re-noising $\mathbf{z}_{t-\Delta t} \sim \mathcal{N}(\mathbf{z}_0, \sigma_{t-\Delta t}^2 \boldsymbol{I})$, which provides *global* corrections—particularly effective in non-linear tasks, though at the expense of weaker local guidance in low-noise or linear settings. SITCOM [60] enforces three consistency conditions—data, forward, and backward diffusion—at each step, thereby enabling sampling with fewer steps. Nevertheless, these methods largely assume zero covariances, limiting posterior coverage.

**Flow Matching in Inverse Problems** has recently proven effective for achieving fast and high-quality solutions across various tasks [45–50]. A prime example is OT-ODE [45], which adopts ΠGDM [22] gradient correction within the flow regime and employs an ODE solver scheme based on the conditional OT path. C-ΠGFM [48] introduces a plug-and-play framework that projects conditional flow dynamics into a more amenable space, accelerating inference. PnP-Flow [61] solves imaging inverse problems by alternating a data-fidelity gradient step, a re-projection onto the flow path via latent-noise interpolation, and a time-dependent FM denoiser, all without backpropagating through the ODE. However, each of these methods typically (1) borrows guidance strategies from diffusion models that assume either zero or prior-agnostic posterior covariances, and (2) operates in pixel space, which limits their scalability and applicability to high-dimensional problems.

## 3 Preliminaries

**Continuous Normalizing Flow (CNF)** [62] constructs a smooth probability path $\{p_t(\mathbf{x}_t)\}_{t \in [0,1]}$ that transports samples from a data distribution $q(\mathbf{x}_0)$, with $\mathbf{x}_0 \in \mathbb{R}^d$, to a standard Gaussian $\mathcal{N}(\mathbf{0}, \boldsymbol{I})$ at $t = 1$. This evolution follows a time-varying vector field $\mathbf{v} : \mathbb{R}^d \times [0,1] \to \mathbb{R}^d$, governed by the ODE $\mathrm{d}\mathbf{x}_t = \mathbf{v}(\mathbf{x}_t, t)\, \mathrm{d}t$. In practice, $\mathbf{v}(\cdot, t)$ is approximated by a learnable field $\mathbf{v}_{\boldsymbol{\theta}}$, trained via maximum likelihood, which requires expensive ODE simulations.

**Flow Matching (FM)** [41] avoids inefficient likelihood-based training for CNFs by directly regressing a learnable vector field toward an analytically defined target field. In particular, *Conditional Flow Matching* (CFM) [41] introduces a time-dependent *conditional vector field* $\mathbf{v}(\mathbf{x}_t \mid \mathbf{x}_0)$ that governs the evolution of samples $\mathbf{x}_t$ conditioned on an initial point $\mathbf{x}_0$. This field induces a *conditional probability path* $p_t(\mathbf{x}_t \mid \mathbf{x}_0)$ satisfying the boundary conditions $p_0 = \delta(\mathbf{x}_0)$ and $p_1 = \mathcal{N}(\mathbf{0}, \boldsymbol{I})$. The training objective then becomes:

$$\mathcal{L}_{\text{CFM}}(\boldsymbol{\theta}) = \mathbb{E}_{t, \mathbf{x}_0 \sim q, \mathbf{x}_t \sim p_t(\mathbf{x}_t|\mathbf{x}_0)} \big\| \mathbf{v}_{\boldsymbol{\theta}}(\mathbf{x}_t, t) - \mathbf{v}(\mathbf{x}_t \mid \mathbf{x}_0) \big\|^2. \tag{3}$$

Using Gaussian paths, we define $p_t(\mathbf{x}_t|\mathbf{x}_0) = \mathcal{N}(\alpha(t)\mathbf{x}_0, \sigma(t)^2\boldsymbol{I})$ with interpolation $\mathbf{x}_t = \alpha(t)\mathbf{x}_0 + \sigma(t)\mathbf{x}_1$, where $\mathbf{x}_1 \sim \mathcal{N}(\mathbf{0}, \boldsymbol{I})$. Differentiating $\mathbf{x}_t$ with respect to $t$ gives $\dot{\alpha}(t)\mathbf{x}_0 + \dot{\sigma}(t)\mathbf{x}_1$. Substituting the inversion $\mathbf{x}_0 = [\mathbf{x}_t - \sigma(t)\mathbf{x}_1]/\alpha(t)$ yields the true vector field:

$$\mathbf{v}(\mathbf{x}_t \mid \mathbf{x}_0) = \frac{\dot{\alpha}(t)}{\alpha(t)} \mathbf{x}_t + \sigma(t)\left( \frac{\dot{\sigma}(t)}{\sigma(t)} - \frac{\dot{\alpha}(t)}{\alpha(t)} \right) \mathbf{x}_1, \tag{4}$$

where $\dot{\alpha}(t)$ and $\dot{\sigma}(t)$ denote time derivatives. In particular, choosing $\alpha(t) = 1 - t$ and $\sigma(t) = t$ recovers the OT path, which induces straight trajectories and improves efficiency.

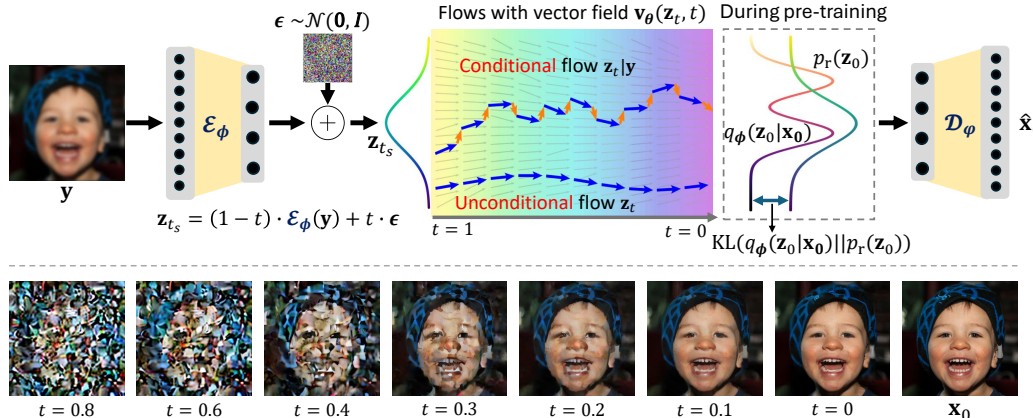

Figure 1: **(Top) LFlow** pipeline: A VAE encoder maps the observation $\mathbf{y}$ to a latent $\mathbf{z}$, from which a noisy start $\mathbf{z}_{t_s}$ is formed. During VAE pre-training, the KL term encourages the encoder's approximate posterior $q_\phi(\mathbf{z}_0|\mathbf{x}_0)$ to approach the Gaussian prior $p_r(\mathbf{z}_0) = \mathcal{N}(\mathbf{0}, \mathbf{I})$. A pretrained flow field $\mathbf{v}_\theta(\mathbf{z}_t, t)$ defines the unconditional trajectory $\mathbf{z}_t$. At inference, orange arrows ($\rightarrow$) denote likelihood-based guidance that corrects the prior field, yielding the conditional latent path $\mathbf{z}_t \mid \mathbf{y}$ toward $p(\mathbf{z}_0|\mathbf{y})$. Decoding with $\mathcal{D}_\varphi$ produces $\hat{\mathbf{x}}_0$. **(Bottom)** Gaussian deblurring snapshots along the conditional path as $t$ decreases from 0.8 to 0.

## 4  Method

To address the ill-posedness of linear inverse problems, we adopt a Bayesian view and target the posterior $p(\mathbf{x}_0 \mid \mathbf{y}) \propto p(\mathbf{y} \mid \mathbf{x}_0)\, p(\mathbf{x}_0)$, where $p(\mathbf{y} \mid \mathbf{x}_0)$ is the likelihood and $p(\mathbf{x}_0)$ is a learned prior induced by a pretrained latent flow (via the decoder). Our goal is to generate samples from this posterior without retraining a task-specific model.

### 4.1  Latent Refinement via Flows (LFlow)

We represent the prior distribution $p(\mathbf{x}_0)$ implicitly via a latent prior $p(\mathbf{z}_0)$ and the decoder push-forward $p(\mathbf{x}_0) = \mathcal{D}_{\varphi\,\#}\, p(\mathbf{z}_0)$. Specifically, let $\mathcal{E}_\phi : \mathbb{R}^d \to \mathbb{R}^k$ and $\mathcal{D}_\varphi : \mathbb{R}^k \to \mathbb{R}^d$ be a pretrained autoencoder. The latent $\mathbf{z}_0 = \mathcal{E}_\phi(\mathbf{x}_0)$ follows $p(\mathbf{z}_0)$, modeled by a flow-matching velocity $\mathbf{v}_\theta(\mathbf{z}_t, t)$ defining the ODE

$$\mathrm{d}\mathbf{z}_t = \mathbf{v}_\theta(\mathbf{z}_t, t)\, \mathrm{d}t, \quad t \in [0, 1]. \tag{5}$$

This flow transports latent samples from $\mathbf{z}_0 \sim p(\mathbf{z}_0)$ to a noise distribution $\mathbf{z}_1 \sim \mathcal{N}(\mathbf{0}, \mathbf{I})$. At inference, we integrate *backward* from $\mathbf{z}_1$ to $\hat{\mathbf{z}}_0$, adding measurement-driven guidance to follow the conditional trajectory, then decode $\hat{\mathbf{x}}_0 = \mathcal{D}_\varphi(\hat{\mathbf{z}}_0)$. Figure 1 illustrates the overall inference pipeline and provides a visual example of decoded reconstructions along the conditional trajectory, offering intuition for how the flow progresses in practice.

**Posterior Sampling via Conditional Flows in Latent Space** We aim to sample from the posterior $p(\mathbf{z}_0 \mid \mathbf{y})$ by simulating a reverse-time ODE characterized by a conditional velocity field. The following proposition provides the theoretical foundation for this procedure, ensuring that if the conditional density evolves according to a continuity equation, then the reverse-time flow recovers samples from the posterior. A formal proof is deferred to Appendix A.1.

**Proposition 4.1** (Posterior Sampling via Reverse-Time Conditional Flows). *Let $p_t(\mathbf{z}_t \mid \mathbf{y})$ denote the conditional distribution of latent variables $\mathbf{z}_t \in \mathbb{R}^k$ at time $t \in [0, 1]$, with terminal condition $p_1(\mathbf{z}_1) = \mathcal{N}(\mathbf{0}, \mathbf{I})$. Suppose this distribution evolves over time according to the reverse-time continuity equation:*

$$\partial_t p_t(\mathbf{z}_t \mid \mathbf{y}) = \nabla_{\mathbf{z}_t} \cdot [p_t(\mathbf{z}_t \mid \mathbf{y})\, \mathbf{v}_t(\mathbf{z}_t \mid \mathbf{y})], \tag{6}$$

*for some conditional velocity field $\mathbf{v}_t(\mathbf{z}_t \mid \mathbf{y})$. Then, the solution to the reverse-time ODE*

$$d\mathbf{z}_t = -\mathbf{v}_t(\mathbf{z}_t \mid \mathbf{y})\, dt, \quad \mathbf{z}_1 \sim \mathcal{N}(\mathbf{0}, \mathbf{I}), \tag{7}$$

*yields samples from the posterior $p(\mathbf{z}_0 \mid \mathbf{y})$ as $t \to 0$.*

**Conditional Vector Field Estimation** Having established that posterior samples can be obtained by integrating the reverse-time ODE in Eq. (7), it remains to estimate the corresponding conditional

vector field $\mathbf{v}_t(\mathbf{z}_t|\mathbf{y})$. Under Gaussian latent dynamics, this field takes the form:

$$\mathbf{v}_t(\mathbf{z}_t \mid \mathbf{y}) = \frac{\dot{\alpha}(t)}{\alpha(t)}\,\mathbf{z}_t + \lambda(t)\,\nabla_{\mathbf{z}_t} \log p_t(\mathbf{z}_t \mid \mathbf{y}), \quad \lambda(t) = \frac{\mathrm{d}}{\mathrm{d}t}\left(\frac{\sigma(t)}{\alpha(t)}\right). \tag{8}$$

By applying Bayes' rule, we obtain a principled decomposition of the velocity field, which leads to the following practical approximation via the pretrained flow and a likelihood correction.

$$\mathbf{v}_t(\mathbf{z}_t \mid \mathbf{y}) \approx \mathbf{v}_{\boldsymbol{\theta}}(\mathbf{z}_t, t) - \frac{t}{1-t}\,\nabla_{\mathbf{z}_t} \log p_t(\mathbf{y} \mid \mathbf{z}_t), \tag{9}$$

where the additional term $\nabla_{\mathbf{z}_t} \log p_t(\mathbf{y} \mid \mathbf{z}_t)$, commonly referred to as *guidance*, steers the flow toward consistency with the measurements $\mathbf{y}$. A complete proof of the results in Eq. (8) and Eq. (9) in the latent domain is provided in Appendix A.2.

**Likelihood Approximation** A key challenge in Eq. (9) arises from approximating the gradient of the noise-conditional distribution, which we define as

$$\nabla_{\mathbf{z}_t} \log p(\mathbf{y} \mid \mathbf{z}_t) = \nabla_{\mathbf{z}_t} \log \int p(\mathbf{y} \mid \mathbf{x}_0, \mathbf{z}_t) p(\mathbf{x}_0 \mid \mathbf{z}_t) \mathrm{d}\mathbf{x}_0 = \nabla_{\mathbf{z}_t} \log \mathbb{E}_{\mathbf{x}_0 \sim p(\mathbf{x}_0 \mid \mathbf{z}_t)}[p(\mathbf{y}|\mathbf{x}_0)]. \tag{10}$$

Here, the likelihood term can be expressed as $p(\mathbf{y} \mid \mathbf{x}_0) = \mathcal{N}(\mathbf{y}; \mathcal{A}\mathcal{D}_{\boldsymbol{\varphi}}(\mathbf{z}_0), \sigma_{\mathbf{y}}^2 \mathbf{I})$, which is a Gaussian distribution with a nonlinear mean induced by the decoder $\mathcal{D}_{\boldsymbol{\varphi}}$. Due to the nonlinearity of $\mathcal{D}_{\boldsymbol{\varphi}}$, the marginal likelihood $p(\mathbf{y} \mid \mathbf{z}_t)$ deviates from Gaussianity. To facilitate the computation of its gradient, we approximate $\mathcal{D}_{\boldsymbol{\varphi}}(\mathbf{z}_0)$ via a first-order Taylor expansion around $\bar{\mathbf{z}}_0 := \mathbb{E}[\mathbf{z}_0 \mid \mathbf{z}_t]$ as:

$$\mathcal{D}_{\boldsymbol{\varphi}}(\mathbf{z}_0) \approx \mathcal{D}_{\boldsymbol{\varphi}}(\bar{\mathbf{z}}_0) + J_{\mathcal{D}}(\mathbf{z}_0 - \bar{\mathbf{z}}_0), \tag{11}$$

where $J_{\mathcal{D}} := J_{\mathcal{D}}(\bar{\mathbf{z}}_0)$ is the Jacobian of $\mathcal{D}_{\boldsymbol{\varphi}}$ at $\bar{\mathbf{z}}_0$. Assuming the posterior $p(\mathbf{z}_0 \mid \mathbf{z}_t) \sim \mathcal{N}(\bar{\mathbf{z}}_0, \boldsymbol{\Sigma}_{\mathbf{z}})$, the distribution of the image $\mathbf{x}_0 = \mathcal{D}_{\boldsymbol{\varphi}}(\mathbf{z}_0)$ becomes approximately Gaussian as well:

$$\mathbf{x}_0 \sim \mathcal{N}(\mathbb{E}[\mathbf{x}_0|\mathbf{z}_t], \mathbb{C}\mathrm{ov}[\mathbf{x}_0|\mathbf{z}_t]), \quad \text{where} \quad \mathbb{E}[\mathbf{x}_0|\mathbf{z}_t] \approx \mathcal{D}_{\boldsymbol{\varphi}}(\bar{\mathbf{z}}_0), \quad \mathbb{C}\mathrm{ov}[\mathbf{x}_0 \mid \mathbf{z}_t] \approx J_{\mathcal{D}}\,\boldsymbol{\Sigma}_{\mathbf{z}}\,J_{\mathcal{D}}^{\top}. \tag{12}$$

Using this approximation, we estimate the gradient of the log-likelihood in latent space based on the decoded mean and the propagated covariance (see Appendix A.4):

$$\nabla_{\mathbf{z}_t} \log p(\mathbf{y} \mid \mathbf{z}_t) \approx \underbrace{(\nabla_{\mathbf{z}_t}\bar{\mathbf{z}}_0)^{\top} J_{\mathcal{D}}^{\top}}_{\mathbf{J}} \underbrace{\mathcal{A}^{\top}\left(\sigma_{\mathbf{y}}^2 \mathbf{I} + \mathcal{A}J_{\mathcal{D}}\,\boldsymbol{\Sigma}_{\mathbf{z}}\,J_{\mathcal{D}}^{\top}\mathcal{A}^{\top}\right)^{-1}(\mathbf{y} - \mathcal{A}\mathcal{D}_{\boldsymbol{\varphi}}(\bar{\mathbf{z}}_0))}_{\mathbf{v}}, \tag{13}$$

which involves a Jacobian-vector product ($\mathbf{J} \cdot \mathbf{v}$), and can be efficiently computed using automatic differentiation (Appendix B). Furthermore, the Gaussian posterior assumption enables the application of Tweedie's formula [63], which connects the pretrained vector field $\mathbf{v}_{\boldsymbol{\theta}}(\mathbf{z}_t, t)$ to the posterior moments $\mathbb{E}[\mathbf{z}_0|\mathbf{z}_t]$ and $\mathbb{C}\mathrm{ov}[\mathbf{z}_0|\mathbf{z}_t]$. Specifically, we have (refer to Appendix A.3):

$$\mathbb{E}[\mathbf{z}_0 \mid \mathbf{z}_t] = \mathbf{z}_t - t\,\mathbf{v}_{\boldsymbol{\theta}}(\mathbf{z}_t, t), \tag{14}$$

$$\mathbb{C}\mathrm{ov}[\mathbf{z}_0 \mid \mathbf{z}_t] = \frac{t^2}{1-t}\left(\mathbf{I} - t\,\nabla_{\mathbf{z}_t}\mathbf{v}_{\boldsymbol{\theta}}(\mathbf{z}_t, t)\right). \tag{15}$$

**Latent Posterior Covariance** Computing the posterior covariance $\mathbb{C}\mathrm{ov}[\mathbf{z}_0|\mathbf{z}_t]$ in low-dimensional latent spaces is more tractable and can be efficiently estimated via automatic differentiation. However, inverting the matrix $\left(\sigma_{\mathbf{y}}^2 \mathbf{I} + \mathcal{A}J_{\mathcal{D}}\,\boldsymbol{\Sigma}_{\mathbf{z}}\,J_{\mathcal{D}}^{\top}\mathcal{A}^{\top}\right)$ in Eq. (13) remains a significant computational bottleneck. To avoid explicit inversion, one may solve the corresponding linear system using the Conjugate Gradient (CG) method. Yet, CG's convergence crucially depends on the symmetry and positive definiteness of the system matrix—conditions that may be violated in practice due to imperfections in the pretrained velocity field $\mathbf{v}_{\boldsymbol{\theta}}$, potentially leading to instability or divergence. To address this, we further analyze the structure of the posterior covariance and the Jacobian $\nabla_{\mathbf{z}_t}\mathbf{v}_{\boldsymbol{\theta}}(\mathbf{z}_t, t)$, under certain regularity assumptions on the latent prior $p(\mathbf{z}_0)$.

**Assumption 4.2** (Strong Log-Concavity of the Latent Prior [64]). *Let $p(\mathbf{z}_0) = \exp(-\Phi(\mathbf{z}_0))$ be the latent prior over $\mathbb{R}^d$, where $\Phi$ is a twice continuously differentiable potential function. We assume that $p(\mathbf{z}_0)$ is $\gamma$-strongly log-concave for some $\gamma > 0$; that is, $\nabla^2\Phi(\mathbf{z}_0) \succeq \gamma\,\mathbf{I}_d$, for all $\mathbf{z}_0 \in \mathbb{R}^d$.*

**(1) Log-concave Prior:** This assumption imposes a uniform lower bound on the Hessian of the prior's potential function, which translates to a lower bound on the Jacobian of the vector field.

**Proposition 4.3 (Bound on the Jacobian of the Vector Field).** *Let $\mathbf{v}_{\boldsymbol{\theta}}(\mathbf{z}_t, t)$ denote the velocity field of the interpolant $\mathbf{z}_t = \alpha(t)\mathbf{z}_0 + \sigma(t)\mathbf{z}_1$ between a standard Gaussian prior and a target distribution $p(\mathbf{z}_0)$, defined via coefficients $\alpha(t), \sigma(t) \in \mathbb{R}$. Under Assumption 4.2, the Jacobian of the vector field satisfies the following bound:*

$$\frac{\mathrm{d}}{\mathrm{d}t}\left(\tfrac{1}{2}\log\left(\alpha(t)^2 + \gamma\,\sigma(t)^2\right)\right) \cdot \boldsymbol{I}_{d_z} \preceq \nabla_{\mathbf{z}_t}\mathbf{v}_{\boldsymbol{\theta}}(\mathbf{z}_t, t) \prec \frac{\mathrm{d}}{\mathrm{d}t}\log\sigma(t) \cdot \boldsymbol{I}_{d_z} \quad \forall t\in(0,1],\, \mathbf{z}\in\mathbb{R}^d. \tag{16}$$

The proof is provided in Appendix A.5. The resulting sandwich bound ensures the Jacobian remains well-behaved—the upper bound guarantees valid and stable posterior covariance, while the lower bound prevents overestimated uncertainty during posterior-guided inference.

**(2) Gaussian Special Case:** For practical inference and efficient computation, we now consider a tractable special case where the latent prior is Gaussian, $\mathbf{z}_0 \sim \mathcal{N}(0, \sigma_{latr}^2\boldsymbol{I})$, a choice that is both theoretically justified and widely used in practice—for instance, in variational autoencoders (VAEs), where the latent prior is regularized toward a standard Gaussian via KL divergence [65]. This enables us to derive the optimal velocity field and its Jacobian in closed form, which is characterized in the next proposition, whose proof can be found in Appendix A.6.

**Proposition 4.4 (Optimal Vector Field).** *Let $\mathbf{z}_0 \sim \mathcal{N}(\mathbf{0}, \sigma_{latr}^2\boldsymbol{I}_{d_z})$ and $\mathbf{z}_1 \sim \mathcal{N}(\mathbf{0}, \boldsymbol{I}_{d_z})$ be independent random variables. Define $\mathbf{z}_t = (1-t)\mathbf{z}_0 + t\mathbf{z}_1$ for $t \in [0,1]$. The optimal vector field $\mathbf{v}^\star(\mathbf{z}_t, t)$ that minimizes the expected squared error $\arg\min_{\mathbf{v}} \mathbb{E}\left[\|\mathbf{v}(\mathbf{z}_t, t) - (\mathbf{z}_0 - \mathbf{z}_1)\|^2\right]$ is given by*

$$\mathbf{v}^\star(\mathbf{z}_t, t) = \frac{(1-t)\sigma_{latr}^2 - t}{(1-t)^2\sigma_{latr}^2 + t^2}\,\mathbf{z}_t. \tag{17}$$

**Remark 4.5 (Tightness of Bound).** *When $\mathbf{z}_0 \sim \mathcal{N}(0, \sigma_{latr}^2\boldsymbol{I})$, the optimal vector field $\mathbf{v}^\star(\mathbf{z}_t, t)$ achieves the Jacobian lower bound in Proposition 4.3. This follows from the potential function $\Phi(\mathbf{z}_0) = \frac{1}{2\sigma_{latr}^2}\|\mathbf{z}_0\|^2$, yielding $\gamma = \sigma_{latr}^{-2}$, and interpolation coefficients $\alpha(t) = 1 - t$, $\sigma(t) = t$. Substituting into the bound confirms it matches the exact Jacobian of $\mathbf{v}^\star$, making the bound tight.*

Plugging the Jacobian of the optimal vector field into Eq. (15) yields:

$$\mathbb{Cov}[\mathbf{z}_0\,|\,\mathbf{z}_t] = r^2(t) \cdot \boldsymbol{I}_{d_z}, \quad \text{with} \quad r^2(t) = \frac{t^2\left[(1-t)(1-2t) + 2t^2\right]}{(1-t)\left[(1-t)^2 + t^2\right]}, \tag{18}$$

where we assumed $\sigma_{\text{latr}} = 1$. As a result, the propagated covariance can also be simplified as

$$\mathbb{Cov}[\mathbf{x}_0\,|\,\mathbf{z}_t] \approx J_{\mathcal{D}} \cdot \mathbb{Cov}[\mathbf{z}_0\,|\,\mathbf{z}_t] \cdot J_{\mathcal{D}}^\top \approx r^2(t) \cdot J_{\mathcal{D}}J_{\mathcal{D}}^\top. \tag{19}$$

Computing the full Jacobian $J_{\mathcal{D}} \in \mathbb{R}^{d_x \times d_z}$ is often infeasible in practice. To simplify, we assume that the decoder acts approximately as a local isometry near $\bar{\mathbf{z}}_0$ [66], such that $J_{\mathcal{D}}J_{\mathcal{D}}^\top \approx \cdot P$, where $P$ is the orthogonal projector onto the image of $J_{\mathcal{D}}$. For computational convenience, we approximate this behavior as isotropic in the full space, resulting in:

$$\mathbb{Cov}[\mathbf{x}_0\,|\,\mathbf{z}_t] \approx r^2(t) \cdot \boldsymbol{I}_{d_x}. \tag{20}$$

**Comparison with $\Pi$GDM and OT-ODE** The posterior covariance in $\Pi$GDM is derived solely from the *forward* process under the strong assumption of *Gaussian data space*, resulting in a variance for the identity covariance as $r^2(t) = \frac{\sigma(t)^2}{\alpha(t)^2 + \sigma(t)^2}\,\boldsymbol{I}$. Please see Appendix A.7 for details.

Additionally, $\Pi$GDM leverages the inverse of this posterior covariance as Fisher information for natural gradient [67] updates on samples during the guidance step, thereby enhancing performance. OT-ODE can be regarded as a flow-based extension of $\Pi$GDM, where it adopts the same covariance expression but replaces the diffusion forward process with a flow-specific noise scheduler, specifically setting $\alpha(t) = 1 - t$ and $\sigma(t) = t$. Therefore, both methods rely on an identity posterior covariance that is *independent* of the learned score or vector field. In contrast, our proposed covariance explicitly incorporates information from the pre-trained vector field. This in turn allows for more effective guidance during flow-based ODE sampling, ultimately improving reconstruction quality. Figure 2 visually illustrates how our time-dependent variance differs in magnitude and effect from the simpler identity-based covariance used in OT-ODE.

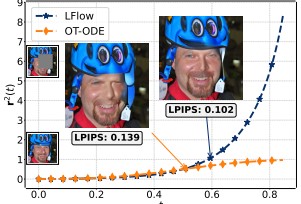

Figure 2: **Posterior covariance values** across $t \in [0, 0.9]$ for our method and OT-ODE, along with reconstructed images for a single FFHQ sample in the super-resolution task.

**Initiating the Flow Sampling Process**  Inspired by prior works in pixel spaces [68, 45], we propose initializing the reverse ODE—across all tasks—from a *partially corrupted* version of the *encoding* of $\mathbf{y}$ in the latent space. Specifically, rather than starting from pure noise $\mathbf{z}_1 \sim \mathcal{N}(0, \boldsymbol{I})$ at $t = 1$, we initialize at $t_s < 1$:

$$\mathbf{z}_{t_s} = (1 - t_s)\, \mathcal{E}_{\boldsymbol{\phi}}(\mathbf{y}) + t_s\, \mathbf{z}_1, \quad \mathbf{z}_1 \sim \mathcal{N}(0, \boldsymbol{I}). \tag{21}$$

This initialization ensures that $\mathbf{z}_{t_s}$ is closer to the posterior mode $\mathbf{z}_0 \mid \mathbf{y}$, making the subsequent backward integration more stable and likely to remain on a plausible manifold.

We summarize the complete sampling algorithm via latent ODE flows in Algorithm 1 (Appendix C).

## 5  Experiments

**Datasets and Tasks**  We evaluate our method on three datasets: FFHQ [69], ImageNet [70], and CelebA-HQ [71], each containing images with a resolution of $256 \times 256 \times 3$ pixels. We use 200 randomly selected validation samples per dataset. All images are normalized to the range $[-1, 1]$. We present our findings on several linear inverse problem tasks, including Gaussian deblurring, motion deblurring, super-resolution, and box inpainting. The measurement operators are configured as follows: (i) Gaussian deblurring convolves images with a Gaussian blurring kernel of size $61 \times 61$ and a standard deviation of 3.0 [29]; (ii) Motion Deblurring uses motion blur kernels that are randomly generated with a size of $61 \times 61$ and an intensity value of 0.5 [29]; (iii) Super-Resolution (SR) involves bicubic downsampling by a factor of 4; and (iv) Box Inpainting simulates missing data by masking a $128 \times 128$ pixel box, which is randomly positioned around the center with a margin of $[16, 16]$ pixels, following the methodology of [21]. All measurements in these experiments are corrupted by Gaussian noise with a standard deviation of $\sigma_{\mathbf{y}} = 0.01$, ensuring that the reconstruction methods are evaluated under realistic noisy conditions.

**Baselines and Metrics**  Since the primary goal of this study is to improve solving inverse problems in the latent space, we focus on comparing our proposed method against state-of-the-art latent diffusion solvers—specifically, **PSLD** [33], **MPGD** [38], **Resample** [36], **DMplug** [51], **DAPS** [37], and **SITCOM** [60] through both quantitative and qualitative analyses. Additionally, we include results from the pixel-based **ΠGDM** [22], **OT-ODE** [45], and **C-ΠGFM** [48], when reported, for a more comprehensive comparison. Following the evaluation protocols of prior works, we report quantitative metrics including Learned Perceptual Image Patch Similarity (LPIPS) [72], Peak Signal-to-Noise Ratio (PSNR), Structural Similarity Index (SSIM), and Fréchet Inception Distance (FID) [73] (Appendix D.3).

**Architecture, Training, and Sampling**  In contrast to previous latent diffusion inverse solvers, which rely on the LDM-VQ-4 or Stable Diffusion (SD) v1.5 models [59] built on U-Net [74], we adopt the LFM-VAE framework [75] based on DiT transformer architectures [76] for all datasets. Following the methodology outlined in [75], we leverage their pre-trained model. It is important to note that, despite our architectural differences, the quality of our prior does not exceed that of SDs or LDMs. This limitation may partly arise from using a lower-dimensional latent space of $32 \times 32$. After evaluating various numerical ODE integration methods, we selected `adaptive_heun` as the default solver, utilizing its reliable implementation from the open-source `torchdiffeq` library [62]. All experiments were conducted on a single NVIDIA 3090 GPU with a batch size of 1. Further details on baselines, model configurations, and other hyper-parameters are provided in Appendix C.

### 5.1  Results

For clarity, the quantitative results for the four tasks on both FFHQ and ImageNet are presented in Tables 1 and 2, while qualitative comparisons are shown in Figures 3 and 4. The following subsections provide a detailed discussion of each task. Due to space constraints, results for CelebA-HQ are included in Table 5 and Figure 7 in Appendix D. Additional results are also presented in this section.

**Gaussian Deblurring**  LFlow achieves the best perceptual quality across both datasets, attaining the lowest LPIPS and highest SSIM—surpassing the second-best method by approximately 8.8% in LPIPS on FFHQ. While PSLD slightly leads in PSNR, LFlow closes the gap within 1.18 dB and avoids the over-smoothing and loss of fine detail often observed in PSLD outputs (e.g., blurred backgrounds and softened skin textures). On ImageNet, LFlow similarly excels in perceptual quality, restoring structures like the dog's snout, fur, and facial wrinkles more faithfully. The reconstructions

Table 1: Quantitative evaluation of inverse problem solving on **FFHQ** samples of the validation dataset. **Bold** and underline indicates the best and second-best respectively. The methods shaded in gray are in pixel space.

| Method | Deblurring (Gaussian) | | | Deblurring (Motion) | | | SR (×4) | | | Inpainting (Box) | | |
|---|---|---|---|---|---|---|---|---|---|---|---|---|
| | PSNR↑ | SSIM↑ | LPIPS↓ | PSNR↑ | SSIM↑ | LPIPS↓ | PSNR↑ | SSIM↑ | LPIPS↓ | PSNR↑ | SSIM↑ | LPIPS↓ |
| LFlow (**ours**) | 29.10 | **0.837** | **0.166** | **30.04** | **0.849** | **0.168** | 29.12 | **0.841** | **0.176** | 23.85 | **0.867** | **0.132** |
| SITCOM [60] | **30.42** | 0.828 | 0.237 | 28.78 | 0.828 | 0.183 | 29.26 | 0.833 | 0.191 | 24.12 | 0.839 | 0.198 |
| DAPS [37] | 28.52 | 0.789 | 0.231 | 29.00 | 0.831 | 0.252 | **29.38** | 0.826 | 0.197 | 24.83 | 0.819 | 0.191 |
| DMplug [51] | 27.43 | 0.784 | 0.240 | 27.95 | 0.817 | 0.243 | 29.45 | 0.838 | 0.183 | 22.55 | 0.807 | 0.220 |
| Resample [36] | 28.73 | 0.801 | 0.201 | 29.19 | 0.828 | 0.184 | 28.90 | 0.804 | 0.189 | 20.40 | 0.825 | 0.243 |
| MPGD [38] | 29.34 | 0.815 | 0.308 | 27.98 | 0.803 | 0.324 | 27.49 | 0.788 | 0.295 | 20.58 | 0.806 | 0.324 |
| PSLD [33] | 30.28 | 0.836 | 0.281 | 29.21 | 0.812 | 0.303 | 29.07 | 0.834 | 0.270 | 24.21 | 0.847 | 0.169 |
| OT-ODE [45] | 29.73 | 0.819 | 0.198 | 28.15 | 0.792 | 0.238 | 28.56 | 0.823 | 0.198 | 25.77 | 0.751 | 0.225 |
| ΠGDM [22] | 28.62 | 0.809 | 0.182 | 27.18 | 0.773 | 0.223 | 27.78 | 0.815 | 0.201 | **26.82** | 0.767 | 0.214 |

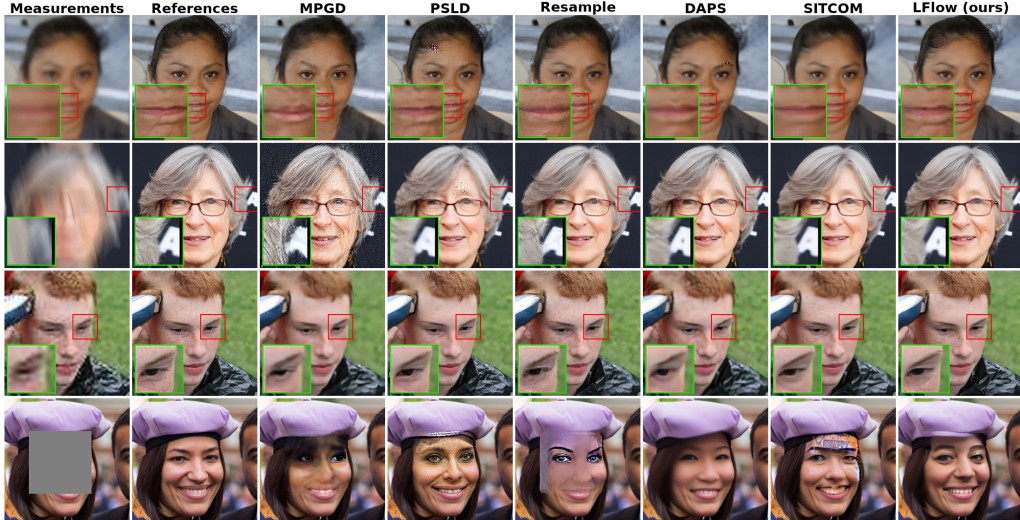

Figure 3: Qualitative results on **FFHQ** test set. Row 1: Deblur (gaussian), Row 2: Deblur (motion), Row 3: SR×4, Row 4: Inpainting. Our approach better preserves fine image details than *latent-based* diffusion methods.

remain sharp and natural, free from the spurious high-frequency artifacts or texture distortions that affect other approaches.

**Motion Deblurring**    LFlow reconstructs motion-blurred scenes with strong structural consistency and minimal artifacts, effectively preserving sharp transitions along motion boundaries. Fine details—such as hair contours, facial edges, and accessories—are recovered with smooth gradients and natural appearances. Without introducing ringing or over-enhancement, LFlow balances fidelity and realism across both FFHQ and ImageNet, achieving state-of-the-art performance on all metrics. Notably, LFlow improves LPIPS by a clear margin of 0.015 over the second-best method, SITCOM, on FFHQ, reflecting its ability to retain high-frequency content while maintaining spatial coherence—qualities that are visually evident in the restored hair and facial structures.

**Super-Resolution**    High-frequency structures such as hair strands, floral patterns, and facial details are reconstructed with remarkable clarity by our method, without the over-sharpening or softness often observed in competing approaches. On FFHQ, it achieves a perceptual gain of 0.021 LPIPS over the next-best method, while also maintaining competitive PSNR. On ImageNet, it establishes a clear lead across all metrics, with an LPIPS margin of 0.018. These improvements translate to visually cleaner textures and more coherent spatial gradients. Unlike MPGD, which may produce overly synthetic details, or PSLD and Resample, which occasionally yield hazy regions, our approach preserves the natural appearance of fine structures while avoiding haloing and texture overshoot.

**Inpainting**    When completing missing regions, LFlow excels in producing semantically coherent and perceptually consistent content that blends naturally with surrounding areas. On FFHQ, it surpasses the second-best method in LPIPS by a margin of 0.037 while also achieving the highest SSIM, reflecting both perceptual sharpness and structural accuracy. On ImageNet, it continues to lead in LPIPS and SSIM, recovering fine textures and edges with minimal boundary artifacts.

Table 2: Quantitative results of inverse problem solving on **ImageNet** samples of the validation dataset. **Bold** and underline indicates the best and second-best respectively. The methods shaded in gray are in pixel space.

| Method | Deblurring (Gaussian) | | | Deblurring (Motion) | | | SR ($\times 4$) | | | Inpainting (Box) | | |
|---|---|---|---|---|---|---|---|---|---|---|---|---|
| | PSNR↑ | SSIM↑ | LPIPS↓ | PSNR↑ | SSIM↑ | LPIPS↓ | PSNR↑ | SSIM↑ | LPIPS↓ | PSNR↑ | SSIM↑ | LPIPS↓ |
| LFlow (**ours**) | 25.55 | 0.697 | **0.328** | **26.10** | **0.711** | **0.344** | 25.29 | **0.696** | **0.338** | 21.92 | **0.772** | **0.227** |
| SITCOM [60] | 25.38 | 0.672 | 0.388 | 24.78 | 0.686 | 0.382 | 25.62 | 0.687 | 0.374 | 20.34 | 0.698 | 0.291 |
| DAPS [37] | 24.12 | 0.681 | 0.413 | 25.97 | 0.706 | 0.362 | 25.18 | 0.667 | 0.356 | 21.13 | 0.701 | 0.286 |
| Resample [36] | 25.04 | 0.665 | 0.408 | 24.32 | 0.623 | 0.390 | 24.81 | 0.683 | 0.404 | 19.42 | 0.663 | 0.305 |
| MPGD [38] | 24.27 | 0.695 | 0.397 | 24.81 | 0.662 | 0.404 | 25.50 | 0.648 | 0.398 | 17.05 | 0.672 | 0.324 |
| PSLD [33] | **26.79** | **0.721** | 0.372 | 25.45 | 0.692 | 0.351 | **26.16** | 0.692 | 0.363 | 20.58 | 0.687 | 0.274 |
| ΠGDM [22] | 25.27 | 0.636 | 0.332 | 23.03 | 0.617 | 0.347 | 24.73 | 0.629 | 0.359 | **22.13** | 0.589 | 0.361 |

Figure 4: Qualitative results on **ImageNet** test set. Row 1: Deblur (gaussian), Row 2: Deblur (motion), Row 3: SR$\times 4$, Row 4: Inpainting. Our method reconstructs fine image details more faithfully than the baselines.

In contrast to methods that may introduce visible seams, blotchy patterns, or inconsistent colors, LFlow reconstructs faces, objects, and natural scenes with smoother transitions and well-aligned local details—resulting in reconstructions that appear complete and visually seamless.

**Perceptual–Fidelity Trade-off** While LFlow occasionally reports slightly lower PSNR than pixel-fidelity-oriented baselines such as PSLD, this reflects the well-known trade-off between distortion and perception. Our approach consistently achieves substantially lower LPIPS and sharper, more natural reconstructions, indicating that it prioritizes perceptual realism and fine-detail preservation over pixel-wise averaging effects that often inflate PSNR.

## 5.2 Ablation Study

**Posterior Covariance** To examine how the time-dependent posterior covariance influences overall performance, we conducted a series of ablation studies. As shown in Table 3, adapting the posterior covariance (labeled `Cov_LFlow`) provides systematic gains over the baseline "`Cov_ΠGDM`". Across both FFHQ and ImageNet, `Cov_LFlow` yields higher PSNR and lower LPIPS for deblurring and $\times 4$ SR, indicating that LFlow helps reduce perceptual artifacts while preserving fine details. Figure 6 presents qualitative comparisons between `Cov_LFlow` and `Cov_ΠGDM` on FFHQ. Across both motion deblurring and inpainting, `Cov_LFlow` yields sharper facial structures, cleaner textures, and fewer artifacts. For motion deblurring, it better restores contours and eye details without ringing, while inpainting results show improved shading consistency and reduced boundary errors. These visual observations align with the quantitative improvements reported in Table 3.

**Starting Time $t_s$** We also investigated the impact of the starting time $t_s$ on the results. Figure 5 illustrates the impact of varying the start time $t_s$ for the flow process (SR task). We see that overly large $t_s$ values tend to slightly degrade perceptual quality, whereas overly small $t_s$ values can cause excessive smoothing or artifacts. The plot indicates that an intermediate choice of $t_s$ (around 0.7–0.8) strikes a favorable balance, leading to consistently lower LPIPS on both FFHQ and ImageNet.

Table 3: Ablations on the effect of $\mathbb{C}\mathrm{ov}[\mathbf{z}_0|\mathbf{z}_t]$. **Bold** indicates the best.

| Dataset | FFHQ | | | | ImageNet | | | |
|---|---|---|---|---|---|---|---|---|
| | Deblur (G) | | Inpainting | | Deblur (M) | | SR ($\times 4$) | |
| | PSNR↑ | LPIPS↓ | PSNR↑ | LPIPS↓ | PSNR↑ | LPIPS↓ | PSNR↑ | LPIPS↓ |
| Cov_LFlow | **29.10** | **0.166** | **23.85** | **0.132** | **26.10** | **0.344** | **25.29** | **0.338** |
| Cov_ΠGDM | 29.04 | 0.179 | 22.69 | 0.151 | 25.22 | 0.363 | 24.80 | 0.351 |

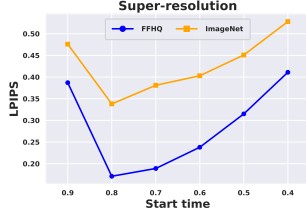

Figure 5: Ablation study on the start time $t_s$.

Table 4: **Average inference time** (seconds per image) for latent-based inverse solvers on ImageNet samples. Timings are measured on an NVIDIA 3090 GPU.

| Method | Gaussian Deblur | SR $\times 4$ |
|---|---|---|
| Resample [36] | 550.26s | 410.68s |
| MPGD [38] | 566.58s | 548.96s |
| PSLD [33] | 705.31s | 675.85s |
| SITCOM [60] | 345.60s | 328.12s |
| LFlow | **267.76**s | **227.92**s |
| LFlow (Cov_ΠGDM) | 477.95s | 406.02s |

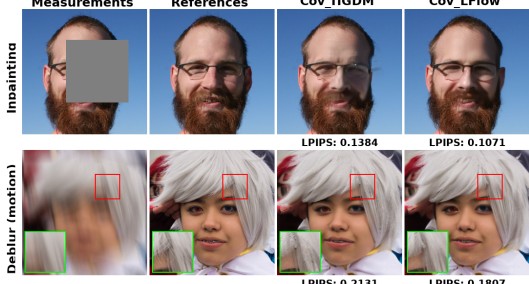

Figure 6: Visual results on the effect of $\mathbb{C}\mathrm{ov}[\mathbf{z}_0|\mathbf{z}_t]$.

**Inference Time**  To highlight the practical advantages of posterior covariance modeling beyond standard evaluation metrics, we report the average inference time per image in Table 4. Despite using an adaptive ODE solver with potentially higher NFE, LFlow achieves significantly faster inference than prior latent-based solvers. This efficiency stems from faster convergence and more accurate trajectory estimation. The variant Cov_ΠGDM, which employs a less accurate covariance, converges more slowly—demonstrating the importance of proper covariance modeling. Notably, PSLD runs for 1000 iterations and Resample for 500, yet both are slower than LFlow.

## 6 Discussions

In this paper, we present LFlow, which efficiently addresses linear inverse problems by utilizing flow matching in the latent space of pre-trained autoencoders without additional training. Based on a justified latent *Gaussian representation assumption*, our approach introduces a theoretically sound, time-dependent latent posterior covariance that enhances gradient-based inference. Experimental results across such tasks as deblurring, super-resolution, and inpainting demonstrate that LFlow outperforms current latent diffusion models in reconstruction quality.

**Limitation.** One limitation of LFlow lies in its runtime: the current implementation can require approximately 3:30–10 minutes to solve inverse problems on an NVIDIA RTX 3090 GPU, with Gaussian deblurring and super-resolution averaging 2:30–6 minutes, and motion deblurring and inpainting taking around 3:30–10 minutes. While this is more efficient than existing latent diffusion solvers, as demonstrated in our ablation studies, it may still pose challenges in time-sensitive applications. Nonetheless, the improved reconstruction quality and the efficiency gains achieved during training help mitigate this drawback.

**Future work.** Future work will focus on optimizing the solver and investigating alternative numerical integration schemes to further reduce inference time. In addition, we plan to extend LFlow to improve its robustness under distributional shifts, enabling broader applicability to real-world scenarios and downstream tasks.

## Acknowledgement

This research was partially supported by the Australian Research Council through an Industrial Transformation Training Centre for Information Resilience (IC200100022).

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

# A  Proofs

**Lemma A.1** (**Tweedie's Mean Formula** ). *Suppose the joint distribution of $\mathbf{z}_0$ and $\mathbf{z}_t$ factors as*

$$p_t(\mathbf{z}_0, \mathbf{z}_t) \; = \; p(\mathbf{z}_0)\, p_t(\mathbf{z}_t \mid \mathbf{z}_0)$$

*with*

$$p_t(\mathbf{z}_t \mid \mathbf{z}_0) \; = \; \mathcal{N}\big(\mathbf{z}_t \mid \alpha(t)\, \mathbf{z}_0,\; \sigma(t)^2\, \boldsymbol{I}\big).$$

*Then*

$$\mathbb{E}\big[\mathbf{z}_0 \mid \mathbf{z}_t\big] \; = \; \frac{1}{\alpha(t)}\Big(\, \mathbf{z}_t \;+\; \sigma(t)^2\, \nabla_{\mathbf{z}_t} \log p_t(\mathbf{z}_t)\Big). \tag{22}$$

*Proof.* Starting from the definition of the score,

$$\nabla_{\mathbf{z}_t} \log p_t(\mathbf{z}_t) = \frac{\nabla_{\mathbf{z}_t} p_t(\mathbf{z}_t)}{p_t(\mathbf{z}_t)} \; = \; \frac{1}{p_t(\mathbf{z}_t)} \nabla_{\mathbf{z}_t} \int p_t(\mathbf{z}_0, \mathbf{z}_t)\, \mathrm{d}\mathbf{z}_0$$

$$= \; \frac{1}{p_t(\mathbf{z}_t)} \int \nabla_{\mathbf{z}_t} \big(p_t(\mathbf{z}_0, \mathbf{z}_t)\big)\, \mathrm{d}\mathbf{z}_0 \; = \; \frac{1}{p_t(\mathbf{z}_t)} \int p_t(\mathbf{z}_0, \mathbf{z}_t)\, \nabla_{\mathbf{z}_t} \log p_t(\mathbf{z}_0, \mathbf{z}_t)\, \mathrm{d}\mathbf{z}_0$$

$$= \; \int p_t(\mathbf{z}_0 \mid \mathbf{z}_t)\, \nabla_{\mathbf{z}_t} \log p_t(\mathbf{z}_t \mid \mathbf{z}_0)\, \mathrm{d}\mathbf{z}_0 \; = \; \int p_t(\mathbf{z}_0 \mid \mathbf{z}_t)\, \frac{1}{\sigma(t)^2}\big(\alpha(t)\, \mathbf{z}_0 \;-\; \mathbf{z}_t\big)\, \mathrm{d}\mathbf{z}_0$$

$$= \; \frac{1}{\sigma(t)^2}\Big(\alpha(t)\, \mathbb{E}\big[\mathbf{z}_0 \mid \mathbf{z}_t\big] \;-\; \mathbf{z}_t\Big).$$

Rearranging completes the proof. $\qquad\square$

**Lemma A.2** (**Tweedie's Covariance Formula**). *For any distribution $p(\mathbf{z}_0)$ and*

$$p_t(\mathbf{z}_t \mid \mathbf{z}_0) \; = \; \mathcal{N}\big(\mathbf{z}_t \mid \alpha(t)\, \mathbf{z}_0,\; \sigma(t)^2\, \boldsymbol{I}\big),$$

*the posterior $p_t(\mathbf{z}_0 \mid \mathbf{z}_t)$ is also Gaussian with mean $\mathbb{E}[\mathbf{z}_0 \mid \mathbf{z}_t]$ and covariance $\mathbb{C}\mathrm{ov}[\mathbf{z}_0 \mid \mathbf{z}_t]$. These are connected to the score function $\nabla_{\mathbf{z}_t} \log p_t(\mathbf{z}_t)$ via*

$$\mathbb{C}\mathrm{ov}[\mathbf{z}_0 \mid \mathbf{z}_t] \; = \; \frac{\sigma(t)^2}{\alpha(t)^2}\Big(\boldsymbol{I} \;+\; \sigma(t)^2\, \nabla^2_{\mathbf{z}_t} \log p_t(\mathbf{z}_t)\Big). \tag{23}$$

*Proof.* We start with the Hessian of $\log p_t(\mathbf{z}_t)$:

$$\nabla^2_{\mathbf{z}_t} \log p_t(\mathbf{z}_t) = \; \nabla_{\mathbf{z}_t}\big(\nabla^\top_{\mathbf{z}_t} \log p_t(\mathbf{z}_t)\big) \; = \; \frac{\partial}{\partial \mathbf{z}_{tj}}\Big(\frac{\partial \log p_t(\mathbf{z}_t)}{\partial \mathbf{z}_{ti}}\Big)$$

$$= \; \nabla_{\mathbf{z}_t}\Big(\frac{\alpha(t)\, \mathbb{E}[\mathbf{z}_0 \mid \mathbf{z}_t] - \mathbf{z}_t}{\sigma(t)^2}\Big)^\top \; = \; \frac{1}{\sigma(t)^2} \nabla_{\mathbf{z}_t}\Big(\alpha(t)\, \mathbb{E}[\mathbf{z}_0 \mid \mathbf{z}_t] \;-\; \mathbf{z}_t\Big)^\top$$

$$= \; \frac{\alpha(t)}{\sigma(t)^2} \nabla_{\mathbf{z}_t}\big(\mathbb{E}[\mathbf{z}_0 \mid \mathbf{z}_t]^\top\big) \;-\; \frac{1}{\sigma(t)^2}\, \boldsymbol{I}$$

$$= \; \frac{\alpha(t)}{\sigma(t)^2} \int p_t(\mathbf{z}_0 \mid \mathbf{z}_t)\, \nabla_{\mathbf{z}_t} \log\Big(\frac{p_t(\mathbf{z}_t \mid \mathbf{z}_0)}{p_t(\mathbf{z}_t)}\Big)\, \mathbf{z}_0^\top\, \mathrm{d}\mathbf{z}_0 \;-\; \frac{1}{\sigma(t)^2}\, \boldsymbol{I}$$

$$= \; \Big(\frac{\alpha(t)}{\sigma(t)^2}\Big)^2 \int p_t(\mathbf{z}_0 \mid \mathbf{z}_t)\, \big(\mathbf{z}_0 - \mathbb{E}[\mathbf{z}_0 \mid \mathbf{z}_t]\big)\, \mathbf{z}_0^\top\, \mathrm{d}\mathbf{z}_0 \;-\; \frac{1}{\sigma(t)^2}\, \boldsymbol{I}$$

$$= \; \Big(\frac{\alpha(t)}{\sigma(t)^2}\Big)^2 \underbrace{\Big(\mathbb{E}[\mathbf{z}_0\, \mathbf{z}_0^\top \mid \mathbf{z}_t] \;-\; \mathbb{E}[\mathbf{z}_0 \mid \mathbf{z}_t]\, \mathbb{E}[\mathbf{z}_0 \mid \mathbf{z}_t]^\top\Big)}_{\mathbb{C}\mathrm{ov}[\mathbf{z}_0 \mid \mathbf{z}_t]} \;-\; \frac{1}{\sigma(t)^2}\, \boldsymbol{I}.$$

Solving for $\mathbb{C}\mathrm{ov}[\mathbf{z}_0 \mid \mathbf{z}_t]$ yields the stated formula. $\qquad\square$

**Lemma A.3** (Connection between Posterior Mean and Vector Field). *Let* $\mathbf{z}_t = \alpha(t)\,\mathbf{z}_0 + \sigma(t)\,\mathbf{z}_1$ *be a one-sided interpolant, where* $\mathbf{z}_0 \sim p_{data}$ *and* $\mathbf{z}_1 \sim \mathcal{N}(\mathbf{0}, \boldsymbol{I})$. *Then the vector field* $\mathbf{v}(\mathbf{z}_t, t)$ *satisfies*

$$\mathbf{v}(\mathbf{z}_t, t) = \frac{\dot{\sigma}(t)}{\sigma(t)}\,\mathbf{z}_t + \left(\dot{\alpha}(t) - \frac{\dot{\sigma}(t)\,\alpha(t)}{\sigma(t)}\right)\mathbb{E}[\mathbf{z}_0 \mid \mathbf{z}_t]. \tag{24}$$

*Proof.* Starting from the definition,

$$\mathbf{v}(\mathbf{z}_t, t) = \mathbb{E}\big[\dot{\alpha}(t)\,\mathbf{z}_0 + \dot{\sigma}(t)\,\mathbf{z}_1 \mid \mathbf{z}_t\big] = \dot{\alpha}(t)\,\mathbb{E}[\mathbf{z}_0 \mid \mathbf{z}_t] + \dot{\sigma}(t)\,\mathbb{E}[\mathbf{z}_1 \mid \mathbf{z}_t].$$

where $\dot{\sigma}(t), \dot{\alpha}(t)$ represent the first-order time derivatives of $\sigma(t)$ and $\alpha(t)$, respectively. Noting that

$$\mathbb{E}[\mathbf{z}_1 \mid \mathbf{z}_t] = \frac{\mathbf{z}_t - \alpha(t)\,\mathbb{E}[\mathbf{z}_0 \mid \mathbf{z}_t]}{\sigma(t)},$$

we obtain

$$\mathbf{v}(\mathbf{z}_t, t) = \dot{\alpha}(t)\,\mathbb{E}[\mathbf{z}_0 \mid \mathbf{z}_t] + \dot{\sigma}(t)\frac{\mathbf{z}_t - \alpha(t)\,\mathbb{E}[\mathbf{z}_0 \mid \mathbf{z}_t]}{\sigma(t)} = \frac{\dot{\sigma}(t)}{\sigma(t)}\,\mathbf{z}_t + \left(\dot{\alpha}(t) - \frac{\dot{\sigma}(t)\,\alpha(t)}{\sigma(t)}\right)\mathbb{E}[\mathbf{z}_0 \mid \mathbf{z}_t].$$

$\square$

**Lemma A.4** (Cramér–Rao inequality). *Let* $\mu(\mathrm{d}\mathbf{x}) = \exp(-\Phi(\mathbf{x}))\mathrm{d}\mathbf{x}$ *be a probability measure on* $\mathbb{R}^d$, *where* $\Phi : \mathbb{R}^d \to \mathbb{R}$ *is twice continuously differentiable* [1]. *Then for any* $f \in C^1(\mathbb{R}^d)$,

$$\mathrm{Var}_\mu(f) \geq \left\langle \mathbb{E}_\mu[\nabla f], \left(\mathbb{E}_\mu[\nabla^2\Phi]\right)^{-1}\mathbb{E}_\mu[\nabla f]\right\rangle. \tag{25}$$

*Proof.* For comprehensive proofs of the Cramér–Rao inequality, we refer readers to [77], and the references cited therein. $\square$

**Lemma A.5** (Posterior Covariance Lower Bound via Cramér–Rao). *Let* $\mathbf{z}_t = \alpha(t)\mathbf{z}_0 + \sigma(t)\mathbf{z}_1$, *where* $\mathbf{z}_0 \sim p(\mathbf{z}_0) \propto \exp(-\Phi(\mathbf{z}_0))$ *with* $\Phi \in C^2(\mathbb{R}^d)$, *and* $\mathbf{z}_1 \sim \mathcal{N}(0, \boldsymbol{I}_d)$ *are independent. Suppose:*

(i) *The likelihood is Gaussian:* $p(\mathbf{z}_t \mid \mathbf{z}_0) = \mathcal{N}(\alpha(t)\mathbf{z}_0, \sigma(t)^2\boldsymbol{I}_d)$.

(ii) $\Phi$ *is* $\gamma$-*strongly convex, i.e.,* $\nabla^2\Phi(\mathbf{z}_0) \succeq \gamma\boldsymbol{I}_d$,

*Then the posterior satisfies:*

$$\mathrm{Cov}(\mathbf{z}_0 \mid \mathbf{z}_t) \succeq \left(\gamma + \frac{\alpha(t)^2}{\sigma(t)^2}\right)^{-1}\boldsymbol{I}_d. \tag{26}$$

*Proof.* Consider assumption (i). By applying Bayes' rule, the negative log-posterior is:

$$-\log p(\mathbf{z}_0 \mid \mathbf{z}_t) = -\log p(\mathbf{z}_t \mid \mathbf{z}_0) - \log p(\mathbf{z}_0) + \mathrm{const}.$$

Taking second derivatives yields:

$$\nabla^2_{\mathbf{z}_0}[-\log p(\mathbf{z}_0 \mid \mathbf{z}_t)] = \frac{\alpha(t)^2}{\sigma(t)^2}\boldsymbol{I}_d + \nabla^2\Phi(\mathbf{z}_0).$$

Using assumption (ii), $\nabla^2\Phi(\mathbf{z}_0) \succeq \gamma\boldsymbol{I}_d$, we obtain:

$$\nabla^2_{\mathbf{z}_0}[-\log p(\mathbf{z}_0 \mid \mathbf{z}_t)] \succeq \left(\gamma + \frac{\alpha(t)^2}{\sigma(t)^2}\right)\boldsymbol{I}_d.$$

Finally, applying Lemma A.4, the Cramér–Rao bound gives:

$$\mathbb{C}\mathrm{ov}[\mathbf{z}_0 \mid \mathbf{z}_t] \succeq \left(\mathbb{E}_{\mathbf{z}_0|\mathbf{z}_t}\left[\nabla^2_{\mathbf{z}_0}(-\log p(\mathbf{z}_0 \mid \mathbf{z}_t))\right]\right)^{-1} \succeq \left(\gamma + \frac{\alpha(t)^2}{\sigma(t)^2}\right)^{-1}\boldsymbol{I}_d.$$

$\square$

---

[1] We write $\Phi \in C^2(\mathbb{R}^d)$ to denote that $\Phi$ has continuous first and second derivatives on $\mathbb{R}^d$. Likewise, $f \in C^1(\mathbb{R}^d)$ means that $f$ has a continuous gradient.

## A.1 Proof of Proposition 4.1

*Proof.* The proof follows from Appendix C of [78]. We assume that the time-dependent latent variable $\mathbf{z}_t \in \mathbb{R}^d$ is generated by a known forward process of the form:

$$\mathbf{z}_t = \alpha(t)\,\mathbf{z}_0 + \sigma(t)\,\boldsymbol{\varepsilon}, \qquad \boldsymbol{\varepsilon} \sim \mathcal{N}(\mathbf{0}, \boldsymbol{I}),$$

where $\mathbf{z}_0$ is the initial (clean) latent sample and $\alpha(t), \sigma(t)$ are scalar functions of time.

This process induces a time-indexed family of conditional distributions $\bar{p}_t(\mathbf{z} \mid \mathbf{y})$, where the expectation is taken over the joint distribution of $\mathbf{z}_0 \mid \mathbf{y}$ and the noise $\boldsymbol{\varepsilon}$.

We define the characteristic function of the marginal at time $t$ as:

$$\hat{p}_t(\mathbf{k}) := \int e^{i\mathbf{k}^\top \mathbf{z}}\, \bar{p}_t(\mathbf{z} \mid \mathbf{y})\, d\mathbf{z} = \widetilde{\mathbb{E}}\left[ e^{i\mathbf{k}^\top \mathbf{z}_t} \right], \tag{27}$$

where $\widetilde{\mathbb{E}}$ denotes expectation over the conditional joint distribution of $\mathbf{z}_0 \mid \mathbf{y}$ and $\boldsymbol{\varepsilon}$.

We now differentiate this characteristic function with respect to time:

$$\partial_t \hat{p}_t(\mathbf{k}) = \partial_t \widetilde{\mathbb{E}}\left[ e^{i\mathbf{k}^\top \mathbf{z}_t} \right] = \widetilde{\mathbb{E}}\left[ \underbrace{\frac{d}{dt} e^{i\mathbf{k}^\top \mathbf{z}_t}}_{= i\mathbf{k}^\top \dot{\mathbf{z}}_t \cdot e^{i\mathbf{k}^\top \mathbf{z}_t}} \right] = \widetilde{\mathbb{E}}\left[ i\mathbf{k}^\top \dot{\mathbf{z}}_t \cdot e^{i\mathbf{k}^\top \mathbf{z}_t} \right].$$

Since $\mathbf{z}_t$ is a deterministic function of $\mathbf{z}_0$ and $\boldsymbol{\varepsilon}$, we may condition on $\mathbf{z}_t$ and write:

$$\partial_t \hat{p}_t(\mathbf{k}) = \mathbb{E}_{\bar{p}_t(\mathbf{z})}\left[ \widetilde{\mathbb{E}}\left[ i\mathbf{k}^\top \dot{\mathbf{z}}_t \mid \mathbf{z}_t = \mathbf{z} \right] \cdot e^{i\mathbf{k}^\top \mathbf{z}} \right]$$

$$= i\mathbf{k}^\top \int \underbrace{\widetilde{\mathbb{E}}\left[ \dot{\mathbf{z}}_t \mid \mathbf{z}_t = \mathbf{z} \right]}_{:= \tilde{\mathbf{v}}_t(\mathbf{z})} \cdot e^{i\mathbf{k}^\top \mathbf{z}}\, \bar{p}_t(\mathbf{z} \mid \mathbf{y})\, d\mathbf{z}.$$

On the other hand, by the definition of the characteristic function:

$$\partial_t \hat{p}_t(\mathbf{k}) = \partial_t \int e^{i\mathbf{k}^\top \mathbf{z}}\, \bar{p}_t(\mathbf{z} \mid \mathbf{y})\, d\mathbf{z}$$

$$= \int e^{i\mathbf{k}^\top \mathbf{z}}\, \partial_t \bar{p}_t(\mathbf{z} \mid \mathbf{y})\, d\mathbf{z}.$$

Equating the two expressions for $\partial_t \hat{p}_t(\mathbf{k})$, we obtain:

$$\int e^{i\mathbf{k}^\top \mathbf{z}}\, \partial_t \bar{p}_t(\mathbf{z} \mid \mathbf{y})\, d\mathbf{z} = i\mathbf{k}^\top \int \tilde{\mathbf{v}}_t(\mathbf{z})\, e^{i\mathbf{k}^\top \mathbf{z}}\, \bar{p}_t(\mathbf{z} \mid \mathbf{y})\, d\mathbf{z}.$$

We now use the identity $\nabla_{\mathbf{z}} e^{i\mathbf{k}^\top \mathbf{z}} = i\mathbf{k} \cdot e^{i\mathbf{k}^\top \mathbf{z}}$ and integration by parts to move the gradient from $e^{i\mathbf{k}^\top \mathbf{z}}$ to the density term:

$$\int e^{i\mathbf{k}^\top \mathbf{z}}\, \partial_t \bar{p}_t(\mathbf{z} \mid \mathbf{y})\, d\mathbf{z} = -\int e^{i\mathbf{k}^\top \mathbf{z}}\, \nabla_{\mathbf{z}} \cdot (\tilde{\mathbf{v}}_t(\mathbf{z})\, \bar{p}_t(\mathbf{z} \mid \mathbf{y}))\, d\mathbf{z}.$$

Finally, since both expressions match for all $\mathbf{k}$, their inverse Fourier transforms must be equal. Hence, we conclude:

$$\partial_t \bar{p}_t(\mathbf{z} \mid \mathbf{y}) = -\nabla_{\mathbf{z}} \cdot (\tilde{\mathbf{v}}_t(\mathbf{z})\, \bar{p}_t(\mathbf{z} \mid \mathbf{y})).$$

This is the continuity (Fokker–Planck) equation with drift field $\tilde{\mathbf{v}}_t$. Therefore, integrating the ODE:

$$\frac{d\mathbf{z}_t}{dt} = -\tilde{\mathbf{v}}_t(\mathbf{z}_t),$$

backward in time from $\mathbf{z}_1 \sim \mathcal{N}(\mathbf{0}, \boldsymbol{I})$, yields samples $\mathbf{z}_0 \sim \bar{p}_0(\mathbf{z} \mid \mathbf{y})$, as desired. $\qquad\square$

## A.2 Conditional Vector Field

*Proof.* From Lemma A.3, the unconditional vector field is

$$\mathbf{v}(\mathbf{z}_t, t) = \frac{\dot{\sigma}(t)}{\sigma(t)} \mathbf{z}_t + \left( \dot{\alpha}(t) - \frac{\dot{\sigma}(t)\,\alpha(t)}{\sigma(t)} \right) \mathbb{E}[\mathbf{z}_0 \mid \mathbf{z}_t].$$

By Lemma A.1, we have

$$\mathbb{E}[\mathbf{z}_0 \mid \mathbf{z}_t] = \frac{1}{\alpha(t)} \left( \mathbf{z}_t + \sigma(t)^2 \nabla_{\mathbf{z}_t} \log p_t(\mathbf{z}_t) \right).$$

Substituting this in,

$$\mathbf{v}(\mathbf{z}_t, t) = \frac{\dot{\alpha}(t)}{\alpha(t)} \mathbf{z}_t + \frac{\sigma(t)}{\alpha(t)} \left( \dot{\alpha}(t)\,\sigma(t) - \dot{\sigma}(t)\,\alpha(t) \right) \nabla_{\mathbf{z}_t} \log p_t(\mathbf{z}_t).$$

Conditioning on $\mathbf{y}$, the score becomes

$$\nabla_{\mathbf{z}_t} \log p_t(\mathbf{z}_t \mid \mathbf{y}) = \nabla_{\mathbf{z}_t} \log p_t(\mathbf{z}_t) + \nabla_{\mathbf{z}_t} \log p_t(\mathbf{y} \mid \mathbf{z}_t).$$

Hence, the conditional vector field is

$$\mathbf{v}(\mathbf{z}_t, \mathbf{y}, t) = \mathbf{v}(\mathbf{z}_t, t) + \frac{\sigma(t)}{\alpha(t)} \left( \dot{\alpha}(t)\,\sigma(t) - \dot{\sigma}(t)\,\alpha(t) \right) \nabla_{\mathbf{z}_t} \log p_t(\mathbf{y} \mid \mathbf{z}_t).$$

Finally, choosing the linear schedule $\alpha(t) = 1 - t$ and $\sigma(t) = t$ simplifies this to

$$\mathbf{v}(\mathbf{z}_t, \mathbf{y}, t) = \mathbf{v}(\mathbf{z}_t, t) - \frac{t}{1-t} \nabla_{\mathbf{z}_t} \log p_t(\mathbf{y} \mid \mathbf{z}_t).$$

Approximating the unconditional velocity $\mathbf{v}(\mathbf{z}_t, t)$ by a parametric estimator $\mathbf{v}_\theta(\mathbf{z}_t, t)$ gives

$$\mathbf{v}(\mathbf{z}_t, \mathbf{y}, t) \approx \mathbf{v}_\theta(\mathbf{z}_t, t) - \frac{t}{1-t} \nabla_{\mathbf{z}_t} \log p_t(\mathbf{y} \mid \mathbf{z}_t).$$

$\square$

## A.3 Derivation of Posterior Mean (Eq. (14)) and Posterior Covariance (Eq. (15))

*Proof.* From Lemma A.3, we know

$$\mathbf{v}(\mathbf{z}_t, t) = \frac{\dot{\sigma}(t)}{\sigma(t)} \mathbf{z}_t + \left( \dot{\alpha}(t) - \frac{\dot{\sigma}(t)\,\alpha(t)}{\sigma(t)} \right) \mathbb{E}[\mathbf{z}_0 \mid \mathbf{z}_t].$$

With $\alpha(t) = 1 - t$ and $\sigma(t) = t$, one obtains

$$\mathbb{E}[\mathbf{z}_0 \mid \mathbf{z}_t] = \mathbf{z}_t - t\,\mathbf{v}(\mathbf{z}_t, t).$$

Meanwhile, from Lemmas A.1 and A.2, one can show

$$\mathbb{C}\mathrm{ov}[\mathbf{z}_0 \mid \mathbf{z}_t] = \frac{\sigma(t)^2}{\alpha(t)} \nabla_{\mathbf{z}_t}^T \mathbb{E}[\mathbf{z}_0 \mid \mathbf{z}_t].$$

Thus,

$$\mathbb{C}\mathrm{ov}[\mathbf{z}_0 \mid \mathbf{z}_t] = \frac{t^2}{1-t} \left( \mathbf{I} - t\,\nabla_{\mathbf{z}_t} \mathbf{v}_\theta(\mathbf{z}_t, t) \right).$$

$\square$

## A.4 Proof of Noise Conditional Score Approximation in Latent Space

*Proof.* We assume the likelihood and posterior over $\mathbf{x}_0$ are Gaussian:

$$p(\mathbf{y} \mid \mathbf{x}_0) = \mathcal{N}(\mathcal{A}\mathbf{x}_0, \sigma_{\mathbf{y}}^2 \mathbf{I}_m), \quad p(\mathbf{x}_0 \mid \mathbf{x}_t) = \mathcal{N}(\mathbf{m}, \boldsymbol{\Sigma})$$

with:

$$\mathbf{x}_0, \mathbf{x}_t \in \mathbb{R}^n, \quad \mathbf{y} \in \mathbb{R}^m, \quad \mathcal{A} \in \mathbb{R}^{m \times n}, \quad \mathbf{m} := \mathbb{E}[\mathbf{x}_0 \mid \mathbf{x}_t], \quad \boldsymbol{\Sigma} := \mathrm{Cov}[\mathbf{x}_0 \mid \mathbf{x}_t]$$

Define:
$$\mathbf{S} := \sigma_{\mathbf{y}}^2 \boldsymbol{I}_m + \mathcal{A}\boldsymbol{\Sigma}\mathcal{A}^\top \in \mathbb{R}^{m \times m}, \quad \mathbf{r} := \mathbf{y} - \mathcal{A}\mathbf{m} \in \mathbb{R}^m$$

Then:
$$p(\mathbf{y} \mid \mathbf{x}_t) = \int p(\mathbf{y} \mid \mathbf{x}_0)\, p(\mathbf{x}_0 \mid \mathbf{x}_t)\, d\mathbf{x}_0 = \mathcal{N}(\mathcal{A}\mathbf{m},\, \mathbf{S})$$

The log-likelihood becomes:
$$\log p(\mathbf{y} \mid \mathbf{x}_t) = -\frac{m}{2}\log(2\pi) - \frac{1}{2}\log\det\mathbf{S} - \frac{1}{2}\mathbf{r}^\top \mathbf{S}^{-1}\mathbf{r}$$

We compute its gradient w.r.t. $\mathbf{x}_t \in \mathbb{R}^n$ in two parts:

(1) Log-determinant term:
$$\nabla_{\mathbf{x}_t} \log\det\mathbf{S} = \text{Tr}\left(\mathbf{S}^{-1}\nabla_{\mathbf{x}_t}\mathbf{S}\right), \quad \frac{\partial\mathbf{S}}{\partial x_{t,i}} = \mathcal{A}\frac{\partial\boldsymbol{\Sigma}}{\partial x_{t,i}}\mathcal{A}^\top$$

So,
$$\nabla_{\mathbf{x}_t}\log\det\mathbf{S} = \begin{bmatrix} \text{Tr}(\mathcal{A}^\top\mathbf{S}^{-1}\mathcal{A} \cdot \frac{\partial\boldsymbol{\Sigma}}{\partial x_{t,1}}) \\ \vdots \\ \text{Tr}(\mathcal{A}^\top\mathbf{S}^{-1}\mathcal{A} \cdot \frac{\partial\boldsymbol{\Sigma}}{\partial x_{t,n}}) \end{bmatrix} \in \mathbb{R}^n$$

(2) Quadratic term:
$$\nabla_{\mathbf{x}_t}\left(\mathbf{r}^\top\mathbf{S}^{-1}\mathbf{r}\right) = -2(\nabla_{\mathbf{x}_t}\mathbf{m})^\top\mathcal{A}^\top\mathbf{S}^{-1}\mathbf{r} - \begin{bmatrix} \text{Tr}\left(\mathbf{S}^{-1}\mathbf{r}\mathbf{r}^\top\mathbf{S}^{-1} \cdot \frac{\partial\mathbf{S}}{\partial x_{t,1}}\right) \\ \vdots \\ \text{Tr}\left(\mathbf{S}^{-1}\mathbf{r}\mathbf{r}^\top\mathbf{S}^{-1} \cdot \frac{\partial\mathbf{S}}{\partial x_{t,n}}\right) \end{bmatrix} \in \mathbb{R}^n$$

Putting both together:
$$\nabla_{\mathbf{x}_t}\log p(\mathbf{y} \mid \mathbf{x}_t) = -(\nabla_{\mathbf{x}_t}\mathbf{m})^\top\mathcal{A}^\top\mathbf{S}^{-1}\mathbf{r} - \frac{1}{2}\begin{bmatrix} \text{Tr}\left(\mathcal{A}^\top\mathbf{S}^{-1}\mathcal{A} \cdot \frac{\partial\boldsymbol{\Sigma}}{\partial x_{t,1}}\right) + \text{Tr}\left(\mathbf{S}^{-1}\mathbf{r}\mathbf{r}^\top\mathbf{S}^{-1} \cdot \frac{\partial\mathbf{S}}{\partial x_{t,1}}\right) \\ \vdots \\ \text{Tr}\left(\mathcal{A}^\top\mathbf{S}^{-1}\mathcal{A} \cdot \frac{\partial\boldsymbol{\Sigma}}{\partial x_{t,n}}\right) + \text{Tr}\left(\mathbf{S}^{-1}\mathbf{r}\mathbf{r}^\top\mathbf{S}^{-1} \cdot \frac{\partial\mathbf{S}}{\partial x_{t,n}}\right) \end{bmatrix}$$

Assuming $\mathbb{C}\text{ov}[\mathbf{x}_0 \mid \mathbf{x}_t]$ is slowly varying w.r.t. $\mathbf{x}_t$, we neglect its gradient, yielding the approximation:
$$\nabla_{\mathbf{x}_t}\log p(\mathbf{y} \mid \mathbf{x}_t) \approx \left(\nabla_{\mathbf{x}_t}\mathbf{m}\right)^\top \mathcal{A}^\top\mathbf{S}^{-1}\mathbf{r}$$
$$= \left(\nabla_{\mathbf{x}_t}\mathbb{E}[\mathbf{x}_0 \mid \mathbf{x}_t]\right)^\top \mathcal{A}^\top \left(\sigma_{\mathbf{y}}^2\boldsymbol{I}_m + \mathcal{A}\boldsymbol{\Sigma}\mathcal{A}^\top\right)^{-1}\left(\mathbf{y} - \mathcal{A}\,\mathbb{E}[\mathbf{x}_0 \mid \mathbf{x}_t]\right)$$

Now, we aim to derive a principled extension of this formula to the *latent space*. Let $\mathbf{z}_0$ and $\mathbf{z}_t$ be latent variables, and $\mathcal{D}_{\boldsymbol{\varphi}}$ be a (generally nonlinear) decoder such that $\mathbf{x}_0 = \mathcal{D}_{\boldsymbol{\varphi}}(\mathbf{z}_0)$. Then the likelihood can be written as:
$$p(\mathbf{y} \mid \mathbf{z}_0) = \mathcal{N}(\mathbf{y}; \mathcal{A}\mathcal{D}_{\boldsymbol{\varphi}}(\mathbf{z}_0), \sigma_{\mathbf{y}}^2\boldsymbol{I})$$

which is a Gaussian with **nonlinear mean**. We further assume:
$$p(\mathbf{z}_0 \mid \mathbf{z}_t) = \mathcal{N}(\bar{\mathbf{z}}_0, \boldsymbol{\Sigma}_z), \quad \text{where } \bar{\mathbf{z}}_0 := \mathbb{E}[\mathbf{z}_0 \mid \mathbf{z}_t], \quad \boldsymbol{\Sigma}_z := \mathbb{C}\text{ov}[\mathbf{x}_0 \mid \mathbf{z}_t]$$

Since $\mathcal{D}_{\boldsymbol{\varphi}}$ is nonlinear, $p(\mathbf{y} \mid \mathbf{z}_t)$ is not Gaussian. However, we approximate $\mathcal{D}_{\boldsymbol{\varphi}}(\mathbf{z}_0)$ via a first-order Taylor expansion around $\bar{\mathbf{z}}_0$:
$$\mathcal{D}_{\boldsymbol{\varphi}}(\mathbf{z}_0) \approx \mathcal{D}_{\boldsymbol{\varphi}}(\bar{\mathbf{z}}_0) + J_{\mathcal{D}}(\bar{\mathbf{z}}_0)(\mathbf{z}_0 - \bar{\mathbf{z}}_0)$$

where $J_{\mathcal{D}}(\bar{\mathbf{z}}_0)$ is the Jacobian of $\mathcal{D}_{\boldsymbol{\varphi}}$ at $\bar{\mathbf{z}}_0$.

Using this approximation, the distribution over $\mathbf{x}_0$ becomes approximately Gaussian:

$$\mathbb{E}[\mathbf{x}_0 \mid \mathbf{z}_t] \approx \mathcal{D}_{\boldsymbol{\varphi}}(\bar{\mathbf{z}}_0), \quad \mathbb{C}\mathrm{ov}[\mathbf{x}_0 \mid \mathbf{z}_t] \approx J_{\mathcal{D}}(\bar{\mathbf{z}}_0)\,\boldsymbol{\Sigma}_z\,J_{\mathcal{D}}(\bar{\mathbf{z}}_0)^{\top}$$

We now apply the same posterior score approximation used in pixel space, but using the decoded mean and propagated covariance:

$$\nabla_{\mathbf{z}_t} \log p(\mathbf{y} \mid \mathbf{z}_t) \approx (\nabla_{\mathbf{z}_t}\bar{\mathbf{z}}_0)^{\top} J_{\mathcal{D}}(\bar{\mathbf{z}}_0)^{\top} \mathcal{A}^{\top} \left( \sigma_{\mathbf{y}}^2 \boldsymbol{I} + \mathcal{A} J_{\mathcal{D}}(\bar{\mathbf{z}}_0)\,\boldsymbol{\Sigma}_z\,J_{\mathcal{D}}(\bar{\mathbf{z}}_0)^{\top} \mathcal{A}^{\top} \right)^{-1} (\mathbf{y} - \mathcal{A}\mathcal{D}_{\boldsymbol{\varphi}}(\bar{\mathbf{z}}_0))$$

This expression is now a well-defined and justified approximation to the posterior score in latent space, based on first-order decoder linearization and Gaussian propagation. $\qquad\square$

## A.5 Proof of Proposition 4.3

*Proof.* We aim to compute the Jacobian $\nabla_{\mathbf{z}_t} \mathbf{v}(\mathbf{z}_t, t)$. From Lemma A.3, we have:

$$\mathbf{v}(\mathbf{z}_t, t) = \frac{\dot{\sigma}(t)}{\sigma(t)}\, \mathbf{z}_t + \left( \dot{\alpha}(t) - \frac{\dot{\sigma}(t)\alpha(t)}{\sigma(t)} \right) \mathbb{E}[\mathbf{z}_0 \mid \mathbf{z}_t].$$

Taking the gradient with respect to $\mathbf{z}_t$, we obtain:

$$\nabla_{\mathbf{z}_t} \mathbf{v}(\mathbf{z}_t, t) = \frac{\dot{\sigma}(t)}{\sigma(t)}\, \boldsymbol{I} + \left( \dot{\alpha}(t) - \frac{\dot{\sigma}(t)\alpha(t)}{\sigma(t)} \right) \nabla_{\mathbf{z}_t} \mathbb{E}[\mathbf{z}_0 \mid \mathbf{z}_t].$$

From Lemma A.1, the posterior mean is given by:

$$\mathbb{E}[\mathbf{z}_0 \mid \mathbf{z}_t] = \frac{1}{\alpha(t)} \left( \mathbf{z}_t + \sigma(t)^2 \nabla_{\mathbf{z}_t} \log p_t(\mathbf{z}_t) \right),$$

and therefore its gradient is:

$$\nabla_{\mathbf{z}_t} \mathbb{E}[\mathbf{z}_0 \mid \mathbf{z}_t] = \frac{1}{\alpha(t)} \left( \boldsymbol{I} + \sigma(t)^2 \nabla_{\mathbf{z}_t}^2 \log p_t(\mathbf{z}_t) \right).$$

Substituting this into the expression for the Jacobian, we get:

$$\nabla_{\mathbf{z}_t} \mathbf{v}(\mathbf{z}_t, t) = \frac{\dot{\sigma}(t)}{\sigma(t)}\, \boldsymbol{I} + \left( \dot{\alpha}(t) - \frac{\dot{\sigma}(t)\alpha(t)}{\sigma(t)} \right) \cdot \frac{1}{\alpha(t)} \left( \boldsymbol{I} + \sigma(t)^2 \nabla_{\mathbf{z}_t}^2 \log p_t(\mathbf{z}_t) \right).$$

Applying Lemma A.2, the Hessian of the log-density can be expressed as:

$$\sigma(t)^2 \nabla_{\mathbf{z}_t}^2 \log p_t(\mathbf{z}_t) = \frac{\alpha(t)^2}{\sigma(t)^2} \mathbb{C}\mathrm{ov}[\mathbf{z}_0 \mid \mathbf{z}_t] - \boldsymbol{I},$$

which implies:

$$\boldsymbol{I} + \sigma(t)^2 \nabla_{\mathbf{z}_t}^2 \log p_t(\mathbf{z}_t) = \frac{\alpha(t)^2}{\sigma(t)^2} \mathbb{C}\mathrm{ov}[\mathbf{z}_0 \mid \mathbf{z}_t].$$

Therefore, the Jacobian becomes:

$$\nabla_{\mathbf{z}_t} \mathbf{v}(\mathbf{z}_t, t) = \frac{\dot{\sigma}(t)}{\sigma(t)}\, \boldsymbol{I} + \left( \dot{\alpha}(t) - \frac{\dot{\sigma}(t)\alpha(t)}{\sigma(t)} \right) \cdot \frac{\alpha(t)}{\sigma(t)^2} \mathbb{C}\mathrm{ov}[\mathbf{z}_0 \mid \mathbf{z}_t].$$

By Lemma A.4, if $p(\mathbf{z}_0)$ is $\gamma$-semi-log-convex, then the Cramér–Rao inequality yields:

$$\mathbb{C}\mathrm{ov}[\mathbf{z}_0 \mid \mathbf{z}_t] \succeq \left( \gamma + \frac{\alpha(t)^2}{\sigma(t)^2} \right)^{-1} \boldsymbol{I}_d.$$

Substituting this bound into the Jacobian expression gives:

$$\nabla_{\mathbf{z}_t} \mathbf{v}(\mathbf{z}_t, t) \succeq \left( \frac{\dot{\sigma}(t)}{\sigma(t)} + \left( \dot{\alpha}(t) - \frac{\dot{\sigma}(t)\alpha(t)}{\sigma(t)} \right) \cdot \frac{\alpha(t)}{\sigma(t)^2} \cdot \left( \gamma + \frac{\alpha(t)^2}{\sigma(t)^2} \right)^{-1} \right) \boldsymbol{I}_d. \qquad (28)$$

This expression simplifies to:

$$\nabla_{\mathbf{z}_t} \mathbf{v}(\mathbf{z}_t, t) \succeq \left( \frac{\gamma \, \sigma(t) \, \dot{\sigma}(t) + \alpha(t) \, \dot{\alpha}(t)}{\gamma \, \sigma(t)^2 + \alpha(t)^2} \right) \mathbf{I}_d. \tag{29}$$

Finally, observe that the right-hand side can be written as:

$$\eta(t) := \frac{\mathrm{d}}{\mathrm{d}t} \left( \tfrac{1}{2} \log \left( \alpha(t)^2 + \gamma \, \sigma(t)^2 \right) \right) = \frac{\gamma \, \sigma(t) \, \dot{\sigma}(t) + \alpha(t) \, \dot{\alpha}(t)}{\gamma \, \sigma(t)^2 + \alpha(t)^2},$$

$$\nabla_{\mathbf{z}_t} \mathbf{v}(\mathbf{z}_t, t) \succeq \eta(t) \mathbf{I}_d.$$

Now, assume the posterior covariance is positive semidefinite:

$$\mathbb{C}\mathrm{ov}[\mathbf{z}_0 \mid \mathbf{z}_t] \succeq 0.$$

Define the scalar coefficient

$$c(t) := \left( \dot{\alpha}(t) - \frac{\dot{\sigma}(t)\alpha(t)}{\sigma(t)} \right) \cdot \frac{\alpha(t)}{\sigma(t)^2}.$$

Then the Jacobian simplifies to

$$\nabla_{\mathbf{z}_t} \mathbf{v}(\mathbf{z}_t, t) = \frac{\dot{\sigma}(t)}{\sigma(t)} \cdot \mathbf{I} + c(t) \cdot \mathbb{C}\mathrm{ov}[\mathbf{z}_0 \mid \mathbf{z}_t].$$

Since $\mathbb{C}\mathrm{ov}[\mathbf{z}_0 \mid \mathbf{z}_t] \succeq 0$, the sign of $c(t)$ determines the direction of the inequality:

- If $c(t) \geq 0$, then $c(t) \cdot \mathbb{C}\mathrm{ov}[\mathbf{z}_0 \mid \mathbf{z}_t] \succeq 0$, and

$$\nabla_{\mathbf{z}_t} \mathbf{v}(\mathbf{z}_t, t) \succeq \frac{\dot{\sigma}(t)}{\sigma(t)} \cdot \mathbf{I}.$$

- If $c(t) < 0$, then $c(t) \cdot \mathbb{C}\mathrm{ov}[\mathbf{z}_0 \mid \mathbf{z}_t] \preceq 0$, and

$$\nabla_{\mathbf{z}_t} \mathbf{v}(\mathbf{z}_t, t) \preceq \frac{\dot{\sigma}(t)}{\sigma(t)} \cdot \mathbf{I}.$$

In particular, for the common case where $\alpha(t) = 1 - t$ and $\sigma(t) = t$, we have $c(t) < 0$ for all $t \in (0, 1]$, and thus the Jacobian is upper bounded. This completes the proof. $\square$

## A.6 Proof of Proposition 4.4

**Proposition A.6** (**Optimal Vector Field**). *Let $\mathbf{z}_0 \sim \mathcal{N}(\mathbf{0}, \sigma_{latr}^2 \, \mathbf{I})$ and $\mathbf{z}_1 \sim \mathcal{N}(\mathbf{0}, \mathbf{I})$ be independent. Define $\mathbf{z}_t = (1 - t) \, \mathbf{z}_0 + t \, \mathbf{z}_1$ for $t \in [0, 1]$. The optimal vector field $\mathbf{v}^\star(\mathbf{z}_t, t)$ that minimizes*

$$\arg \min_{\mathbf{v}} \ \mathbb{E}\big[\|\mathbf{v}(\mathbf{z}_t, t) - (\mathbf{z}_0 - \mathbf{z}_1)\|^2\big]$$

*is*

$$\mathbf{v}^\star(\mathbf{z}_t, t) \ = \ \frac{(1 - t) \, \sigma_{latr}^2 \ - \ t}{(1 - t)^2 \, \sigma_{latr}^2 + t^2} \, \mathbf{z}_t.$$

*Proof.* Since the loss function is quadratic and $\mathbf{v}^\star(\mathbf{z}_t, t)$ depends only on $\mathbf{z}_t$ and $t$, the optimal vector field is the conditional expectation:

$$\mathbf{v}^\star(\mathbf{z}_t, t) = \mathbb{E}\left[\mathbf{z}_0 - \mathbf{z}_1 | \mathbf{z}_t\right].$$

Given that $\mathbf{z}_0$ and $\mathbf{z}_1$ are independent Gaussian random variables, and $\mathbf{z}_t$ is a linear combination of $\mathbf{z}_0$ and $\mathbf{z}_1$, the joint distribution of $\mathbf{z}_0$, $\mathbf{z}_1$, and $\mathbf{z}_t$ is multivariate Gaussian. We will compute $\mathbb{E}\left[\mathbf{z}_0 | \mathbf{z}_t\right]$ and $\mathbb{E}\left[\mathbf{z}_1 | \mathbf{z}_t\right]$ using the properties of multivariate normal distributions.

First, we identify the covariance matrices:

$$\boldsymbol{\Sigma}_{\mathbf{z}_0 \mathbf{z}_0} = \sigma_{latr}^2 \mathbf{I}, \quad \boldsymbol{\Sigma}_{\mathbf{z}_1 \mathbf{z}_1} = \mathbf{I}, \quad \boldsymbol{\Sigma}_{\mathbf{z}_0 \mathbf{z}_1} = \mathbf{0},$$

since $\mathbf{z}_0$ and $\mathbf{z}_1$ are independent.

Next, compute the covariance between $\mathbf{z}_0$ and $\mathbf{z}_t$:

$$\boldsymbol{\Sigma}_{\mathbf{z}_0 \mathbf{z}_t} = \mathbb{E}\left[\mathbf{z}_0 \mathbf{z}_t^\top\right] = (1-t)\mathbb{E}\left[\mathbf{z}_0 \mathbf{z}_0^\top\right] + t\mathbb{E}\left[\mathbf{z}_0 \mathbf{z}_1^\top\right] = (1-t)\sigma_{latr}^2 \boldsymbol{I}.$$

Similarly, the covariance between $\mathbf{z}_1$ and $\mathbf{z}_t$:

$$\boldsymbol{\Sigma}_{\mathbf{z}_1 \mathbf{z}_t} = \mathbb{E}\left[\mathbf{z}_1 \mathbf{z}_t^\top\right] = (1-t)\mathbb{E}\left[\mathbf{z}_1 \mathbf{z}_0^\top\right] + t\mathbb{E}\left[\mathbf{z}_1 \mathbf{z}_1^\top\right] = t\boldsymbol{I}.$$

The variance of $\mathbf{z}_t$ is:

$$\boldsymbol{\Sigma}_{\mathbf{z}_t \mathbf{z}_t} = (1-t)^2 \boldsymbol{\Sigma}_{\mathbf{z}_0 \mathbf{z}_0} + t^2 \boldsymbol{\Sigma}_{\mathbf{z}_1 \mathbf{z}_1} = (1-t)^2 \sigma_{latr}^2 \boldsymbol{I} + t^2 \boldsymbol{I} = \left((1-t)^2 \sigma_{latr}^2 + t^2\right) \boldsymbol{I} = s^2 \boldsymbol{I},$$

where $s^2 = (1-t)^2 \sigma_{latr}^2 + t^2$.

The joint covariance matrix of $\mathbf{z}_0$, $\mathbf{z}_1$, and $\mathbf{z}_t$ is:

$$\boldsymbol{\Sigma}_{\mathbf{w}} = \begin{bmatrix} \boldsymbol{\Sigma}_{\mathbf{z}_0 \mathbf{z}_0} & \boldsymbol{\Sigma}_{\mathbf{z}_0 \mathbf{z}_1} & \boldsymbol{\Sigma}_{\mathbf{z}_0 \mathbf{z}_t} \\ \boldsymbol{\Sigma}_{\mathbf{z}_1 \mathbf{z}_0} & \boldsymbol{\Sigma}_{\mathbf{z}_1 \mathbf{z}_1} & \boldsymbol{\Sigma}_{\mathbf{z}_1 \mathbf{z}_t} \\ \boldsymbol{\Sigma}_{\mathbf{z}_t \mathbf{z}_0} & \boldsymbol{\Sigma}_{\mathbf{z}_t \mathbf{z}_1} & \boldsymbol{\Sigma}_{\mathbf{z}_t \mathbf{z}_t} \end{bmatrix} = \begin{bmatrix} \sigma_{latr}^2 \boldsymbol{I} & \mathbf{0} & (1-t)\sigma_{latr}^2 \boldsymbol{I} \\ \mathbf{0} & \boldsymbol{I} & t\boldsymbol{I} \\ (1-t)\sigma_{latr}^2 \boldsymbol{I} & t\boldsymbol{I} & s^2 \boldsymbol{I} \end{bmatrix}.$$

Using the properties of conditional expectations for multivariate normals, we compute the conditional expectations:

$$\mathbb{E}\left[\mathbf{z}_0 | \mathbf{z}_t\right] = \boldsymbol{\Sigma}_{\mathbf{z}_0 \mathbf{z}_t}\left(\boldsymbol{\Sigma}_{\mathbf{z}_t \mathbf{z}_t}\right)^{-1} \mathbf{z}_t = \frac{(1-t)\sigma_{latr}^2}{s^2} \mathbf{z}_t,$$

$$\mathbb{E}\left[\mathbf{z}_1 | \mathbf{z}_t\right] = \boldsymbol{\Sigma}_{\mathbf{z}_1 \mathbf{z}_t}\left(\boldsymbol{\Sigma}_{\mathbf{z}_t \mathbf{z}_t}\right)^{-1} \mathbf{z}_t = \frac{t}{s^2} \mathbf{z}_t.$$

Therefore, the optimal vector field is:

$$\mathbf{v}^\star(\mathbf{z}_t, t) = \mathbb{E}\left[\mathbf{z}_0 - \mathbf{z}_1 | \mathbf{z}_t\right] = \mathbb{E}\left[\mathbf{z}_0 | \mathbf{z}_t\right] - \mathbb{E}\left[\mathbf{z}_1 | \mathbf{z}_t\right] = \left(\frac{(1-t)\sigma_{latr}^2}{s^2} - \frac{t}{s^2}\right)\mathbf{z}_t = \frac{(1-t)\sigma_{latr}^2 - t}{(1-t)^2 \sigma_{latr}^2 + t^2} \mathbf{z}_t.$$

$\square$

## A.7   Proof of Covariance in $\Pi$GDM

($\Pi$GDM [22]). *Let $\mathbf{x}_0 \sim \mathcal{N}(\mathbf{0}, \sigma_{data}^2 \boldsymbol{I})$, and consider the forward process*

$$\mathbf{x}_t = \alpha(t)\mathbf{x}_0 + \sigma(t)\mathbf{x}_1, \quad \mathbf{x}_1 \sim \mathcal{N}(\mathbf{0}, \boldsymbol{I}).$$

*Then the conditional covariance $\mathbb{C}\mathrm{ov}[\mathbf{x}_0 | \mathbf{x}_t]$ is*

$$\mathbb{C}\mathrm{ov}[\mathbf{x}_0 | \mathbf{x}_t] = \frac{\sigma_{data}^2 \sigma(t)^2}{\alpha(t)^2 \sigma_{data}^2 + \sigma(t)^2} \boldsymbol{I}. \tag{30}$$

*Proof.* Again $\mathbf{x}_0$ and $\mathbf{x}_1$ are independent Gaussians, and $\mathbf{x}_t$ is a linear combination. Thus $\mathbf{x}_0, \mathbf{x}_t$ are jointly Gaussian with

$$\mathbb{C}\mathrm{ov}[\mathbf{x}_t] = \alpha(t)^2 \sigma_{data}^2 \boldsymbol{I} + \sigma(t)^2 \boldsymbol{I},$$

and

$$\mathbb{C}\mathrm{ov}[\mathbf{x}_t, \mathbf{x}_0] = \mathbb{E}\left[(\mathbf{x}_t - \mathbb{E}[\mathbf{x}_t])(\mathbf{x}_0 - \mathbb{E}[\mathbf{x}_0])^\top\right] = \alpha(t)\mathbb{C}\mathrm{ov}[\mathbf{x}_0] = \alpha(t)\sigma_{data}^2 \boldsymbol{I}.$$

Using the standard formula for conditional covariances in a Gaussian,

$$\mathbb{C}\mathrm{ov}[\mathbf{x}_0 | \mathbf{x}_t] = \mathbb{C}\mathrm{ov}[\mathbf{x}_0] - \mathbb{C}\mathrm{ov}[\mathbf{x}_0, \mathbf{x}_t]\mathbb{C}\mathrm{ov}[\mathbf{x}_t]^{-1}\mathbb{C}\mathrm{ov}[\mathbf{x}_t, \mathbf{x}_0].$$

Since $\mathbb{C}\mathrm{ov}[\mathbf{x}_0] = \sigma_{data}^2 \boldsymbol{I}$, one obtains

$$\mathbb{C}\mathrm{ov}[\mathbf{x}_0 | \mathbf{x}_t] = \sigma_{data}^2 \boldsymbol{I} - \left(\alpha(t)\sigma_{data}^2 \boldsymbol{I}\right)\left(\alpha(t)^2 \sigma_{data}^2 + \sigma(t)^2\right)^{-1}\left(\alpha(t)\sigma_{data}^2 \boldsymbol{I}\right),$$

which simplifies to

$$\mathbb{C}\mathrm{ov}[\mathbf{x}_0 | \mathbf{x}_t] = \frac{\sigma_{data}^2 \sigma(t)^2}{\alpha(t)^2 \sigma_{data}^2 + \sigma(t)^2} \boldsymbol{I}.$$

$\square$

# B    Closed-form Solutions for computing vector V in Eq. (13)

In this section, we derive efficient closed-form expressions for computing the vector $\mathbf{v}$ under the assumption of isotropic posterior covariance, i.e. $\mathbb{Cov}[\mathbf{z}_0 \mid \mathbf{z}_t] = r(t)^2 \, \mathbf{I}$. We begin by introducing essential notation.

**Notations.**

- Let $\mathbf{m} \in \{0,1\}^{d \times 1}$ represent the sampling positions in an image or signal.

- The downsampling operator associated with $\mathbf{m}$ is $\mathbf{D_m} \in \{0,1\}^{\|\mathbf{m}\|_0 \times d}$. It selects only those rows (i.e. entries) of a vector or matrix corresponding to the non-zero entries of $\mathbf{m}$. For example, $s$-fold downsampling with evenly spaced ones is denoted $\mathbf{D}_{\downarrow s}$.

- $\mathbf{D}_{\Downarrow s}$ represents a *distinct block* downsampler, which averages $s$ blocks (each of size $d/s$) from a vector.

- $\mathcal{F}$ is the (unitary) Fourier transform matrix of dimension $d \times d$, and $\mathcal{F}_{\downarrow s}$ is the analogous transform matrix for signals of dimension $d/s$.

- $\hat{\mathbf{v}}$ denotes the Fourier transform of the vector $\mathbf{v}$, and $\bar{\mathbf{v}}$ denotes its complex conjugate.

- The notation $\odot$ refers to element-wise (Hadamard) multiplication. Divisions such as '$/$' or '$\div$' also apply element-wise when the vectors/matrices match in dimension.

**Lemma B.1** (Downsampling Equivalence). *Standard $s$-fold downsampling in the spatial domain is equivalent to $s$-fold* block *downsampling in the frequency domain. Concretely,*

$$\mathbf{D}_{\Downarrow s} \;=\; \mathcal{F}_{\downarrow s} \, \mathbf{D}_{\downarrow s} \, \mathcal{F}^{-1}.$$

*Proof.* Please see [79, 29] for details. □

## B.1    Image Inpainting

The observation model for image inpainting can be written as

$$\mathbf{y} \;=\; \underbrace{\mathbf{D_m}}_{=\,\mathcal{A}} \mathbf{x}_0 \;+\; \mathbf{n}, \tag{31}$$

where $\mathbf{n}$ is noise. A convenient *zero-filling* version of $\mathbf{y}$ can be defined as

$$\tilde{\mathbf{y}} \;=\; \mathbf{D_m^\top} \mathbf{y} \;=\; \mathbf{m} \odot (\mathbf{x}_0 \;+\; \tilde{\mathbf{n}}), \quad \tilde{\mathbf{n}} \sim \mathcal{N}(\mathbf{0}, \mathbf{I}).$$

The closed-form solution for $\mathbf{v}$ in image inpainting is then

$$\mathbf{v} \;=\; \frac{\tilde{\mathbf{y}} \;-\; \big(\mathbf{m} \odot \mathcal{D}_{\boldsymbol{\varphi}}(\mathbb{E}[\mathbf{z}_0 \mid \mathbf{z}_t])\big)}{\sigma_{\mathbf{y}}^2 \;+\; r(t)^2}.$$

*Proof.* Starting with the more general form,

$$\mathbf{v} = \mathbf{D_m^\top} \Big(\sigma_{\mathbf{y}}^2 \, \mathbf{I} \;+\; r(t)^2 \, \mathbf{D_m D_m^\top}\Big)^{-1} \Big(\mathbf{y} \;-\; \mathbf{D_m} \, \mathcal{D}_{\boldsymbol{\varphi}}(\mathbb{E}[\mathbf{z}_0 \mid \mathbf{z}_t])\Big).$$

Since $\mathbf{D_m D_m^\top} = \mathbf{I}$ on the support of $\mathbf{y}$ and $\big(\sigma_{\mathbf{y}}^2 \, \mathbf{I} + r(t)^2 \, \mathbf{I}\big)^{-1} = 1/(\sigma_{\mathbf{y}}^2 + r(t)^2)$, it simplifies to

$$\mathbf{v} = \frac{\mathbf{D_m^\top}\big(\mathbf{y} \;-\; \mathbf{D_m} \, \mathcal{D}_{\boldsymbol{\varphi}}(\mathbb{E}[\mathbf{z}_0 \mid \mathbf{z}_t])\big)}{\sigma_{\mathbf{y}}^2 + r(t)^2} = \frac{\tilde{\mathbf{y}} \;-\; \mathbf{m} \odot \mathcal{D}_{\boldsymbol{\varphi}}\big(\mathbb{E}[\mathbf{z}_0 \mid \mathbf{z}_t]\big)}{\sigma_{\mathbf{y}}^2 + r(t)^2}.$$

Recalling that $\tilde{\mathbf{y}} = \mathbf{D_m^\top} \mathbf{y} = \mathbf{m} \odot (\mathbf{x}_0 + \tilde{\mathbf{n}})$, we arrive at the stated closed-form solution. □

## B.2 Image Deblurring

For image deblurring, the observation model is

$$\mathbf{y} = \mathbf{x}_0 * \mathbf{k} + \mathbf{n}, \tag{32}$$

where $\mathbf{k}$ is the blurring kernel and $*$ is the circular convolution operator. Using the Fourier transform, this can be written as

$$\mathbf{y} = \underbrace{\mathcal{F}^{-1} \operatorname{diag}(\hat{\mathbf{k}}) \mathcal{F}}_{= \mathcal{A}} \mathbf{x}_0 + \mathbf{n},$$

where $\hat{\mathbf{k}}$ is the DFT of $\mathbf{k}$. Under isotropic-covariance assumption, the closed-form solution for $\mathbf{v}$ is

$$\mathbf{v} = \mathcal{F}^{-1}\left(\bar{\hat{\mathbf{k}}} \odot \frac{\mathcal{F}\left[\mathbf{y} - \mathcal{A}\,\mathcal{D}_\varphi\left(\mathbb{E}[\mathbf{z}_0 \mid \mathbf{z}_t]\right)\right]}{\sigma_\mathbf{y}^2 + r(t)^2 \, |\hat{\mathbf{k}}|^2}\right).$$

*Proof.* Because $\mathcal{A}$ is a real linear operator of convolution type, we have $\mathcal{A}^T = \mathcal{A}^H$. Thus,

$$\mathbf{v} = \mathcal{A}^\top \left(\sigma_\mathbf{y}^2\, \boldsymbol{I} + r(t)^2\, \mathcal{A}\mathcal{A}^\top\right)^{-1}\left[\mathbf{y} - \mathcal{A}\,\mathcal{D}_\varphi\left(\mathbb{E}[\mathbf{z}_0 \mid \mathbf{z}_t]\right)\right].$$

Substituting $\mathcal{A} = \mathcal{F}^{-1}\operatorname{diag}(\hat{\mathbf{k}})\,\mathcal{F}$ and simplifying in the Fourier domain (using the diagonal structure in frequency space), one obtains

$$\mathbf{v} = \mathcal{F}^{-1}\left(\bar{\hat{\mathbf{k}}} \odot \frac{\mathcal{F}[\mathbf{y} - \mathcal{A}\,\mathcal{D}_\varphi(\cdot)]}{\sigma_\mathbf{y}^2 + r(t)^2\, |\hat{\mathbf{k}}|^2}\right).$$

$\square$

## B.3 Super-Resolution

Following [79], the super-resolution observation model is approximately

$$\mathbf{y} = \left(\mathbf{x}_0 * \mathbf{k}\right)_{\downarrow s} + \mathbf{n}, \tag{33}$$

which, in "canonical form," can be written as

$$\mathbf{y} = \mathbf{D}_{\downarrow s}\, \mathcal{F}^{-1} \operatorname{diag}(\hat{\mathbf{k}})\, \mathcal{F}\, \mathbf{x}_0 + \mathbf{n}.$$

Hence, $\mathcal{A} = \mathbf{D}_{\downarrow s}\, \mathcal{F}^{-1}\operatorname{diag}(\hat{\mathbf{k}})\, \mathcal{F}$. The closed-form solution under the isotropic assumption is

$$\mathbf{v} = \mathcal{F}^{-1}\left(\bar{\hat{\mathbf{k}}} \odot_s \frac{\mathcal{F}_{\downarrow s}\left[\mathbf{y} - \mathcal{A}\,\mathcal{D}_\varphi\left(\mathbb{E}[\mathbf{z}_0 \mid \mathbf{z}_t]\right)\right]}{\sigma_\mathbf{y}^2 + r(t)^2\, \left(\bar{\hat{\mathbf{k}}} \odot \hat{\mathbf{k}}\right)_{\Downarrow s}}\right),$$

where $\odot_s$ denotes block-wise Hadamard multiplication.

*Proof.* Since $\mathcal{A}^T = \mathcal{A}^H$ and

$$\mathcal{A} = \mathbf{D}_{\downarrow s}\, \mathcal{F}^{-1} \operatorname{diag}(\hat{\mathbf{k}})\, \mathcal{F},$$

we get $\mathcal{A}^\top = \mathcal{F}^\top \operatorname{diag}(\bar{\hat{\mathbf{k}}}) \left(\mathcal{F}^{-1}\right)^\top \mathbf{D}_{\downarrow s}^\top$. Applying Lemma B.1, namely $\mathbf{D}_{\downarrow s}\, \mathcal{F}^{-1} = \mathcal{F}_{\downarrow s}^{-1} \mathbf{D}_{\Downarrow s}$, and its conjugate-transpose version, reduces the inverse $\left(\sigma_\mathbf{y}^2\, \boldsymbol{I} + r(t)^2\, \mathcal{A}\mathcal{A}^\top\right)^{-1}$ to diagonal form in the "downsampled" Fourier domain. One obtains

$$\mathbf{v} = \mathcal{F}^{-1}\left(\bar{\hat{\mathbf{k}}} \odot_s \frac{\mathcal{F}_{\downarrow s}[\mathbf{y} - \mathcal{A}\,\mathcal{D}_\varphi(\cdot)]}{\sigma_\mathbf{y}^2 + r(t)^2\, \left(\hat{\mathbf{k}} \odot \bar{\hat{\mathbf{k}}}\right)_{\Downarrow s}}\right).$$

$\square$

# C  Implementation details

---

**Algorithm 1 LFlow** Sampling: Posterior-Guided Latent ODE Inference for Linear Inverse Problems

---

1: **Input:** measurements $\mathbf{y}$, encoder $\mathcal{E}_\phi$, decoder $\mathcal{D}_\varphi$, forward operator $\mathcal{A}$, pre-trained vector field $\mathbf{v}_\theta(\cdot, t)$ for $t \in [t_s, 0]$, $t_s = 0.8$, and $K = 2$.

2: **Initialize.** $\mathbf{z}_{t_s} \leftarrow (1 - t_s)\, \mathcal{E}_\phi(\mathbf{y}) + t_s\, \mathbf{z}_1, \quad \mathbf{z}_1 \sim \mathcal{N}(0, \boldsymbol{I})$

3: **for** $t = t_s$ **down to** $0$ **do**
4: $\quad \bar{\mathbf{z}}_0 \leftarrow \mathbf{z}_t - t\, \mathbf{v}_\theta(\mathbf{z}_t, t)$ $\hfill \triangleright$ Posterior mean prediction Eq. (14)
5: $\quad r^2(t) \leftarrow \frac{t^2}{1-t}\left(1 - t\, \nabla_{\mathbf{z}_t} \mathbf{v}^\star(\mathbf{z}_t, t)\right)$ $\hfill \triangleright$ Posterior covariance estimation Eq. (18)
6: $\quad \nabla_{\mathbf{z}_t} \log p(\mathbf{y} \mid \mathbf{z}_t) \leftarrow \left(\nabla_{\mathbf{z}_t} \mathcal{D}_\varphi(\bar{\mathbf{z}}_0)\right)^\top \mathcal{A}^\top \frac{(\mathbf{y} - \mathcal{A}\mathcal{D}_\varphi(\bar{\mathbf{z}}_0))}{(\sigma_\mathbf{y}^2 \boldsymbol{I} + r^2(t) \mathcal{A}\,\mathcal{A}^\top)}$ $\hfill \triangleright$ Eq. (13)
7: $\quad$ **for** $k = 1$ **to** $K$ **do**
8: $\quad\quad \mathbf{v}_\theta(\mathbf{z}_t, \mathbf{y}, t) \leftarrow \mathbf{v}_\theta(\mathbf{z}_t, t) - \frac{t}{1-t}\, \nabla_{\mathbf{z}_t} \log p(\mathbf{y} \mid \mathbf{z}_t)$ $\hfill \triangleright$ $K$-step update of vector field Eq. (9)
9: $\quad$ **end for**
10: $\quad \mathbf{z}_{t-\Delta t} \leftarrow \text{ODESolverStep}\left(\mathbf{z}_t, \mathbf{v}_\theta(\mathbf{z}_t, \mathbf{y}, t)\right)$
11: **end for**
12: **Return:** $\mathbf{x}_0 \leftarrow \mathcal{D}_\varphi(\mathbf{z}_0)$

---

## C.1  LFlow

- For the solver parameters, we set the absolute and relative tolerances (`atol` and `rtol`) to $10^{-3}$ for inpainting and motion deblurring tasks, and to $10^{-5}$ for Gaussian deblurring and super-resolution tasks.
- We set the hyperparameters to $K = 2$ and $t_s = 0.8$ for all tasks. These values were selected via ablation on validation performance and were found to balance guidance strength and reconstruction fidelity across tasks.
- To ensure dimensional compatibility, the measurement $\mathbf{y}$ is upsampled (e.g., via bicubic interpolation) to match the input size of the latent encoder. For consistency, we adopt the same super-resolution and deblurring operators as in [29] across both our method and the relevant baselines.
- For inpainting tasks, we incorporate the strategy proposed in the PSLD method [33] into our LFlow algorithm. This strategy reconstructs missing regions that align seamlessly with the known parts of the image, expressed as $\mathbf{x}_0 = \mathcal{A}^T \mathcal{A}\mathbf{x}_0 + (\boldsymbol{I} - \mathcal{A}^T \mathcal{A})\mathcal{D}_\varphi(\mathbf{z}_0)$. Unlike the DPS sampler, which generates the entire image and may lead to inconsistencies with the observed data, this approach ensures that observations are directly applied to the corresponding parts of the generated image, leaving unmasked areas unchanged [80]. For other tasks, such as motion deblurring, Gaussian deblurring, and super-resolution, this extra step is unnecessary since no box inpainting is involved, i.e., $\mathbf{x_0} = \mathcal{D}_\varphi(\mathbf{z_0})$.

## C.2  Comparison methods

**PSLD [33]** applies an orthogonal projection onto the subspace of $\mathcal{A}$ between decoding and encoding to enforce fidelity:

$$\mathbf{z}_{t-1} = \text{DDIM}(\mathbf{z}_t) - \rho \nabla_{\mathbf{z}_t}\Big( \|\mathbf{y} - \mathcal{A}\mathcal{D}_\varphi\left(\mathbb{E}[\mathbf{z}_0 \mid \mathbf{z}_t]\right)\|_2^2 + \gamma \|\mathbb{E}[\mathbf{z}_0 \mid \mathbf{z}_t] - \mathcal{E}_\phi\left(\mathcal{D}_\varphi\left(\mathbb{E}[\mathbf{z}_0 \mid \mathbf{z}_t]\right)\right)\|_2^2$$

$$- \mathcal{E}_\phi\left(\mathcal{A}^\top \mathbf{y} + \left(\boldsymbol{I} - \mathcal{A}^\top \mathcal{A}\right)\mathcal{D}_\varphi\left(\mathbb{E}[\mathbf{z}_0 \mid \mathbf{z}_t]\right)\right) \Big). \tag{34}$$

We use a fixed step size of $\rho$ and select $\gamma$ as recommended in [36]. For our experiments, we rely on the official PSLD implementation [2] with its default configurations. Specifically, we conduct ImageNet experiments using Stable Diffusion v1.5, which is generally considered more robust compared to the LDM-VQ4 models.

**PSLD** aims to ensure that latent variables remain close to the natural manifold by enforcing them to be fixed points after autoencoding. While this approach seems to be theoretically justified, it has proven empirically ineffective [34].

---

[2] https://github.com/LituRout/PSLD

**Resample [36]** estimates first a clean latent prediction $\mathbf{z}_0^{\text{est}}(\mathbf{z}_{t+1})$ from the previous sample $\mathbf{z}_{t+1}$ using Tweedie's formula, as described in Eq. (22). This prediction is then used to update the latent state via DDIM:

$$\mathbf{z}_t' = \text{DDIM}\left(\mathbf{z}_0^{\text{est}}\left(\mathbf{z}_{t+1}\right), \mathbf{z}_{t+1}\right). \tag{35}$$

The updated sample $\mathbf{z}_t'$ is then projected back to a measurement-consistent latent variable $\mathbf{z}_t^{\text{proj}}$ via:

$$\mathcal{N}\left(\mathbf{z}_t^{\text{proj}}; \frac{\sigma_t^2 \sqrt{\bar{\alpha}_t}\mathbf{z}_0^{\text{cond}} + (1 - \bar{\alpha}_t)\mathbf{z}_t'}{\sigma_t^2 + (1 - \bar{\alpha}_t)}, \frac{\sigma_t^2(1 - \bar{\alpha}_t)}{\sigma_t^2 + (1 - \bar{\alpha}_t)}\boldsymbol{I}_k\right), \tag{36}$$

where $\mathbf{z}_0^{\text{cond}}$ is a latent vector that satisfies the measurement constraint, obtained by solving the following optimization problem:

$$\mathbf{z}_0^{\text{cond}} \in \arg\min_{\mathbf{z}} \frac{1}{2} \|\mathbf{y} - \mathcal{A}\left(\mathcal{D}_{\boldsymbol{\varphi}}(\mathbf{z})\right)\|_2^2, \quad \text{initialized at } \mathbf{z}_0^{\text{est}}\left(\mathbf{z}_{t+1}\right). \tag{37}$$

Here, $\sigma_t^2$ is a tunable hyperparameter controlling the trade-off between the prior and the data fidelity, and $\bar{\alpha}_t$ is a predefined DDIM noise schedule parameter. For our experiments, we adopt the publicly available implementation provided by the authors [3], using the pre-trained LDM-VQ4 models on FFHQ and ImageNet [59], along with their default hyperparameters and a 500-step DDIM sampler.

**Resample** refines latent diffusion sampling by balancing the reverse-time prior from the unconditional model with a measurement-informed likelihood centered on a consistent latent—ensuring the sample aligns with both the data manifold and observed measurements.

**MPGD [38]** accelerates inference by computing gradients only with respect to the clean latent estimate instead of the noisy input, thus avoiding heavy chain-rule expansions. Their gradient update in latent space is as follows:

$$\mathbf{z}_{t-1} = \text{DDIM}(\mathbf{z}_t) - \eta \nabla_{\mathbb{E}[\mathbf{z}_0|\mathbf{z}_t]} \|\mathbf{y} - \mathcal{A}\left(\mathcal{D}_{\boldsymbol{\varphi}}(\mathbb{E}[\mathbf{z}_0|\mathbf{z}_t])\right)\|_2. \tag{38}$$

Note that in MPGD, we leveraged Stable Diffusion v1.5 as for PSLD. For more information, please refer to the GitHub repository [4].

**DAPS [37]** [5] refines latent estimates via Langevin dynamics guided by a latent prior and a measurement likelihood. The initial estimate $\mathbf{z}_0^{(0)}$ is obtained by solving a probability flow ODE from $\mathbf{z}_t$ using the latent score model. At each inner iteration $j = 0, \ldots, N-1$, the latent estimate $\mathbf{z}_0^{(j)} \in \mathbb{R}^d$ is updated as follows:

$$\mathbf{z}_0^{(j+1)} = \mathbf{z}_0^{(j)} + \eta_t \left(\nabla_{\mathbf{z}_0^{(j)}} \log p(\mathbf{z}_0^{(j)} \mid \mathbf{z}_t) + \nabla_{\mathbf{z}_0^{(j)}} \log p(\mathbf{y} \mid \mathbf{z}_0^{(j)})\right) + \sqrt{2\eta_t}\,\boldsymbol{\epsilon}_j, \quad \boldsymbol{\epsilon}_j \sim \mathcal{N}(\mathbf{0}, \boldsymbol{I}). \tag{39}$$

Here, $\eta_t$ denotes the Langevin step size. The first term reflects prior guidance via the latent score model, while the second enforces consistency with the measurement through decoding and evaluating the likelihood.

**DAPS** avoids the limitations of local Markovian updates in diffusion models by decoupling time steps and directly sampling each noisy state $\mathbf{z}_t$ from the marginal posterior $p(\mathbf{z}_t \mid \mathbf{y})$. It performs posterior sampling by alternating between posterior-guided denoising via MCMC and noise re-injection, enabling large global corrections and improved inference in nonlinear inverse problems.

**SITCOM (Step-wise Triple-Consistent Sampling) [60]** enforces three complementary consistency conditions—measurement, forward diffusion, and step-wise backward diffusion—allowing diffusion trajectories to remain measurement-consistent with fewer reverse steps. By optimizing the input of a pre-trained diffusion model at each step, SITCOM ensures triple consistency across the data manifold, measurement space, and diffusion process, leading to efficient inverse problem solving.

At each step $t$, SITCOM enforces three consistencies:

---

[3] https://github.com/soominkwon/resample
[4] https://github.com/KellyYutongHe/mpgd_pytorch/
[5] https://github.com/zhangbingliang2019/DAPS

**(S1) Measurement-consistent optimization:**

$$\hat{\mathbf{v}}_t := \arg\min_{\mathbf{v}'_t} \underbrace{\left\| \mathcal{A} \left( \tfrac{1}{\sqrt{\bar{\alpha}_t}} \overbrace{\left[\mathbf{v}'_t - \sqrt{1-\bar{\alpha}_t}\,\epsilon_\theta(\mathbf{v}'_t, t)\right]}^{\mathbf{C}_2} \right) - \mathbf{y} \right\|_2^2}_{\mathbf{C}_1} + \lambda \underbrace{\|\mathbf{z}_t - \mathbf{v}'_t\|_2^2}_{\mathbf{C}_3}. \tag{40}$$

**(S2) Denoising:**

$$\hat{\mathbf{z}}'_0 = \tfrac{1}{\sqrt{\bar{\alpha}_t}}\left[\hat{\mathbf{v}}_t - \sqrt{1-\bar{\alpha}_t}\,\epsilon_\theta(\hat{\mathbf{v}}_t, t)\right], \tag{41}$$

**(S3) Sampling (Forward step):**

$$\mathbf{z}_{t-1} = \sqrt{\bar{\alpha}_{t-1}}\,\hat{\mathbf{z}}'_0 + \sqrt{1-\bar{\alpha}_{t-1}}\,\epsilon. \tag{42}$$

Together, these steps enforce **C1:** measurement, **C2:** backward trajectory, and **C3:** forward diffusion consistency. For implementation details and hyperparameters, we rely on the official GitHub repository [6].

**SITCOM** nudges the input to the denoiser at each diffusion step so its denoised output matches the measurements while staying close to the current state. It then computes the clean estimate (Tweedie) and re-noises via the forward kernel to keep the next input in distribution.

**DMPlug [51]** [7] views the entire reverse diffusion process $R(\cdot)$ as a deterministic function mapping seeds to objects, and solves the inverse problem by optimizing directly over the seed $\mathbf{z}$:

$$\mathbf{z}^* \in \arg\min_{\mathbf{z}} \ \ell\big(\mathbf{y}, \mathcal{A}(R(\mathbf{z}))\big) + \Omega(R(\mathbf{z})), \quad \mathbf{x}^* = R(\mathbf{z}^*). \tag{43}$$

Most existing DM-based methods for inverse problems interleave reverse diffusion with measurement projections in a *step-wise* manner, but this often breaks both *manifold feasibility* (staying on the data manifold $\mathcal{M}$) and *measurement feasibility* (satisfying $\{\mathbf{x} \mid \mathbf{y} = \mathcal{A}(\mathbf{x})\}$). In contrast, DMPlug is not step-wise: it preserves manifold feasibility by keeping the pretrained reverse process **intact** while promoting $\mathbf{y} \approx \mathcal{A}(\mathbf{x})$ via global optimization.

**ΠGDM [22]** considers the following gradient update scheme

$$\mathbf{x}_{t-1} = \mathrm{DDIM}(\mathbf{x}_t) - \eta \left( (\mathbf{y} - \mathcal{A}(\mathbb{E}\left[\mathbf{x}_0 \mid \mathbf{x}_t\right]))^\top \left(r_t^2 \mathcal{A}\mathcal{A}^\top + \sigma_{\mathbf{y}}^2 \boldsymbol{I}\right)^{-1} \mathcal{A}\, \frac{\partial \mathbb{E}\left[\mathbf{x}_0 \mid \mathbf{x}_t\right]}{\partial \mathbf{x}_t} \right)^\top. \tag{44}$$

where $\eta$ controls the step size, $\sigma_{\mathbf{y}}$ represents the noise level of the measurement, and $r_t$ is the time-dependent scale for identity posterior covariance. For this method, we utilize the official, reliable code provided by [29].

**OT-ODE [45]** extends the gradient guidance of ΠGDM [22] to ODE sampling via an optimal transport path, resulting in a variance for the identity covariance as $r^2(t) = \frac{\sigma(t)^2}{\alpha(t)^2 + \sigma(t)^2}\,\boldsymbol{I}$. Moreover, the conditional expectation $\mathbb{E}[\mathbf{x}_0 \mid \mathbf{x}_t]$ is computed from the velocity field $\mathbf{v}_\theta(\mathbf{x}_t, t)$, according to the relation in A.3. For a fair comparison, we used the same solver as LFlow, i.e., the adaptive Heun.

---

[6]`https://github.com/sjames40/SITCOM`
[7]`https://github.com/sun-umn/DMPlug`

# D   Additional Experiments and Ablations

## D.1   CelebA-HQ ($256 \times 256 \times 3$).

Table 5: Quantitative results of linear inverse problem solving on **CelebA-HQ** samples of the validation dataset. **Bold** and underline indicate the best and second-best respectively. The method shaded in gray is in pixel space.

| Method | Deblurring (Gauss) | | | Deblurring (Motion) | | | SR ($\times 4$) | | | Inpainting (Box) | | |
|---|---|---|---|---|---|---|---|---|---|---|---|---|
| | PSNR↑ | SSIM↑ | LPIPS↓ | PSNR↑ | SSIM↑ | LPIPS↓ | PSNR↑ | SSIM↑ | LPIPS↓ | PSNR↑ | SSIM↑ | LPIPS↓ |
| LFlow (**ours**) | 29.06 | 0.825 | **0.164** | 30.14 | **0.845** | **0.167** | 28.92 | **0.830** | **0.170** | 24.82 | **0.876** | **0.123** |
| SITCOM [60] | 28.74 | 0.792 | 0.275 | 28.05 | 0.776 | 0.324 | 28.45 | 0.812 | 0.208 | 21.45 | 0.733 | 0.216 |
| DAPS [37] | 26.66 | 0.773 | 0.314 | 27.22 | 0.766 | 0.251 | 28.29 | 0.798 | 0.227 | 21.15 | 0.807 | 0.202 |
| Resample [36] | 28.07 | 0.742 | 0.239 | 28.37 | 0.804 | 0.232 | 29.84 | 0.806 | 0.193 | 19.49 | 0.797 | 0.237 |
| PSLD [33] | 29.47 | **0.833** | 0.310 | 29.75 | 0.821 | 0.313 | **31.65** | 0.829 | 0.246 | 24.03 | 0.812 | 0.165 |
| MPGD [38] | **29.85** | 0.821 | 0.302 | 29.09 | 0.792 | 0.348 | 29.01 | 0.760 | 0.280 | 23.80 | 0.773 | 0.198 |
| OT-ODE [45] | 27.83 | 0.789 | 0.292 | 26.15 | 0.758 | 0.326 | 28.95 | 0.784 | 0.251 | 22.37 | 0.790 | 0.225 |
| C-ΠGFM [48] | 28.26 | 0.801 | 0.280 | 27.18 | 0.743 | 0.335 | 29.52 | 0.805 | 0.226 | 22.84 | 0.798 | 0.219 |

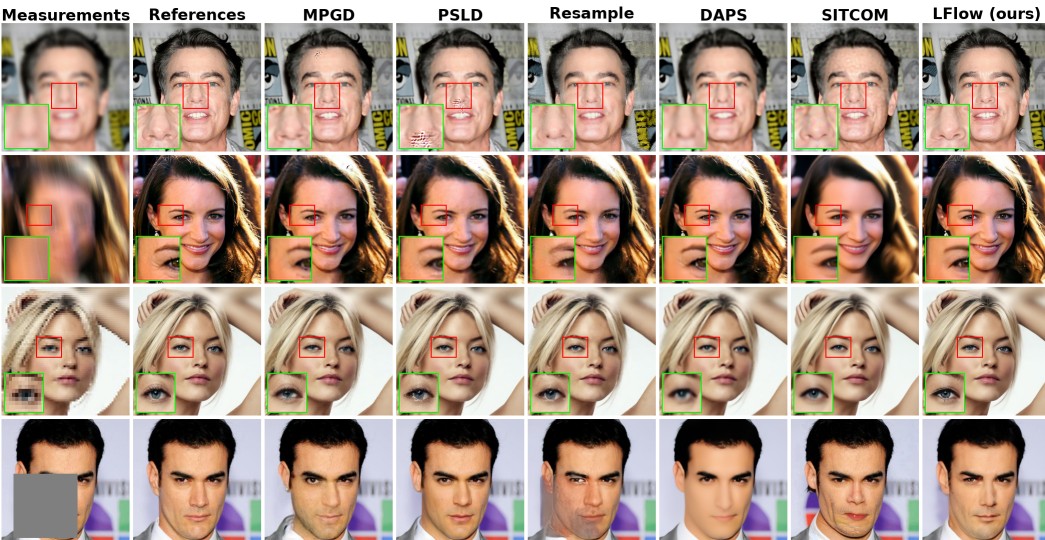

Figure 7: Qualitative results on **CelebA-HQ** test set. Row 1: Deblur (gauss), Row 2: Deblur (motion), Row 3: SR$\times 4$, Row 4: Inpainting.

**CelebA-HQ**   LFlow attains the lowest LPIPS across all four tasks and the highest SSIM in three of four tasks. It reports LPIPS values of 0.164 (Gaussian deblurring), 0.167 (motion deblurring), 0.170 (super-resolution), and 0.123 (inpainting), surpassing the second-best method by 0.042 in inpainting. PSNR remains competitive throughout, ranking first in two tasks and remaining close elsewhere. These quantitative gains are reflected in the visual results: for both deblurring tasks, LFlow restores sharper eye contours, facial edges, and skin textures while avoiding ringing or oversharpening seen in MPGD and PSLD. In super-resolution, it preserves fine details such as eyelashes and lips with smooth transitions, maintaining natural gradients without introducing artifacts. In the inpainting task, LFlow offers semantically consistent completions with coherent tone and geometry—whereas other methods exhibit mismatched shading, seams, or patchy textures. These results highlight LFlow's ability to recover fine structures while maintaining perceptual realism across diverse facial reconstructions.

## D.2 CelebA-HQ (512 × 512 × 3).

We further tested and evaluated our method on the high-resolution CelebA-HQ dataset (512 × 512 × 3), demonstrating its robust capabilities in handling complex image processing tasks. In Figure 8, we showcase the effectiveness and versatility of our approach in enhancing image quality across various tasks.

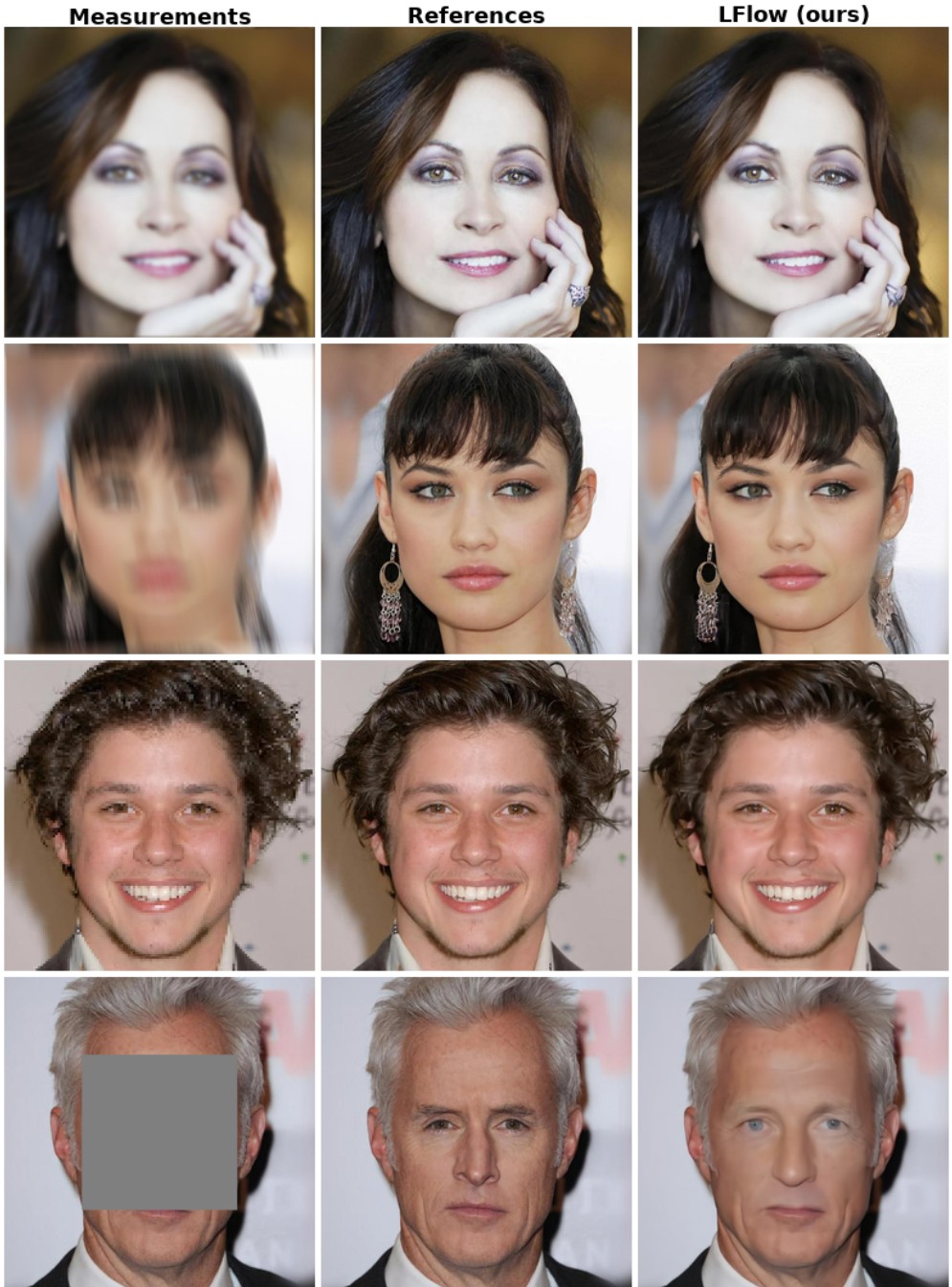

Figure 8: **Additional results on CelebA-HQ 512 × 512 dataset. Row 1: Deblur (gaussian), Row 2: Deblur (motion), Row 3: SR×4, Row 4: Inpainting.**

### D.3 FID Score Results

Table 6: FID ↓ scores across four inverse problems on **FFHQ**, **CelebA-HQ**, and **ImageNet**.

| Method | Deblurring (Gaussian) | | | Deblurring (Motion) | | | SR ($\times 4$) | | | Inpainting (Box) | | |
|---|---|---|---|---|---|---|---|---|---|---|---|---|
| | FFHQ | CelebA | ImageNet | FFHQ | CelebA | ImageNet | FFHQ | CelebA | ImageNet | FFHQ | CelebA | ImageNet |
| LFlow (ours) | **52.48** | **47.79** | 88.76 | 57.11 | **48.53** | **82.89** | **58.49** | 51.07 | 92.28 | **34.40** | **30.78** | **117.45** |
| SITCOM [60] | 74.80 | 76.72 | 72.95 | 70.18 | 70.26 | 68.64 | 67.30 | 62.58 | 90.55 | 45.25 | 50.05 | 123.62 |
| DAPS [37] | 72.45 | 65.58 | 75.51 | 76.23 | 67.23 | 89.17 | 65.78 | 48.47 | 83.42 | 51.28 | 45.59 | 126.36 |
| DMplug [51] | 78.50 | —— | —— | 75.35 | —— | —— | 80.86 | —— | —— | 60.36 | —— | —— |
| Resample [36] | 59.64 | 52.47 | **63.35** | 69.74 | 63.12 | 85.90 | 78.62 | 59.47 | 105.25 | 55.60 | 68.31 | 138.84 |
| MPGD [38] | 64.20 | 61.37 | 102.58 | 70.32 | 88.7 | 146.58 | 90.55 | 84.43 | 119.12 | 84.53 | 53.15 | 154.28 |
| PSLD [33] | 62.49 | 57.92 | 87.39 | 68.94 | 75.21 | 124.73 | 66.22 | 70.66 | **80.58** | 43.89 | 40.18 | 119.12 |
| OT-ODE [45] | 56.72 | 62.23 | —— | 53.55 | 58.12 | —— | 60.71 | 47.83 | —— | 40.31 | 37.92 | —— |
| C-ΠGFM [48] | —— | 56.85 | —— | —— | 50.54 | —— | —— | **45.10** | —— | —— | 34.96 | —— |

**PSNR** and **SSIM** are commonly used recovery metrics that quantify pixel-level fidelity, while **LPIPS** and **FID** are considered perceptual metrics that assess high-level semantic similarity or perceptual quality. In this paper, we focus on image reconstruction tasks from noisy measurements. In such settings, perceptual metrics like FID—although effective for evaluating generative models that prioritize visual realism—primarily measure distribution-level similarity and may overlook structural details, especially when fine-grained information is critical. Moreover, FID can be misleading when reconstructed images appear perceptually plausible but deviate significantly from the ground truth [49]. In contrast, PSNR and SSIM offer objective evaluations of noise suppression and content preservation, which are crucial in our experiments. That said, we report FID scores across all three datasets considered above for four different tasks. As shown in Table 6, our algorithm achieves a balanced trade-off between recovery and perceptual metrics. In the task of noisy image reconstruction, it not only delivers the best recovery metrics but also achieves strong perceptual scores.

### D.4 Ablation on hyperparameter K

To evaluate the effect of hyperparameter K, we conducted an ablation study on the FFHQ dataset. As shown in Table 7, setting K=2 yields slightly better average performance across tasks compared to K=1. Although we also tested K $\geq$ 3, the results showed negligible improvements while incurring higher computational costs. For this reason, we adopt K=2 as it provides consistent gains while maintaining reasonable inference time.

Table 7: Ablation of parameter $K$ on FFHQ. Increasing $K$ offers marginal gains with higher cost; $K = 2$ achieves the best trade-off between performance and efficiency.

| $K$ | GDB | | | MDB | | | SR | | | BIP | | |
|---|---|---|---|---|---|---|---|---|---|---|---|---|
| | PSNR↑ | SSIM↑ | LPIPS↓ | PSNR↑ | SSIM↑ | LPIPS↓ | PSNR↑ | SSIM↑ | LPIPS↓ | PSNR↑ | SSIM↑ | LPIPS↓ |
| 1 | 28.72 | 0.829 | 0.172 | 29.71 | 0.842 | 0.173 | 28.80 | 0.834 | 0.183 | 23.59 | 0.859 | 0.136 |
| 2 | **29.10** | **0.837** | **0.166** | **30.04** | **0.849** | **0.168** | **29.12** | **0.841** | **0.176** | **23.85** | **0.867** | **0.132** |

### D.5 Comparing LFlow with supervised methods

We incorporated **supervised** results obtained using a conditional diffusion model. Our method offers several key advantages over **supervised** inverse approaches such as **SR3** [81] and **InvFussion** [82]:

- **LFlow** is a zero-shot method that generalizes across diverse tasks without retraining, whereas **supervised** methods require training a separate model for each specific task.

- **LFlow** is robust to varying degradation types, while **supervised** methods often exhibit poor generalization when faced with distribution shifts.

- **LFlow** achieves significantly better performance on certain datasets and resolutions— for example, FFHQ at $256 \times 256$.

These advantages are clearly demonstrated in the results presented in Table 8.

Table 8: **Comparison of LFlow with supervised baselines** on two tasks: super-resolution and box inpainting. For *super-resolution*, we compare against **SR3**, trained on the *ImageNet* dataset. For *box inpainting*, we compare against **InvFussion**, trained on the FFHQ dataset.

| Method | SR | | | Inpainting (Box) | | |
|---|---|---|---|---|---|---|
| | PSNR↑ | SSIM↑ | LPIPS↓ | PSNR↑ | SSIM↑ | LPIPS↓ |
| SR3 [81] | 24.65 | 0.708 | 0.347 | – | – | – |
| InvFussion [82] | – | – | – | 20.12 | 0.827 | 0.215 |
| LFlow (**ours**) | **25.29** | 0.696 | **0.338** | **23.85** | **0.867** | **0.132** |

## D.6 Pretrained Models and Algorithm Conversion

**Conversion of Pretrained Models.** Conversion from discrete-time diffusion model to continuous-time flow model was first introduced in [45] by aligning the signal-to-noise ratio (SNR) of the two processes. In principle, this enables mapping discrete diffusion steps $\{\tau\}$ to continuous flow times $t$ and rescaling the noise accordingly. However, this mapping is practically valid under restrictive assumptions—namely, that both trajectories follow the same distributional path (e.g., Gaussian) and employ a linear noise schedule. While such conversion is feasible for pretrained models with similar linear schedules, it becomes non-trivial for discrete-time latent diffusion models (e.g., LDMs, Stable Diffusion). Their custom nonlinear schedules break the clean SNR alignment, often requiring the solution of nonlinear or even cubic equations to infer consistent flow times.

**Algorithmic Conversion.** Beyond converting the pretrained model itself, one must also consider the sampling *algorithm*. Several baselines are inherently discrete-time: their update rules depend on the availability of a finite noise grid and stepwise re-noising kernels. Such designs do not always admit a continuous-time analogue, and therefore direct conversion is not universally possible. Nevertheless, we successfully extended two representative methods, **PSLD** and **MPGD**, to the continuous-time flow setting. By reformulating their projection–correction steps as infinitesimal updates within an ODE sampler, we obtain continuous-time counterparts that closely follow the spirit of the original algorithms while operating with a pretrained flow prior. In contrast, algorithms that fundamentally rely on discrete re-noising (e.g., DAPS) cannot be faithfully mapped without substantial redesign.

Table 9: Quantitative comparison of LFlow results on **FFHQ** dataset against continuous-time versions of CT-MPGD and CT-PSLD across four inverse problems. **Bold** and underline indicate the best and second-best respectively.

| Method | Deblurring (Gaussian) | | | Deblurring (Motion) | | | SR (×4) | | | Inpainting (Box) | | |
|---|---|---|---|---|---|---|---|---|---|---|---|---|
| | PSNR↑ | SSIM↑ | LPIPS↓ | PSNR↑ | SSIM↑ | LPIPS↓ | PSNR↑ | SSIM↑ | LPIPS↓ | PSNR↑ | SSIM↑ | LPIPS↓ |
| LFlow (**ours**) | 29.10 | **0.837** | **0.166** | **30.04** | **0.849** | **0.168** | 29.12 | **0.841** | **0.176** | 23.85 | **0.867** | **0.132** |
| PSLD [33] | **30.28** | 0.836 | 0.281 | 29.21 | 0.812 | 0.303 | 29.07 | 0.834 | 0.270 | 24.21 | 0.847 | 0.169 |
| CT-PSLD | 27.11 | 0.783 | 0.332 | 25.98 | 0.770 | 0.356 | 26.53 | 0.772 | 0.331 | 20.60 | 0.790 | 0.252 |
| MPGD [38] | 29.34 | 0.815 | 0.308 | 27.98 | 0.803 | 0.324 | 27.49 | 0.788 | 0.295 | 20.58 | 0.806 | 0.324 |
| CT-MPGD | 24.96 | 0.765 | 0.343 | 24.30 | 0.751 | 0.375 | 24.69 | 0.740 | 0.338 | 20.01 | 0.780 | 0.353 |

**Discussion.** The results in Table 9 clearly show that *algorithmic conversion* is non-trivial. Although we successfully reformulated PSLD and MPGD into their continuous-time counterparts (CT-PSLD and CT-MPGD), both suffer a significant performance drop compared to their original discrete-time versions. Importantly, we kept the comparison fair by employing the same ODE solver (`adaptive Heun`) as used in LFlow. This indicates that simply replacing the diffusion prior with a flow prior, while retaining the algorithmic structure, does not guarantee competitive performance in continuous time. Instead, the gap highlights the necessity of designing sampling strategies that are intrinsically compatible with ODE-based flow formulations, as achieved in our proposed LFlow framework.

## D.7 Additional Visual Results (Best Viewed When Zoomed in)

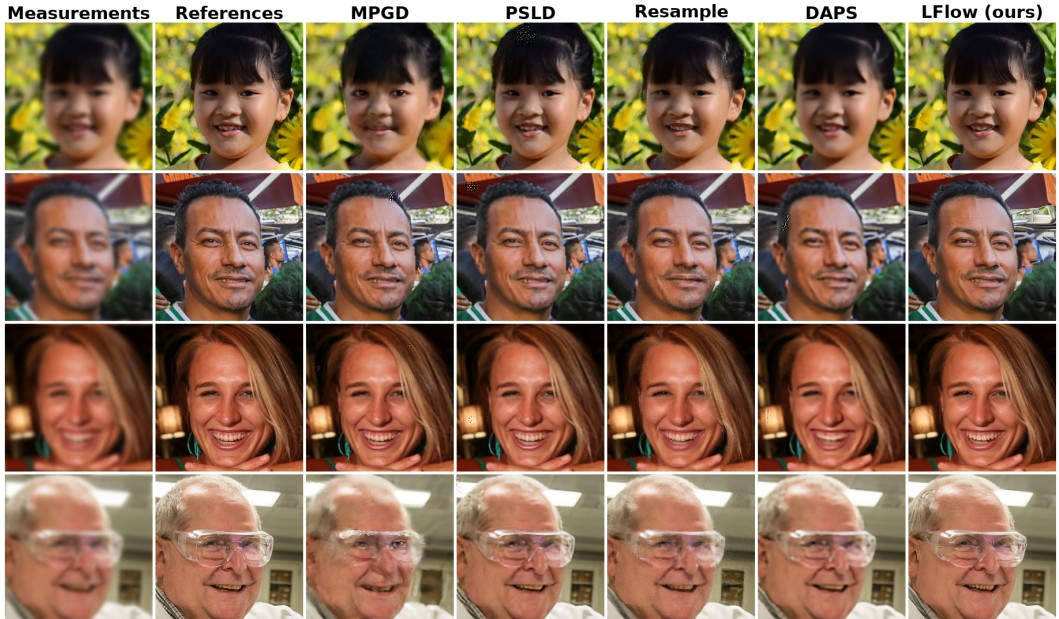

Figure 9: Additional **Gaussian deblurring** results on the **FFHQ** dataset.

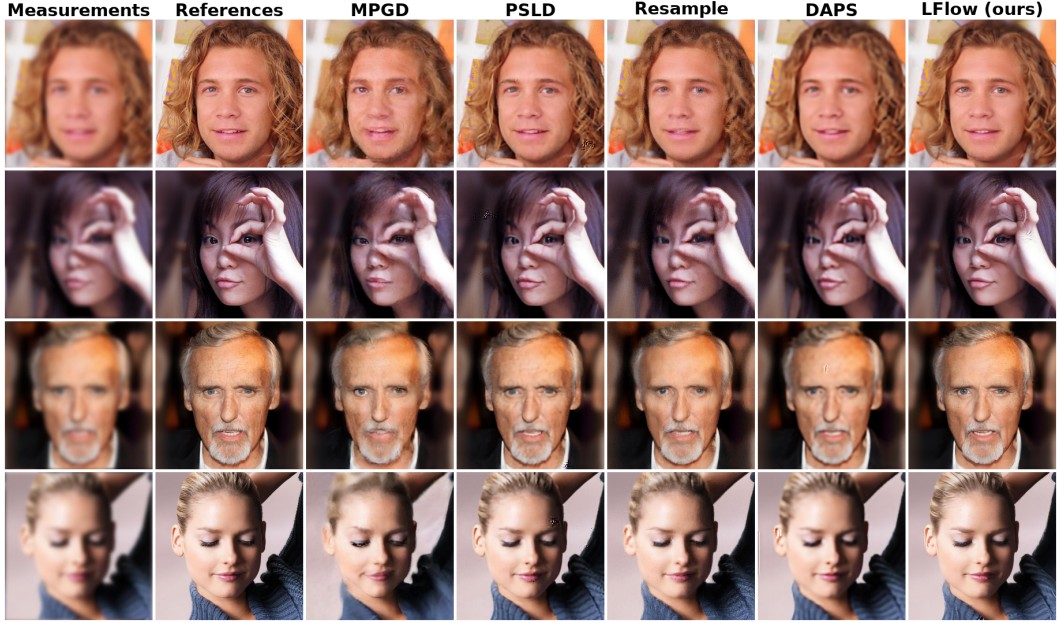

Figure 10: Additional **Gaussian deblurring** results on the **CelebA-HQ** dataset.

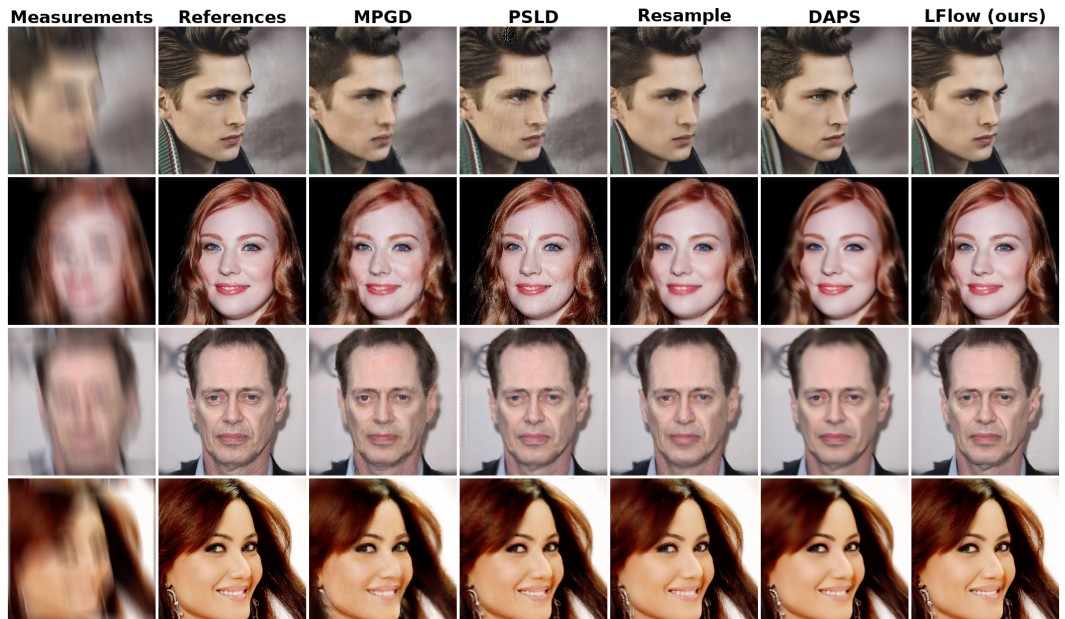

Figure 11: Additional **motion deblurring** results on the **CelebA-HQ** dataset.

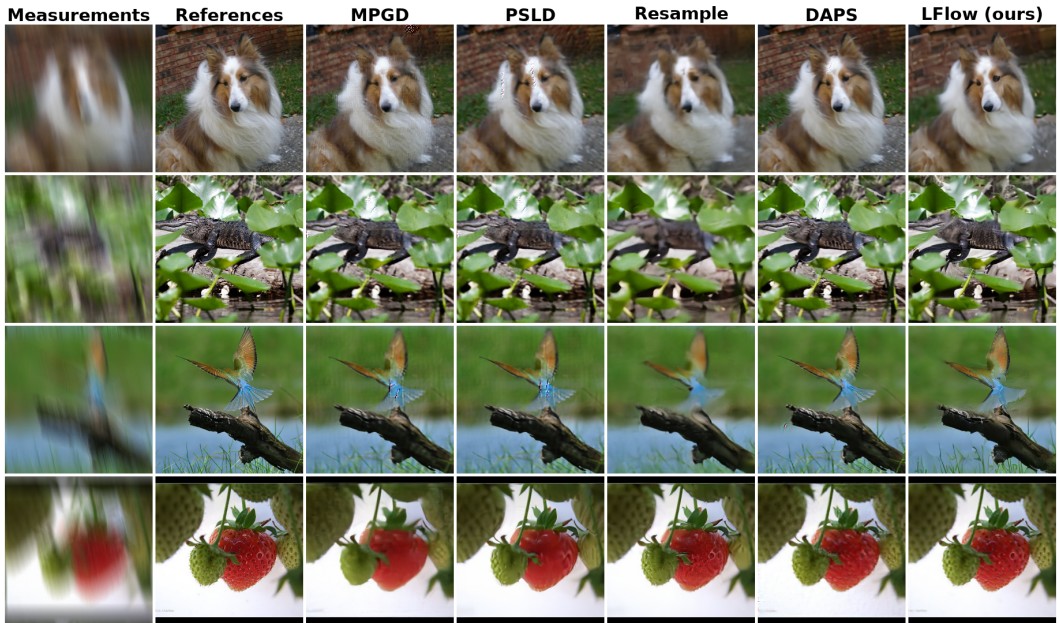

Figure 12: Additional **motion deblurring** results on the **ImageNet** dataset.

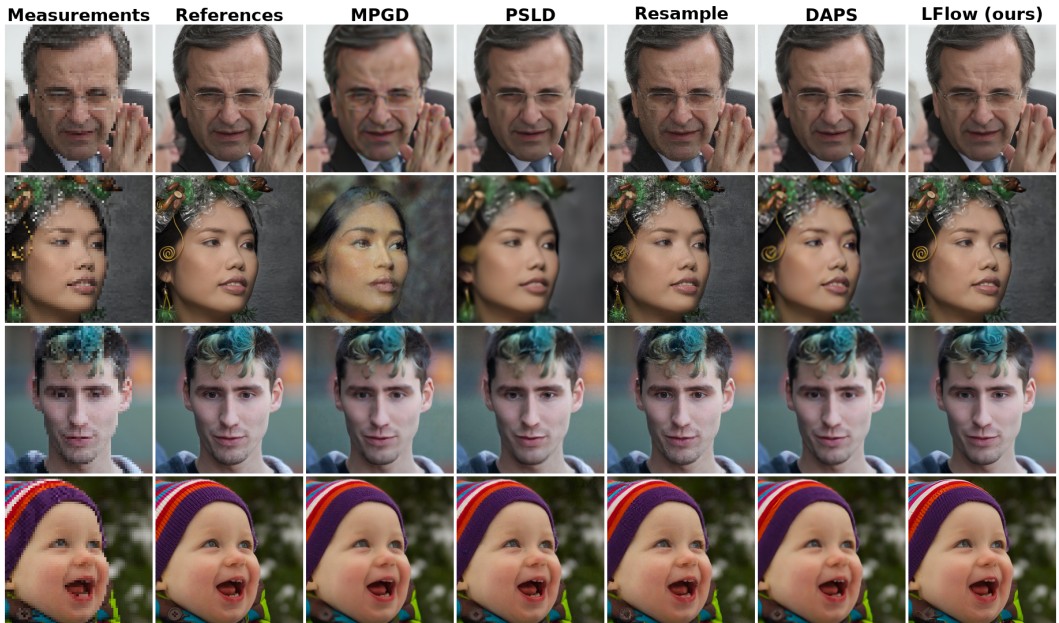

Figure 13: Additional **Super-resolution** results on the **FFHQ** dataset.

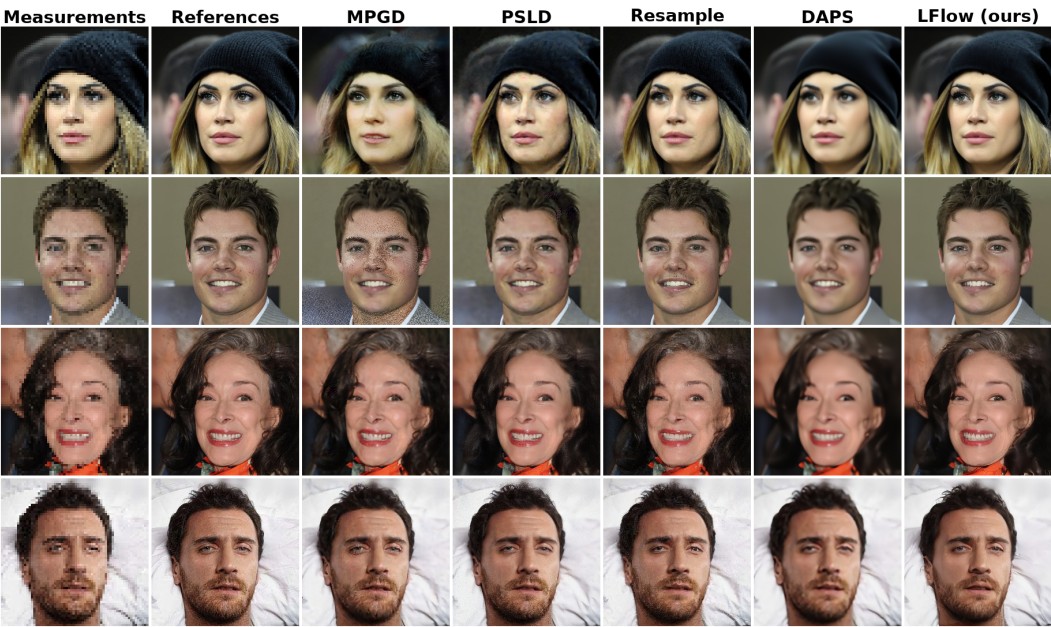

Figure 14: Additional **Super-resolution** results on the **CelebA-HQ** dataset.

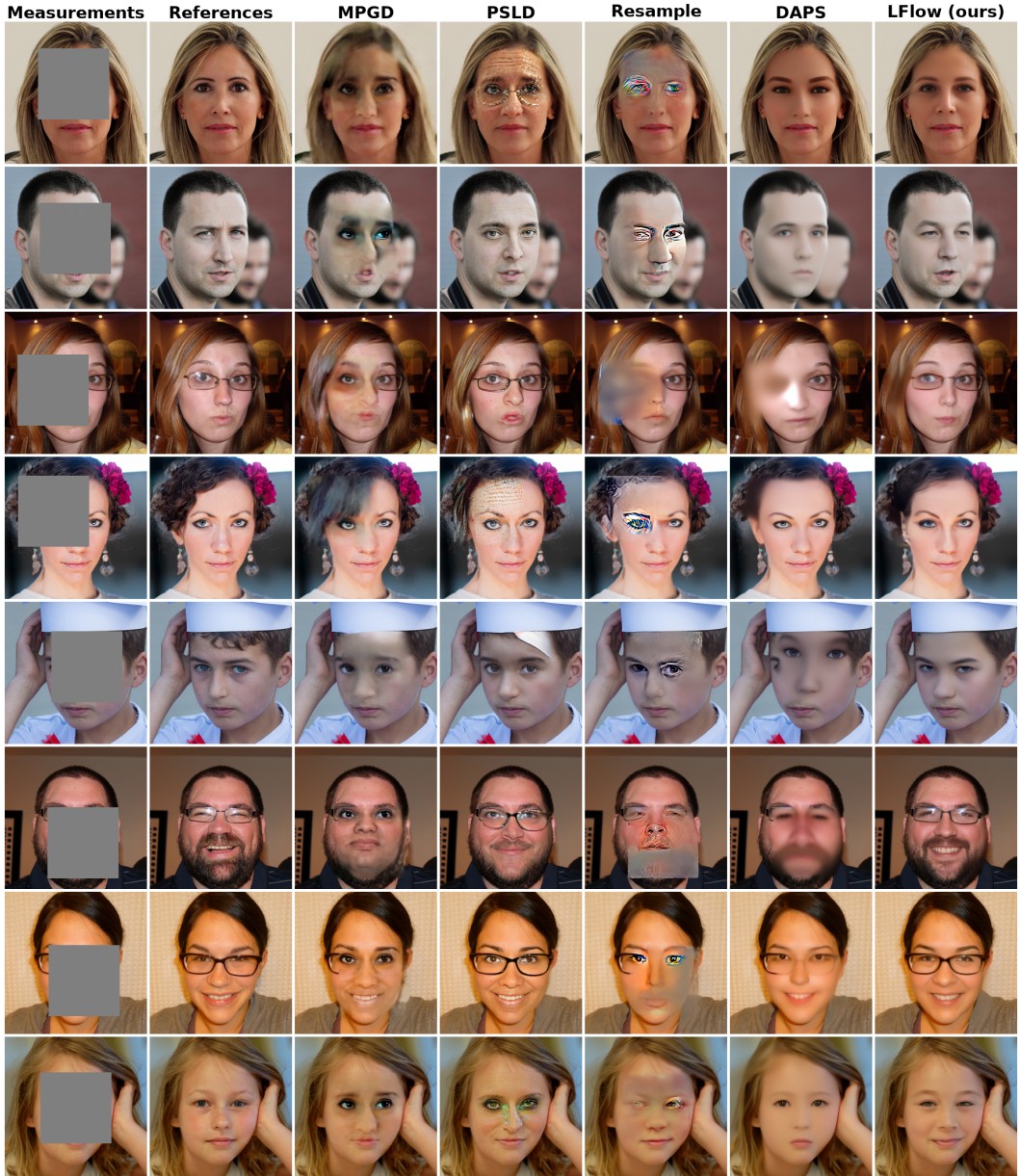

Figure 15: Additional **Inpainting** results on **FFHQ** dataset.

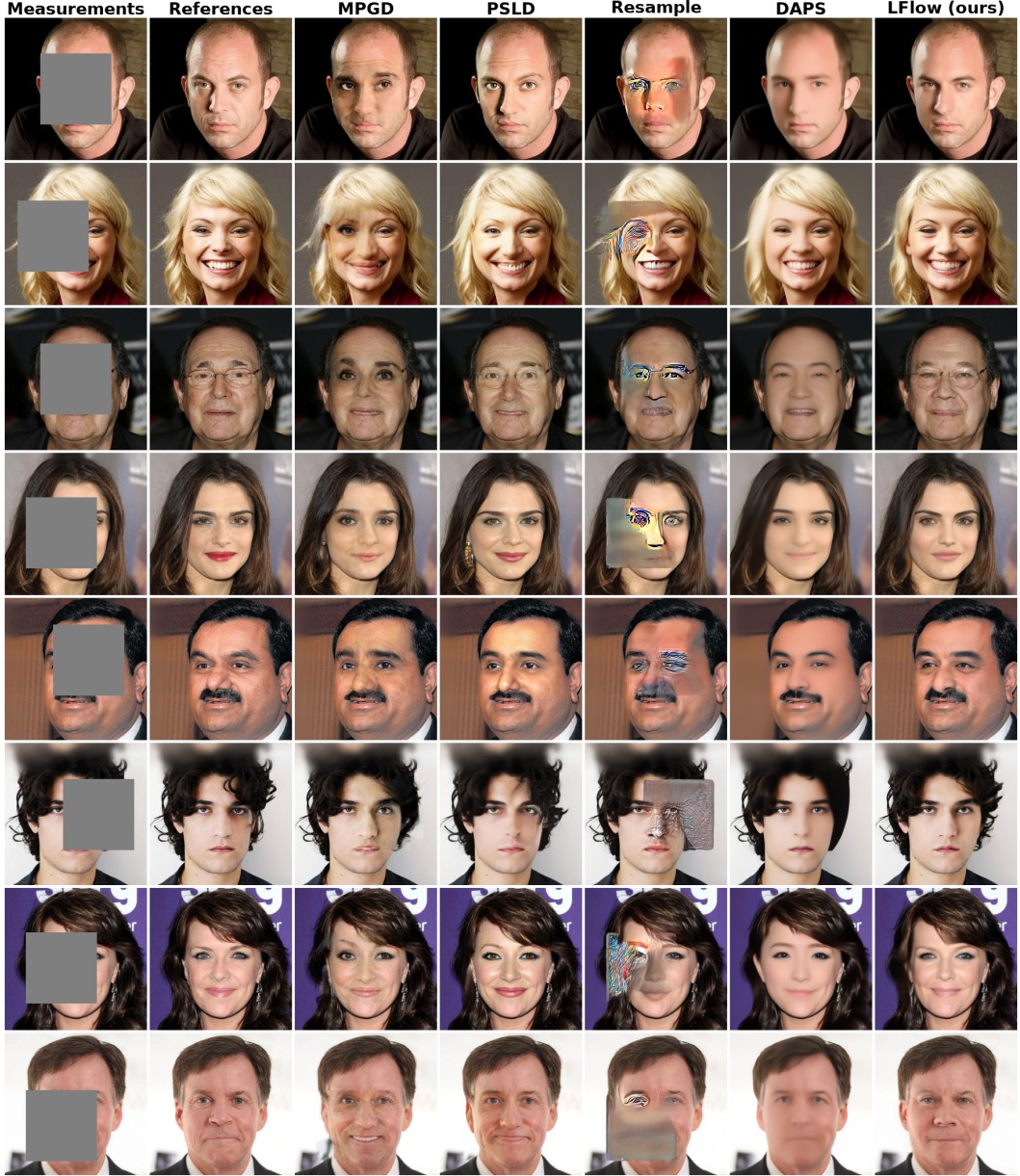

Figure 16: Additional **Inpainting** results on **CelebA-HQ** dataset.

