# OpenReview forum: "Latent Refinement via Flow Matching for Training-free Linear Inverse Problem Solving"
_NeurIPS.cc/2025/Conference — NeurIPS 2025 poster_

### Official Review · Reviewer_H6V9 · 2025-06-28

**Clarity:** 3
**Significance:** 2
**Originality:** 2
**Rating:** 4
**Confidence:** 4

**Summary:**

This paper introduces LFlow, a novel training-free framework for solving linear inverse problems in imaging. The core idea is to leverage a pretrained latent flow-matching model to define a prior distribution. The method operates in a compressed latent space for computational efficiency. The key contributions are twofold: (1) performing ODE-based sampling in the latent space of a pretrained autoencoder, which is more scalable than pixel-space methods , and (2) introducing a theoretically-grounded, time-dependent posterior covariance for the guidance term, which is derived under a Gaussian latent representation assumption to improve sampling accuracy and convergence.

**Questions:**

no

**Ethical Concerns:**

["NO or VERY MINOR ethics concerns only"]

**Final Justification:**

While there are still issues remained, the authors have resolved most of my concerns.

**Quality:**

2

**Strengths And Weaknesses:**

pros:
1. A major practical advantage of LFlow is its computational efficiency. The ablation study on inference time (Table 4) shows that LFlow is substantially faster than its latent diffusion-based counterparts.  This is a crucial benefit, as the computational cost of iterative refinement models is often a significant barrier to their practical application. The paper effectively demonstrates that the proposed posterior covariance model leads to faster convergence.
2. The paper is well-written, clearly structured, and easy to follow. The authors provide a detailed appendix containing full proofs, algorithm specifics, and extensive additional results, which bolsters the credibility of the work and provides a clear path for reproducibility.

cons:
1. The derivation of the optimal time-dependent posterior covariance (Proposition 3.5 and Eq. 17) hinges on the assumption that the latent prior is a standard Gaussian $(z_0 \sim \mathcal{N}(0,\sigma_{latr}^{2}I))$.  While this is a common regularizer for VAEs, the true latent distribution of a powerful autoencoder is unlikely to be perfectly Gaussian. In fact, the VAEs used in diffusion models—such as Stable Diffusion or Flux—are only weakly regularized by a Gaussian prior. The paper does not analyze the sensitivity of the method to deviations from this assumption, which could affect its performance in practice.
    - This also raises concerns about whether the Optimal Vector Field can be effectively applied to pixel-space inverse problem solving.

2. The authors state they use an LFM-VAE framework based on a DiT transformer architecture, while the baseline methods primarily use LDM/Stable Diffusion models built on U-Nets. Although the authors claim their prior's quality does not exceed that of the baselines, this architectural difference is a potential confounding variable. It makes it harder to isolate the performance gains as being solely due to the proposed LFlow methodology versus the underlying generative architecture. Note that for solving inverse problem with pretrained diffusion models, to the best of my knowledge, all the related papers conducted benchmarks on the same backbone model.

3. Baseline comparisons and fairness of evaluation:
The paper would benefit from a more comprehensive comparison with existing flow matching baselines. While the authors mention OT-ODE, they do not include it in the experimental comparison. Although OT-ODE was originally applied to pixel-space models, it can be readily adapted to the latent space—just as with \$\Pi\$GDM. Similarly, other recent flow-matching solvers, such as [C-\$\Pi\$GFM](https://proceedings.neurips.cc/paper_files/paper/2024/file/2f46ef5725a8eca24f7f24a17955ad1a-Paper-Conference.pdf), should also be considered. Moreover, some empirical studies (e.g., Stable Diffusion 3) suggest that the conventional noise schedules used in diffusion models are less efficient than flow matching. This raises a concern about the fairness of comparisons between the proposed method and diffusion-based baselines. To strengthen the claim that "we designed a better sampler," it would be helpful to control for the training objective (e.g., flow matching vs. diffusion) and isolate the benefit from the proposed sampler design itself.

4. Section 3 presents a mixture of existing methods and the proposed contributions, which makes it difficult to clearly identify the novelty of the paper. For example, techniques such as posterior sampling, conditional vector fields, and the initialization of the flow sampling process have already been introduced in prior works. I would recommend restructuring this section to more explicitly distinguish between what is novel in this paper and what is borrowed from previous literature.

---

> ### Author Rebuttal · Authors · 2025-07-27
>
> We sincerely thank the reviewer for their feedback. We address all comments individually below.
> ### **1. What are Assumptions on Latent Prior $p(\mathbf{z}_0)$ ?**
>
> > - **(1). Log-concave Prior.** Our assumption is first grounded in a bound on Jacobian of vector field $\nabla_{\mathbf{z}_t} \mathbf{v} _\theta(\mathbf{z}_t, t)$ (Revised Proposition 3.3, which holds for any strongly log-concave prior, including **Smoothed Laplace**,  **Logistic Distribution**, **Generalized Gaussian with hight $\beta$**,  and **Sub-Gaussian priors with strong tail curvature**, ensuring theoretical robustness even when the prior deviates mildly from exact Gaussianity. We further refer the reviewer to our complementary response to the third question of Reviewer xFYk regarding this point.
>
>
> > - **(2). Gaussian prior.** Using a Gaussian latent prior is standard practice in models such as Stable Diffusion and LDMs, enabling closed-form sampling and efficient inference. In VAEs, the latent codes are explicitly regularized towards a standard normal prior ($\mathcal{N}(0, \mathbf{I})$) during training, so the aggregate posterior $q(\mathbf{z})$ is encouraged to match this prior—which also serves as the base distribution for score-based generative modeling (e.g., Vahdat et al., 2021). Although the true posterior may be complex, this procedure results in an aggregate latent distribution that is approximately Gaussian, as established in prior work. Importantly, if the latent prior were highly non-Gaussian, reliable sampling from the autoencoder (and thus from any latent generative model) would become fundamentally challenging. Since standard practice in VAEs and latent diffusion models is to sample $\mathbf{z}_0 \sim \mathcal{N}(0, \mathbf{I})$ for generation, a significant mismatch would lead to unrealistic or out-of-distribution decodings—impacting all approaches relying on pretrained autoencoders, not just ours. Thus, our assumption is both practical and broadly consistent with the requirements of latent generative modeling. Furthermore, our assumption is significantly less restrictive than the Gaussian assumption made by $\Pi$GDM on the **data** itself (i.e., $p(\mathbf{x}_0) \sim \mathcal{N}(0, \boldsymbol{\Sigma}))$. This is shown in proof A.7.
>
> >- **(3). Remark.** In this remark (please see response **3** to **xFYk**), we show that the Gaussian prior is a tight instance of this bound (The Gaussian prior makes the inequality in the bound turn into an equality), justifying the closed-form posterior covariance in Eq. 17.
>
> > - **Posterior Covariance in pixel space.**
> > Under *Gaussian data assumption*, an analogous posterior covariance can be derived in **pixel space**, making the Optimal Vector Field applicable there as well. However, the latent space formulation is much more natural and theoretically justified: the latent Gaussian prior is better supported (e.g., by VAE training), and the reduced dimensionality enables more stable and tractable posterior estimation.
>
> ---
> ### **2. Architectural Differences**
>
> > We respectfully offer a different perspective and provide two complementary reasons:
>
> > - **First**, the generative process is evaluated based on **image quality**, not the architecture itself. What ultimately matters in evaluating a generative prior is the quality of the samples it produces, as measured by such metrics as LPIPS and FID—not the specific architecture used. Therefore, if the prior in our method does not outperform the baselines in **unconditional generation** (as we explicitly state), any observed improvement in inverse problem performance can be reasonably attributed to the **inference strategy** rather than architectural differences. The comparison remains valid and well-justified.
>
> > - **Second**, solving inverse problems inherently requires **balancing** *prior fidelity* and *measurement consistency*. While strong priors (e.g., Stable Diffusion 1.5) have advantages, they may compromise consistency and overall performance, as shown in baseline experiments. This reflects a general principle: even if a stronger prior is used, it does not necessarily yield better inverse problem results unless it aligns well with the measurement model.
>
> ---
> ### **3. Baselines and Fair Comparisons**
>
> > **Baseline Comparisons:**
> >- All existing flow matching baselines (e.g., OT-ODE, C-$\Pi$GFM) operate in **pixel space**, whereas our method is designed for **latent space**. In this work, we primarily focused on latent-space baselines to enable a fair and meaningful comparison within the same modelling paradigm.  That said, we acknowledge that OT-ODE can, in principle, be extended to the latent space. In fact, we already explored such a variant in our **ablation study**, using the OT-ODE posterior covariance (denoted **Cov-$\Pi$GDM**), as described in the Appendix (p. 25).
>
>
> >-  We have now included additional **pixel-space** comparison on the CelebA-HQ dataset. For completeness, we also provide results for the **C-$\Pi$GFM**. The additional experimental results are shown in the following table.  Corresponding qualitative results are available but cannot be included here due to policy restrictions.
>
> >- Results are now presented in two categories: **latent-based methods** (*MPGD*, *Resample*, *PSLD*, *LatentDAPS* on three datasets, and *LatentDMPlug* on FFHQ) and **pixel-based methods** ($\Pi$GDM, *OT-ODE*, and *C-$\Pi$GFM* on CelebA-HQ).
>
>
> ---
> | Methods        |        | **GDB**     |        |        | **MDB**     |        |        | **SR**     |        |        | **BIP**      |        |
> |---------------|--------|-------------|--------|--------|-------------|--------|--------|-------------|--------|--------|-------------|--------|
> |               | PSNR↑  | SSIM↑       | LPIPS↓ | PSNR↑  | SSIM↑       | LPIPS↓ | PSNR↑  | SSIM↑       | LPIPS↓ | PSNR↑  | SSIM↑       | LPIPS↓ |
> | $\Pi$GDM        | 27.93  | 0.795       | 0.187  | 27.42  | 0.754   |  0.231 | 27.16   | 0.801   | 0.198  | 25.77  | 0.774  | 0.208  |
> | OT-ODE    | 28.62  |0.804 |  0.175 | 27.27 | 0.767  | 0.224 | 27.47 | 0.812 |  0.192 | 26.81 | 0.798 | 0.184 |
> | C-$\Pi$GFM        |  28.85  |  0.813 |  0.172   |  26.55  | 0.775   | 0.219   | 26.85  |  0.808  |  0.204 |  26.79 |  0.789 | 0.186 |
> | LFlow (**ours**)     | **29.06**     |   **0.825**         | **0.164**      | **30.14**      | **0.845**   | **0.167**      | 28.92   | **0.830**   | **0.170**  | 24.82     | **0.876**   | **0.123**   |
>
> **Table 3:** *Quantitative comparison of LFlow against other Pixel-based flow solvers across four inverse problems (BIP: Box InPainting, GDB: Gaussian DeBlurring, MDB: Motion DeBlurring, SR: Super-Resolution) on CelebA-HQ datasets*.
>
>
> ---
>
> > **Fair comparison:**
> >- As noted in our paper, flow matching models offer greater **efficiency** during both training and inference by producing straighter sampling paths under modified noise schedules. This efficiency can be further improved when applied in the latent space. However, this does not necessarily mean that ODE-based sampling yields **higher sample quality** compared to SDE-based sampling. This is precisely why other flow-based methods still compare their performance against **diffusion-based counterparts**.
>
>
>
> ### **4. Restructuring the Method Section**
>
> > Thank you for this helpful suggestion. We have revised the methodology section to improve clarity and better distinguish our novel contributions from prior work. We also updated and relocated the Related Work section to Section 2. We added a conceptual figure showing the flow of our method and its difference with OT-ODE in posterior covariance estimation. Regarding the specific examples you mentioned:
>
>
> > - **Posterior sampling:**  While posterior sampling using $\underline{\textit{unconditional} \text{ vector fields}}$ has been explored in **pixel space** for inverse problems, it has not been *theoretically justified*. In this work, we provide a formal **proposition 3.1** establishing when and why posterior sampling is valid under our setting. Building on this foundation, we further extend the framework to handle **conditional vector fields**, enabling guided inference consistent with measurements. We further improved our methodology section (for example, for posterior covariance estimation, we refer the reviewer to our response to the third question of Reviewer xFYk).
>
>
> > - **Initialization strategy:** We have now moved this explanation to the Implementation section. While we acknowledge being "inspired by" prior works, our initialization strategy is notably distinct. Specifically, **across all tasks**, we encode the measurement directly using the encoder $\mathcal{E}$ to obtain the latent code—without applying the forward operator $\mathcal{A}$ or its pseudoinverse $\mathcal{A}^\dagger$, as commonly done in other latent diffusion or pixel-space flow-based methods (e.g., C-$\Pi$GFM).

---

> > ### Comment · Reviewer_H6V9 · 2025-08-03
> > **Thanks for the rebuttal**
> >
> > I think the authors have addressed some of my concerns, but I still believe the architectural differences remain an issue. When solving an inverse problem, the focus should be on evaluating the **sampler**, which means minimizing the influence of other variables. It is not necessary to use the best-performing or most recent pre-trained diffusion or flow model; rather, it is important to use **the same** pre-trained model across comparisons. For example, two diffusion models even with identical architectures and similar unconditional generation performance could still behave differently in inverse problems if they were trained on different datasets or with different hyper-parameters. This could confound the results attributed to the sampler.

---

> > > ### Author Response · Authors · 2025-08-04
> > > **Architectural Differences**
> > >
> > > We sincerely appreciate the reviewer’s ongoing engagement with our work.
> > >
> > > We understand the concern raised regarding architectural differences. While we agree that using **the same** pre-trained models would provide the most clear isolation of **the sampler**’s effect, we still believe that our experimental setup remains justified for the following reasons:
> > >
> > > > **1.** We emphasize that existing latent diffusion inverse solvers are inherently **discrete-time** methods, which typically rely on **pretrained** Latent Diffusion Models (LDMs) or Stable Diffusion (SD) variants **[1]** during inference. These pretrained models are **trained** on a fixed set of discrete timesteps, e.g., $t \in \{1/N, 2/N, \dots, 1\}$. In contrast, **continuous-time** models—such as flow-based inverse solvers and flow matching frameworks—sample $t$ continuously from $[0, 1]$ during **training**. One option that we had was to train a continuous-time latent diffusion model from scratch on three datasets. We experimented with this on one FFHQ dataset by extending the continuous diffusion architecture introduced in **[2]** to the latent space, as well as adapting the architecture from EDM2 **[3]**. However, this required **modifications** to the baseline inference samplers, which ultimately led to performance that was even worse than the results reported by the baselines. Also, the architectures were not matched yet to those of our pretrained flow models.
> > >
> > > > **2.** Unlike latent-based diffusion inverse solvers that use **LDM-VQ4** or **Stable Diffusion (SD)** variants—typically trained with VQ regularization—our method introduces ODE sampling via pre-trained latent vector fields for inverse problems, a setting not addressed by previous methods. Our pre-trained flow model is distinct in its training objective, noise schedulers (linear versus non-linear diffusion schedulers), and hyperparameters, making direct comparison using identical models infeasible. **Thus**, even if the architectures were identical, differences in training objectives and other key parameters would persist. This is true not only for our method, but also for other flow‑based inverse solvers in pixel spaces, which compare their method with diffusion counterparts. Consequently, it remains unclear what realistic baseline alternatives could exist **for latent vector‑field‑based posterior sampling**, given the inherent incompatibilities in training objectives, and regularizations between flow‑based and diffusion‑based approaches.
> > >
> > > > **3.** Our LFlow method is theoretically formulated for latent flow models whose autoencoders are trained with a VAE objective. This VAE‑based regularization is essential for our latent posterior covariance modeling. In contrast, methods such as Resample and DAPS employ available LDMs with vector‑quantized regularization (i.e., LDM-VQ-4), which cannot be suitable for our method, as they cannot provide the same theoretical justification.
> > >
> > > > **4. Given the limitations above, it is highly reasonable to resort to the fact that** since our model’s unconditional generative quality does not surpass that of the baselines, the method is training‑free, and observed performance gains emerge solely in the inverse problem setting, these gains can be attributed primarily to the LFlow inference strategy rather than to architectural differences.
> > >
> > > > **5.** The literature shows that several leading works compare their methods with baselines that use different architectures, often due to practical limitations. For example,
> > >
> > > > - **DPS**, implemented with the ADM architecture **[4]**, is compared against **DDRM** using the architecture from **[5]**.
> > >
> > > > - **PSLD** (a latent-based method with autoencoders involved) is compared to **DPS** (pixel-based), with each employing a different architecture.
> > >
> > > > - **OT-ODE** is compared to **$\Pi$GDM**, but the methods are evaluated using different architectures: OT-ODE learns vector fields with the ADM architecture **[4]**, while $\Pi$GDM is applied to VP-SDE, where the score function is learned using Score-SDE backbones **[2]**.
> > >
> > >
> > > ---
> > > **[1]. High-resolution image synthesis with latent diffusion models (rombach2022high)**
> > >
> > > **[2]. Score-based generative modeling through stochastic differential equations (song2021scorebased)**
> > >
> > > **[3]. Analyzing and improving the training dynamics of diffusion models (karras2024analyzing)**
> > >
> > > **[4]. Diffusion models beat gans on image synthesis (dhariwal2021diffusion)**
> > >
> > > **[5]. Denoising diffusion probabilistic models (ho2020denoising)**

---

> > > > ### Comment · Reviewer_H6V9 · 2025-08-06
> > > > **thanks for the comment**
> > > >
> > > > I partially agree that achieving an absolutely fair comparison is inherently challenging.
> > > >
> > > > In the case of PSLD, I acknowledge that they adopt a less fair comparison setting. However, for DPS, the authors explicitly state in Appendix D.2 that they use the **same model checkpoint** for all methods evaluated—including DDRM, MCG, Score-SDE, and their own. This provides a more controlled basis for comparison.
> > > >
> > > > OT-ODE presents two settings: (1) they reuse a pretrained VP-SDE and convert the score function to a velocity field for fairer comparison; and (2) they train a generic flow-based model. These setups illustrate both reuse of pretrained models and custom training, depending on the evaluation goal.
> > > >
> > > > C-$\Pi$GDM also consider 2 pretrained models to isolate the influence of training objectives, one for diffusion and one for flow.
> > > >
> > > > Importantly, recent works increasingly adopt the practice of evaluating different sampling strategies using a **shared pretrained model** to isolate the effect of the sampler. A representative example is RED-Diff’s official repository ([https://github.com/NVlabs/RED-diff/tree/master/algos](https://github.com/NVlabs/RED-diff/tree/master/algos)), where multiple sampling algorithms are evaluated using the same underlying model.

---

> ### Author Response · Authors · 2025-08-06
>
> We sincerely appreciate your follow-up comments and clarifications.
>
> Although we still believe that our initial response regarding architecture is both theoretically sound and intuitively convincing, we are more than happy to provide further clarification on the following points.
>
> > **1. On PSLD**
> We are glad that you agree with us regarding PSLD adopting a less fair comparison setting.
>
> > **2. On OT‑ODE**
> As you noted, OT‑ODE considers two settings:
> (1) reusing a pretrained VP‑SDE and converting the score function to a velocity field; and
> (2) training a generic flow‑based model.
>
> > For our scenario, the second setting provides a more relevant comparison, as converting pretrained score functions into velocity fields is impractical for latent-based methods (we can elaborate further if needed). However, could you please clarify whether, in OT-ODE, the “generic flow model” is trained under **the same training objective** as its diffusion-based counterpart? Our understanding is that it is not.
>
> > **3. On your last point regarding shared pretrained models**
> We agree that some recent works evaluate sampling strategies using a shared pretrained model to isolate the effect of the sampler. However, these works—such as RED‑Diff—are, **first**, all in **pixel space and discrete-time**, and **second**, are primarily situated in the **variational perspective** or **variable splitting** of posterior inference.
>
>
> > **Training‑free** inverse problem solvers leveraging diffusion or flow priors span multiple methodological classes:
> > 1. **Variable splitting** methods decompose inference into alternating data‑fidelity and regularization steps~\cite{wang2024dmplug, song2024solving, zhu2023denoising}.
> > 2. **Variational Bayesian** methods introduce a parameterized surrogate posterior, often Gaussian, and optimize it using a variational objective~\cite{Feng_2023_ICCV, feng2023efficient, mardani2024a}.
> > 3. **Asymptotically exact** methods combine generative priors with classical samplers (MCMC, SMC, Gibbs) for provably correct posterior approximation~\cite{wu2023practical, trippe2023diffusion, cardoso2024monte, dou2024diffusion}.
> > 4. **Guidance‑based** methods correct the generative trajectory with an approximate likelihood gradient to steer samples toward the posterior~\cite{jalal2021robust, chung2023diffusion, song2023pseudoinverseguided, boys2024tweedie}.
>
> > Our method explicitly aligns with the guidance-based category. Given the differing assumptions and inference paradigms among these methodological classes, we believe fairness criteria must be contextually assessed.
>
> > **4.** Essentially, comparisons with pixel‑based methods cannot be entirely fair, but this remains common practice among all latent‑based methods.
>
>
> > **5.** Lastly, we wish to explicitly address a crucial point: **If someone intends to utilize a pretrained latent flow model for posterior sampling**, it remains unclear which baselines would constitute fair and meaningful comparisons, and under precisely what conditions. Given your expertise, we would greatly appreciate specific guidance on this matter. We firmly believe **this clarification is essential**—not only to strengthen our current analysis but also to inform future methodological practices. If there are approaches or considerations that we have inadvertently overlooked, we kindly ask you to point them out.

---

> ### Author Response · Authors · 2025-08-08
>
> Dear Reviewer,
>
> We have made continuous efforts, collectively as all authors, to address the issues raised by the reviewer. We sincerely hope that the clarifications and additional experiments we have provided will assist the reviewer in reassessing their evaluation.
>
> We would greatly appreciate it if the reviewer could let us know whether our clarifications have satisfactorily addressed their concerns. If not, we would be happy to know where and how our response may have fallen short, so that we can further clarify.
>
> Best regards,
>
> The Authors

---

> > ### Comment · Reviewer_H6V9 · 2025-08-09
> >
> > Thanks for the reply. Let me make my final comment to simplify my concern: why don't you consider using OT-ODE with the same pretrained latent flow model as a baseline?

---

> ### Author Response · Authors · 2025-08-09
>
> Dear Reviewer, Thank you so much for your engagement.
>
> We did, actually — we really did. Both for the **FFHQ** dataset and the **ImageNet** dataset, we have **complete** results for OT-ODE in latent space. For **at least two tasks** from each dataset, these results are presented in **Table 3**. However, because part of our contribution is **posterior covariance estimation**, we preferred to treat this as an **ablation study** to highlight the significance of our work — not only in terms of metrics (Table 3), but also in terms of **convergence** (Table 4). Otherwise, it would have been no difference for us to place these results in **Table 1** and **Table 2**.
>
>
> **We already have done many experiments with OT-ODE with pre-trained latent flow model and we can present the full results if you want.**

---

> > ### Comment · Reviewer_H6V9 · 2025-08-09
> >
> > I see your point. I suggest making the comparison more explicit, perhaps by adding it directly to the main table. Regarding my earlier concerns, other baselines should be evaluated using the same backbone model as yours (e.g., Resample, DAPS). This could be done relatively easily by converting the flow objective to a score-based formulation. In addition, it would be beneficial to include other metrics such as FID and KID for a more comprehensive evaluation. That said, although some concerns remain, I am willing to improve my score now.

---

> ### Author Response · Authors · 2025-08-09
>
> We are truly grateful for your willingness to reconsider the score and for the constructive suggestions provided.
>
> We agree that making the OT-ODE comparison more explicit in the main table can further clarify our results, and we will incorporate this change now. Regarding the backbones, we will take steps to meet your concern. We also agree that including additional perceptual metrics such as **FID** and **KID** would make the evaluation more comprehensive. We actually already reported **FID** scores for all datasets in **Table 6** and included some discussion on this point. We also consider this score for other new baselines.
>
>
> Once again, we sincerely thank the reviewer for the thoughtful engagement.

---

### Official Review · Reviewer_xFYk · 2025-07-02

**Clarity:** 3
**Significance:** 3
**Originality:** 2
**Rating:** 4
**Confidence:** 4

**Summary:**

This paper presents an inference-time inverse problem solver with pretrained flow models. The proposed algorithm (LFlow) extends posterior sampling methods from diffusion models to flow models, especially latent flow models. Similar to previous works on diffusion posterior sampling, LFlow approximates the noise-conditional likelihood $\nabla_{z_t} \log p(y|z_t)$ using Gaussian approximation on $p(z_0|z_t)$, and derive a close-form expression assuming the linearity of the forward model. The authors further approximate the covariance matrix $Cov(z_0|z_t)$ assuming the prior is Gaussian. Experimental results on image restoration tasks show advantage of  LFlow against several existing baselines.

**Questions:**

I have some more questions regarding the experiments:

- What is the detailed setup in posterior covariance ablation study that results in Table 3 and Figure 4?
- Why is DAPS included in Table 2 but not in Table 4?

**Ethical Concerns:**

["NO or VERY MINOR ethics concerns only"]

**Final Justification:**

The authors have addressed most of my original concerns on the manuscript, and I therefore decide to increase my score to 4.

**Limitations:**

Yes. The authors address the limitation of this paper, e.g., relatively slow generation.

**Quality:**

3

**Strengths And Weaknesses:**

Strength:

- This paper targets an interesting task of solving blind inverse problems with pretrained generative priors, and adapts diffusion posterior sampling methods to flow models.
- The methodology, from estabilishment to practical implementation with approximation, is supported by rigorous theory.
- The experimental results show promising advantages of the proposed method against previous methods.

Weakness:

- Flow models and diffusion models are known to be two sides of a coin, as they can be transferred from one to another by altering parameterization. The posterior sampling of conditional flow is not a result as exciting as it seems in the paper, given that diffusion posterior sampling is well-established.
- Missing conceptual comparison to previous methods. There is no comparison to existing methods in likelihood approximation and posterior covariance approximation in Section 3, which obfuscates existing techniques with the original contribution of this paper. The likelihood approximation looks essentially the same as $\Pi$GDM[1] and TMPD[2], and approximating the covariance matrix (e.g., assuming Gaussian prior) is also discussed in these previous works. It is not clear what is unique in LFlow at least from how it is presented in the current form.

- The experiments are missing several recent baselines, such as DMPlug [3] and SITCOM[4]. The selected pixel-space method  $\Pi$GDM is not the most competitive baseline as well. Also, LFlow is only applicable to linear inverse problems, while most of the compared baselines are applicable to general inverse problems.



[1] Pseudoinverse-Guided Diffusion Models for Inverse Problems. Song et al. ICLR 2023.

[2] Boys et al. Tweedie Moment Projected Diffusions for Inverse Problems

[3] Wang et al. DMPlug: A Plug-in Method for Solving Inverse Problems with Diffusion Models. NeurIPS 2024.

[4] Alkhouri et al. SITCOM: Step-wise Triple-Consistent Diffusion Sampling for Inverse Problems. ICML 2025.

---

> ### Author Rebuttal · Authors · 2025-07-27
>
> We appreciate the reviewer’s comments, which helped us improve the clarity of the paper.
>
> ### **1. Flow vs. Diffusion: Two Sides of the Same Coin?**
>
> > - Diffusion trajectories can be converted into flow paths by matching the signal-to-noise ratio, which allows one to derive the corresponding **time** and **scale** mappings. However, this holds only under certain assumptions—namely, that both processes share the same distributional path (e.g., Gaussian) and the diffusion model is trained in continuous time. For discrete-time models, which operate at fixed steps, this mapping becomes nontrivial. Although heuristic rounding methods exist [Karras 2022], they are often impractical—*particularly* for **latent diffusion models** or **Stable Diffusion** with non-linear noise schedulers—because they require solving a **cubic equation** to infer the flow time, making conversion extremely difficult, if not infeasible in practice.
>
> > - Flow-based models offer significant advantages in both **training** and inference efficiency, thanks to their straighter sampling trajectories. This led the researchers to apply it for inverse problems in *pixel space*, as demonstrated by methods such as OT-ODE [43] and D-Flow [44].
>
> > - **Latent** flow models, such as Stable Diffusion3 (Esser et al., 2024) and OpenSora-2, have shown great success in synthesis but remain underexplored for inverse problems. Our work takes a step in this direction by enabling efficient and accurate posterior sampling in the latent flow framework.
>
> ---
> ### **2. Missing Conceptual Comparison**
>
> > - We have a conceptual figure in **Appendix (page 25)** that illustrates the differences between our method and prior methods such as OT-ODE and $\Pi$GDM (we moved this now to the method section). We have also clarified the distinctions in the revised Related Work section (now Section 2). Additionally, we have now included a conceptual figure that illustrates the overall flow of our proposed method.
>
> ---
> ### **3. Likelihood and Posterior Covariance Estimation**
>
> > We should have included our related work section earlier, before the method section (this has now been revised), which would have made the differences more apparent. We approximate the likelihood as $\nabla _{\mathbf{x}_t} \log \mathbb{E} _{\mathbf{x}_0 \sim p(\mathbf{x}_0 | \mathbf{x}_t)}[p(\mathbf{y} | \mathbf{x}_0)]$, whereas previous methods approximate it as $\nabla _{\mathbf{z}_t} \log \mathbb{E} _{\mathbf{x}_0 \sim p(\mathbf{x}_0 | \mathbf{z}_t)}[p(\mathbf{y} | \mathbf{x}_0)]$. We believe that, roughly, all methods assume the reverse process $p(\mathbf{x}_0|\mathbf{x}_t)$ is Gaussian, characterized by a posterior mean and posterior covariance. The key difference lies in modeling posterior covariance: Most of methods, whether in pixel or latent spaces, such as DPS, DAPS, Resample, and so on, consider it zero. $\Pi$GDM assumes **Gaussian data distribution** and derives scaling from a **forward process**, which is prior-agnostic (proved in page 22 and 25), TMD applies a second-order Tweedie’s formula with a row-sum approximation. In contrast, we achieved as follows (revised version).
>
> **Latent Posterior Covariance.**
>
> To address this, we further analyze the structure of the posterior covariance and the Jacobian $\nabla _{\mathbf{z}_t}\mathbf{v} _{\boldsymbol{\theta}}(\mathbf{z}_t, t)$, under certain regularity assumptions on the latent prior $p(\mathbf{z}_0)$.
>
>  **Assumption (Strong Log-Concavity of the Latent Prior \cite{cattiaux2014semi})**
> Let $p(\mathbf{z}_0) = \exp(-\Phi(\mathbf{z}_0))$ be the latent prior over $\mathbb{R}^d$, where $\Phi$ is a twice continuously differentiable potential function. We assume that $p(\mathbf{z}_0)$ is $\gamma $-strongly log-concave for some $\gamma > 0$; that is, $\nabla^2 \Phi(\mathbf{z}_0) \succeq \gamma \, \boldsymbol{I}_d, \text{for all } \mathbf{z}_0 \in \mathbb{R}^d.$
>
> **(1) Log-concave Prior:** This assumption imposes a uniform lower bound on the Hessian of the prior's potential function, which translates to a lower and upper bound on the Jacobian of the vector field, as formalized below.
>
>
> **Proposition (Bound on the Jacobian of the Vector Field).** Let $\mathbf{v}(\mathbf{z}_t, t)$ denote the velocity field of the interpolant $\mathbf{z}_t = \alpha(t)\mathbf{z}_0 + \sigma(t)\mathbf{z}_1$ between a standard Gaussian prior and a target distribution $p(\mathbf{z}_0)$,  defined via coefficients $\alpha(t), \sigma(t) \in \mathbb{R}$. Under Assumption 1, the Jacobian of the vector field satisfies the following bound:
>
>
> $\frac{\mathrm{d}}{\mathrm{d}t}( \tfrac{1}{2} \log(\alpha(t)^2 + \gamma \sigma(t)^2)) \cdot \boldsymbol{I}_{d_z} \preceq \nabla _{\mathbf{z}_t} \mathbf{v} _{\boldsymbol{\theta}}(\mathbf{z}_t, t) \prec \frac{\mathrm{d}}{\mathrm{d}t} \log \sigma(t) \cdot \boldsymbol{I} _{d_z} \quad \text{for all } t \in (0,1],\; \mathbf{z} \in \mathbb{R}^d$
>
> The proof is provided in Appendix A. The resulting sandwich bound ensures the Jacobian remains well-behaved—the upper bound guarantees valid and stable posterior covariance, while the lower bound prevents overestimated uncertainty during posterior-guided inference.
>
> **(2) Gaussian Prior:** We now consider a tractable special case where the latent prior is Gaussian, i.e., $\mathbf{z}_0 \sim \mathcal{N}(0, \sigma _{\textit{latr}}^2 \boldsymbol{I})$,  a choice that is both theoretically justified and widely used in practice—for instance, in variational autoencoders (VAEs), where the latent prior is regularized toward a standard Gaussian via KL divergence~\cite{vahdat2021scorebased}. This enables us to derive the optimal velocity field and its Jacobian in closed form, which is characterized in the next proposition, whose proof can be found in Appendix A.
>
> **Proposition (Optimal Vector Field).** Let  $\mathbf{z}_0 \sim \mathcal{N}(\mathbf{0}, \sigma _{\text{latr}}^2 \boldsymbol{I} _{d_z})$  and $\mathbf{z}_1 \sim \mathcal{N}(\mathbf{0}, \boldsymbol{I} _{d_z})$ be independent random variables.  Define $\mathbf{z}_t = (1 - t) \mathbf{z}_0 + t \mathbf{z}_1$ for $t \in [0, 1]$. The optimal vector field $\mathbf{v}^{\star}(\mathbf{z}_t, t)$ that minimizes the expected squared error
>
>
> $\arg \min_{\mathbf{v}} \, \mathbb{E}\left[ \left\| \mathbf{v}(\mathbf{z}_t, t) - (\mathbf{z}_0 - \mathbf{z}_1) \right\|^2 \right]$ is given by  $\mathbf{v}^{\star}(\mathbf{z}_t, t) = \frac{(1 - t) \sigma _{\textit{latr}}^2 - t}{(1 - t)^2 \sigma _{\textit{latr}}^2 + t^2} \mathbf{z}_t.$
>
>
>  **Remark (Tightness of Bound).** When $\mathbf{z}_0 \sim \mathcal{N}(0, \sigma _{\textit{latr}}^2 \boldsymbol{I})$,  the optimal vector field $\mathbf{v}^\star(\mathbf{z}_t, t)$ achieves the Jacobian lower bound in Proposition 3. This follows from the potential function $\Phi(\mathbf{z}_0)=\frac{1}{2\sigma _{\textit{latr}}^2} \|\mathbf{z}_0\|^2$,  yielding $\gamma = \sigma _{\textit{latr}}^{-2}$, and interpolation coefficients $\alpha(t) = 1 - t$, $\sigma(t) = t$. Substituting into the bound confirms it matches the exact Jacobian of $\mathbf{v}^\star$, making the bound tight.
>
> Plugging the Jacobian of the optimal vector field into \eqref{} yields:
>
> $\mathbb{C}\mathrm{ov}[\mathbf{z}_0 \mid \mathbf{z}_t] = r^2(t) \cdot \boldsymbol{I} _{d_z}, \quad \text{with} \quad
> r^2(t)  = \frac{t^2 \left[(1 - t)(1 - 2t) + 2t^2 \right]}{(1 - t)\left[(1 - t)^2 + t^2\right]},$
>
> where we assumed $\sigma_{\text{latr}} = 1$. As a result, the propagated covariance can also be simplified as
>
> $\mathbb{C}\mathrm{ov}[\mathbf{x}_0 \mid \mathbf{z}_t] \approx J _{\mathcal{D}} \cdot \mathbb{C}\mathrm{ov}[\mathbf{z}_0 \mid \mathbf{z}_t] \cdot J _{\mathcal{D}}^\top \approx r^2(t) \cdot J _{\mathcal{D}} J _{\mathcal{D}}^\top.$
>
> Computing the full Jacobian $J_{\mathcal{D}} \in \mathbb{R}^{d_x \times d_z}$ is often infeasible in practice. To simplify, we assume that the decoder acts approximately as a local isometry near $\bar{\mathbf{z}}_0$ \cite{dai2020usual}, such that $J _{\mathcal{D}} J _{\mathcal{D}}^\top \approx \cdot P$, where $P$ is the orthogonal projector onto the image of $J _{\mathcal{D}}$.  For computational convenience, we approximate this behavior as isotropic in the full space, resulting in:
>
> $\mathbb{C}\mathrm{ov}[\mathbf{x}_0 \mid \mathbf{z}_t] \approx r^2(t) \cdot \boldsymbol{I} _{d_x}.$
>
> ---
> ### **4. Baselines**
>
> > We considered five baselines and provided both quantitative and qualitative results for four tasks of each. We notice that **SITCOM** was published at ICML 2025, with its latest arXiv version appearing after our NeurIPS submission. Nonetheless, we now cite it appropriately. For **DMPlug**, we have included additional results **Table 2**.
>
> | Methods      |        | **GDB** |        |        | **MDB** |        |        | **SR**  |        |        | **BIP** |        |
> |-------------|--------|---------|--------|--------|---------|--------|--------|---------|--------|--------|---------|--------|
> |             | PSNR↑  | SSIM↑   | LPIPS↓ | PSNR↑  | SSIM↑   | LPIPS↓ | PSNR↑  | SSIM↑   | LPIPS↓ | PSNR↑  | SSIM↑   | LPIPS↓ |
> | LatentDMplug       | 27.43  | 0.784   | 0.240  | 27.95  | 0.828   | 0.243  | **29.45**  | 0.838   | 0.183  | 22.50  | 0.819   | 0.220  |
> | LFlow | **29.10** | **0.837** | **0.166** | **30.04** | **0.849** | **0.168** | 29.12 | **0.841** | **0.176** | **23.85** | **0.867** | **0.132** |
>
> **Table 2:** *Quantitative comparison of LFlow and LatentDMplug across four inverse problems (BIP: Box InPainting, GDB: Gaussian DeBlurring, MDB: Motion DeBlurring, SR: Super-Resolution) on the FFHQ dataset*.
>
>
> > We respectfully note that several recent top-conference works have also focused exclusively on linear inverse problems, including but not limited to **OT-ODE** (TMLR 2024), **PSLD** (NeurIPS 2024), **PnP-Flow** (ICLR 2025), **FPS-SMC** (ICLR 2025), and **C-$\Pi$GFM** (NeurIPS 2024).
> ---
> ### **5. Questions**
>
> > 1. Details of the posterior covariance ablation are on page 25, but now explained in the right place.
> > 2. We added DAPS result in Table 4.

---

> > ### Author Response · Authors · 2025-08-05
> > **Kindly Reviewing and Sharing Your Feedback on Our Response**
> >
> > Dear Reviewer,
> >
> > We sincerely appreciate the time and effort you have already devoted to reviewing our work. We have carefully prepared a detailed response to address the concerns you raised. When convenient, we would be grateful if you could kindly review it and let us know whether it resolves the points in question. We would, of course, be happy to address any further questions or suggestions you may have.
> >
> > Thank you once again for your valuable time and consideration.
> >
> > Best regards,
> >
> > The Authors

---

> > ### Comment · Reviewer_xFYk · 2025-08-06
> >
> > I would like to thank the authors for their clarifications. I appreciate the detailed explanation regarding the differences between LFlow and existing methods that incorporate covariance estimation, as well as the additional comparison with DMPlug. However, I still have concerns about the discussion on the relationship between flow models and diffusion models.
> >
> > The authors argue that converting discrete-time diffusion models into flow models is impractical. I respectfully disagree. Even for quadratic schedulers, this conversion is feasible, as that the cubic equation introduced by EDM can be solved efficiently. This implies that LFlow could be directly applied to diffusion model checkpoints as well. Conversely, flow model checkpoints can also be converted into diffusion models [1, 2], allowing existing diffusion posterior sampling algorithms to be applied with minimal effort.
> >
> > While I agree with the authors that flow-based models demonstrate better empirical performance, I believe that designing a stronger generative prior is not the central focus or main contribution of this paper. The current form of the manuscript mixes the advantage of flow models with its core contribution of posterior covariance estimation, which I believe somewhat obfuscates the key message of this paper.
> >
> > Given how easy checkpoints can be transferred between diffusion and flow models, I kindly suggest that the authors instantiate LFlow using the same checkpoints employed by previous methods or, alternatively, evaluate previous methods using the checkpoints trained via flow matching. While I understand that completing such a study within a short timeframe is impractical, I believe a controlled study is essential and would substantially enhance the overall quality of the paper.
> >
> > Reference:
> > [1] Albergo et al. Stochastic Interpolants: A Unifying Framework for Flows and Diffusions, 2023.
> > [2] Ma et al. SiT: Exploring Flow and Diffusion-based Generative Models with Scalable Interpolant Transformers, 2024.

---

> ### Author Response · Authors · 2025-08-06
> **Flow vs Diffusion: Conversion**
>
> We appreciate the reviewer’s engagement and address the issue as follows.
>
> **1.** "flow model... be converted into diffusion models [1, 2], ... with minimal effort"...
>
> > We believe both references [1, 2] cited by the reviewer—one of which we already cite as [74]—only establish the theoretical connection between the vector field and the score function **for a single model with one specific noise schedule**.
>
> > For two models (flow and diffusion) with **different noise schedules**, we clarify that a practical conversion procedure was first introduced in **[3]** and later refined in **[4]**, which provided an extended proof and a **minor correction**.
>
> > In the first version of **[4]**, the authors stated: *“Proposition 1 allows us to initialize the Reflow with pre-trained diffusion models such as EDM [21] or Stable Diffusion [38].”* We had the opportunity to discuss with the authors of **[4]** some of the practical challenges, particularly in applying these methods to models such as SDs and LDMs, which operate in latent space. Notably, in the final version of **[4]**, the reference to Stable Diffusion was **removed**. We interpret this as an indication that the authors came to realise that implementing such theoretical ideas in practice—especially for latent-space models—is a far cry from what the theory might suggest.
>
> ---
> **2.** "Even for quadratic schedulers..., as ... cubic equation introduced by EDM can ...."
>
> > In the **EDM** reference **[5]**, no **cubic equation** is introduced. We would like to explain further what we meant by "cubic equation". Based on **[4]**, now the question is how we can find $s_{\textbf{vp}}$ and $\tau_{\textbf{vp}}$ for Latent Diffusion Models (LDM) or stable diffusions!. One practical solution is as follows: LDM uses a modification of the DDPM schedule. Both are variance preserving schedules, i.e., $\sigma'(t) = \sqrt{1 - \alpha'(t)^2}$, and define $\alpha'(t)$ for discrete timesteps in terms of diffusion coefficients $\mathbf{\beta_t}$ as $\alpha'(t) = \left(\prod_{s=0}^{\mathbf{t}} (1 - \mathbf{\beta_s})\right)^{\frac{1}{2}}$. For given boundary values $\mathbf{\beta_0}$ and $\mathbf{\beta_{T-1}}$, LDM uses $\mathbf{\beta_t} = \left(\sqrt{\mathbf{\beta_0}} + \frac{\mathbf{t}}{T-1}(\sqrt{\mathbf{\beta_{T-1}}} - \sqrt{\mathbf{\beta_0}})\right)^2$. **If we use**  Eq. 29 in Song's SDE paper **[6]** to compute the perturbation kernel of SD or LDM's noise schedule, we can get the following cubic equation, from which we may obtain $(\tau_{\mathrm{vp}}, s_{\mathrm{vp}})$ given $\beta_{\mathrm{start}}$ and $\beta_{\mathrm{end}}$.
>
> > $\frac{(\sqrt{\beta_{\text{end}}} - \sqrt{\beta_{\text{start}}})^2}{3} \tau^3 + (\sqrt{\beta_{\text{start}}} \(\sqrt{\beta_{\text{end}}} -\sqrt{\beta_{\text{start}}})) \tau^2 + \beta_{\text{start}} \tau + \ln (\frac{(1-t)^2}{(1-t)^2+ t^2})= 0$
>
> > While, in theory, it is possible to recover $\tau$ from a given value of $t$, in practice the presence of the logarithmic term introduces significant numerical challenges. In particular, to cover any meaningful range of $\tau$ (e.g., $0$ to $999$), we must evaluate the roots of the above cubic equation for values of $t$ increasingly close to $1$. The table below shows that even for $t$ values extremely close to $1$, the roots of the equation fail to span the desired useful range of $\tau$. This indicates that the practical difficulty is not at all reflected in the theoretical formulation. This is yet another example where the complexities of practical implementation can hinder—and at times even render impossible— the realisation of what theoretical results predict.
>
> | $t$  | $\tau$ |
> |-------|-------|
> | 0.20 | 2.68  |
> | 0.50 | 6.49 |
> | 0.90 | 12.33 |
> | 0.999 | 18.21  |
> | 0.999999 | 23.04 |
> | 0.999999999 | 26.43 |
>
> ---
> **3.** "I believe that designing a stronger generative prior is not ...of this paper"...
>
> > We agree with the reviewer; however, we never made such a claim. On the contrary, our generative prior does not exceed that of LDMs or Stable Diffusion v1.5, as explicitly stated in both the main text and the appendix.
>
> ---
> **Most importantly**, *regardless of conversion*, we must consider that our LFlow method is theoretically formulated for latent flow models whose autoencoders are trained with a VAE objective. This **VAE‑based regularization** is essential for our latent posterior covariance modeling. In contrast, other latent-based inverse models such as Resample and DAPS employ available LDMs with vector‑quantized regularization (i.e., **LDM-VQ-4**), which may not be suitable for our method, as they cannot provide the same theoretical justification.
>
> ---
> **[3]. Training-free linear image inverses via flows (pokle2023training)**
>
> **[4]. Improving the training of rectified flows (lee2024improving)**
>
> **[5]. Elucidating the design space of diffusion-based generative models (karras2022elucidating)**
>
> **[6]. Score-based generative modeling through stochastic differential equations (song2021scorebased)**

---

> ### Author Response · Authors · 2025-08-08
>
> Dear Reviewer,
>
> We have made continuous efforts, collectively as all authors, to address the issues raised by the reviewer. We sincerely hope that the clarifications and additional experiments we have provided will assist the reviewer in reassessing their evaluation.
>
> We would greatly appreciate it if the reviewer could let us know whether our clarifications have satisfactorily addressed their concerns. If not, we would be happy to know where and how our response may have fallen short, so that we can further clarify.
>
> Best regards,
>
> The Authors

---

> > ### Author Response · Authors · 2025-08-09
> > **Clarification on Conversion Between Flow and Diffusion Models**
> >
> > Dear Reviewer,
> >
> > We sincerely thank you for your thoughtful engagement. We hope our earlier response has addressed the concern regarding the conversion between latent-space **flow** and **diffusion** models with **two different noise schedules**. If any part remains unclear, please let us know, and we would more than happy to provide further clarification.
> >
> >
> > Best regards,
> >
> > The Authors

---

> > > ### Comment · Reviewer_xFYk · 2025-08-09
> > >
> > > I appreciate the authors’ clarifications regarding the conversion between flow models and diffusion models, as well as their discussion with other reviewers. The justification provided is convincing. While I think a controlled study testing LFlow on a transferred diffusion model would further strengthen the paper, I understand that this would be a nontrivial task, and I believe it does not detract from the paper’s main contribution to the inverse problem-solving community. I will therefore increase my score and now recommend acceptance.

---

> > > > ### Author Response · Authors · 2025-08-09
> > > >
> > > > Dear Reviewer,
> > > >
> > > > We greatly thank you for taking the time to re-engage in the discussion and for updating your recommendation. Your initial feedback and subsequent input have been invaluable in helping us improve our work, and your consideration is greatly appreciated.
> > > >
> > > > Best regards,
> > > >
> > > > The Authors

---

### Official Review · Reviewer_kCk5 · 2025-07-03

**Clarity:** 2
**Significance:** 3
**Originality:** 3
**Rating:** 5
**Confidence:** 3

**Summary:**

This work proposes a framework to solve linear inverse problems via latent flow matching models. Prior works based on diffusion models assume posterior covariance that is independent of the learned score or vector field which results in suboptimal solutions of the inverse problem. In contrast, this work derives an expression for posterior covariance  in latent space based on Tweedie’s formula.  In addition, the paper provides an expression for sampling in latent space from posterior by simulating a reverse time ODE. The formal proof shows that the probability density of the marginal distribution evolves according to the continuity equation. In practice, the expression for computing the conditional vector field is a sum of the unconditional vector field and a likelihood correction term. The latter can be evaluated with a Jacobian vector product. The method has been evaluated on different linear inverse problems like Gaussian deblurring, super-resolution, box inpainting, motion deblurring etc. under the set up of noisy linear inverse problem on $256 \times 256$ resolution images.

**Questions:**

__Questions__
1. Why is K-step update of vector field needed in Algorithm 1 (Line 7)? What quantity is re-used between the two iterations in this case (this is unclear from the current Algorithm 1)? As K is a hyper parameter, how does the performance of the algorithm vary with different values of K?
2. Line 118-119 correctly state that the derivation assumes that the dimensionality of $z_t$ matches that of $y$. However, it is unclear to me how the paper correctly relaxes this assumption.
3. How many NFEs on average are needed by adaptive heun sampler to converge? Are the NFEs in Table 3 for $\Pi$GDM and LFlow comparable?
4. How does LFlow perform in completely noiseless setting (i.e. $\sigma_y = 0.01$? How does its performance compare against $\Pi$GDM in this case? In my understanding, $\Pi$GDM performs very well under the noiseless setting.

__Suggestions__
1. Table 2 and Table 4 could be augmented with NFEs for each method too for a more comprehensive  overview of the methods.

**Ethical Concerns:**

["NO or VERY MINOR ethics concerns only"]

**Final Justification:**

The reviewers provided additional empirical results that addressed my concerns about baselines. My other questions and concerns have been satisfactorily addressed. Therefore I'm increasing my score.

**Limitations:**

yes

**Quality:**

3

**Strengths And Weaknesses:**

__Strengths__
1. Writing; The paper explains the main steps of the method well. Even though the method is quite involved, all the steps and assumptions involved in the algorithm have been listed well.
2. The setting considered in this work of latent space flow matching model is more realistic and opens avenues to solve inverse problems on high resolution images.
3. Qualitative and quantitative results seem good on different datasets and metrics as shown in Table 1 and Table 2.

__Weaknesses__
1. The method is slow and can take 4-10 mins to solve inverse problems even though it is faster than prior methods as shown in Table  4.
2. The PSNR metric seems consistently weak and it is unclear how it is reflected in qualitative results. Theoretically, it should result in slightly more burrier images.
3. (Minor) The paper only considers the noisy setting for linear inverse problem and not the noiseless setting which is also widely used.

---

> ### Author Rebuttal · Authors · 2025-07-27
>
> We are grateful for the reviewer’s insights and suggestions. We have carefully considered each point and provide our responses below.
>
>
> ### **1. Inference time**
>
> > We appreciate the reviewer’s observation. While our inference time is already faster than that of prior methods (as shown in Table 4), we acknowledge that the current implementation can still require approximately **3 min 40 s** to **10 min** to solve inverse problems. Specifically, for **deblurring** and **super-resolution**, the average time ranges from about **2 min 30 s** to **5 min 40s**, whereas for **inpainting** it is around **3 min 40 s** to **10 min**. This is why we generally report a range of **4 to 10 minutes**. We believe this runtime is primarily due to two factors:
>
>
>
>
> > - **Use of adaptive solvers**: Our inference relies on an adaptive ODE solver, which results in variable computation time across samples. While some samples complete in ~3 minutes, others—depending on task complexity—can take longer.
>
> > - **Repeated decoding for data consistency**: Similar to other latent-based inverse solvers, our approach enforces measurement consistency by repeatedly decoding intermediate latent states during inference. This decoding step, while necessary for accurate reconstruction, is computationally expensive unless the measurement operator is learned during training.
>
> > For context, D-Flow **[1]** reports sampling times ranging from **5 to 15** minutes on average.  We believe our method offers a reasonable trade-off, and in future work we aim to further reduce inference time through **solver optimization or by exploring alternative numerical methods like accelerated integration techniques** **[2]**.
>
> ---
> ### **2. PSNR values**
>
> > We appreciate the reviewer’s observation. While our PSNR is competitive but not always the highest among all baselines, our method consistently outperforms all baselines in SSIM and LPIPS across all tasks, as shown in Table 1. These two metrics are more aligned with perceptual and structural fidelity, which is reflected in the qualitative results in Figure 1—our reconstructions preserve fine textures (e.g., skin, hair) and edges more faithfully.  The slightly lower PSNR may be due to small misalignments or high-frequency texture recovery, which benefit perceptual quality but increase pixel-wise error. This trade-off has also been observed in prior perceptual-focused methods such as Resample. Please note that on **CelebA-HQ**, our method achieves the best PSNR in three tasks.
>
> ---
> ### **3. Noiseless setting**
>
> > We thank the reviewer for pointing this out. All our baselines implemented in the latent space are evaluated only under the noisy setting, so we followed this setting for a fair and consistent comparison. We now included results for the completely noiseless setting on the CelebA-HQ dataset for super resolution and inpainting task.
>
> ---
> ### **4. Questions**
>
> > **1.** For computational efficiency, we experimented with  $K=1$ and $K=2$. As shown in the ablation results provided below (**Table 1**), using $K=2$ yields slightly better average performance across tasks while maintaining reasonable inference time. We also tested $K=>3$, but observed little noticeable improvement—but increased computational cost. In what follows, we present our ablation study of hyperparameter $K$ on FFHQ dataset.
>
> ---
> | LFLow        |        | **GDB** |           |        | **MDB**     |           |        | **SR** |           |        | **BIP**     |           |
> |---------------|--------|-------------|-----------|--------|-------------|-----------|--------|------------------|-----------|--------|-------------|-----------|
> |               | PSNR↑  | SSIM↑       | LPIPS↓    | PSNR↑  | SSIM↑       | LPIPS↓    | PSNR↑  | SSIM↑            | LPIPS↓    | PSNR↑  | SSIM↑       | LPIPS↓    |
> | *K* = 1       | 28.72  | 0.829       | 0.172     |29.71  | 0.842       | 0.173    | 28.80  | 0.834            | 0.183     | 23.59  | 0.859       | 0.136     |
> | **K** = 2  | **29.10**  | **0.837**       | **0.166**     | **30.04**  | **0.849**       | **0.168**     | **29.12**  | **0.841**            | **0.176**     | **23.85**  | **0.867**       | **0.132**     |
> > **Table 1.** *Ablation of parameter* $K$ *on FFHQ. Increasing* $K$ *offers marginal gains with higher cost;* $K{=}2$ *achieves the best trade-off between performance and efficiency.*
>
> > **2.**  While Eq. (8) assumes matched dimensions for notational simplicity, our method is fully general and does not require $\mathbf{z}_t \in \mathbb{R}^k$ and $\mathbf{y} \in \mathbb{R}^m$ to have the same dimension. The likelihood is defined in image space as $p(\mathbf{y} \mid \mathbf{x}_0)$, with $\mathbf{x}_0 = \mathcal{D} _\varphi(\bar{\mathbf{z}}_0)$ and $\bar{\mathbf{z}}_0 = \mathbb{E}[\mathbf{z}_0 \mid \mathbf{z}_t]$. The guidance gradient is computed using the chain rule: $\nabla _{\mathbf{z}_t} \log p(\mathbf{y} \mid \mathbf{z}_t) =(\nabla _{\mathbf{z}_t} \bar{\mathbf{z}}_0)^\top J _\mathcal{D}^\top \mathcal{A}^\top \nabla _{\mathbf{x}_0} \log p(\mathbf{y} \mid \mathbf{x}_0)$. For a Gaussian likelihood, this yields the explicit formula given in Eq. (12) of the paper. We have clarified this mapping and removed any ambiguity in the revised manuscript.
>
> > Later, posterior mean of $\mathbf{z}_t$ given $\mathbf{z}_t$ is passed through the decoder, and consistency is enforced in pixel space as following
>
>
> >- $\nabla_{\mathbf{z}_t} \log p(\mathbf{y} \mid \mathbf{z}_t) \approx \underbrace{(\nabla _{\mathbf{z}_t} \bar{\mathbf{z}}_0)^\top J _{\mathcal{D}}^\top} \underbrace{.... (\mathbf{y} - \mathcal{A} \mathcal{D} _\varphi(\bar{\mathbf{z}}_0))} _{\mathbf{v}}$
>
>
> >- $ \underbrace{(\nabla _{\mathbf{z}_t} \bar{\mathbf{z}}_0)^\top J _{\mathcal{D}}^\top} = (J _{\mathcal{D}} \cdot \nabla _{\mathbf{z}_t} \bar{\mathbf{z}}_0)^\top = (\nabla _{\mathbf{z}_t} \mathcal{D}(\bar{\mathbf{z}}_0))^\top $
>
> > **3.** Please note that we use an adaptive solver, where the number of function evaluations (NFEs) naturally varies across samples. For this reason, instead of reporting a fixed NFE, we focus on inference time, which provides a more practical and consistent measure of efficiency across tasks and methods.
>
> > **4.** We thank the reviewer for the question. While our main focus is the noisy setting—in line with all baselines implemented in latent space—we included the results for the completely noiseless setting on the CelebA dataset for two tasks. We agree that $\Pi$GDM performs well in the noiseless setting; however, its performance drops significantly under noisy conditions.
>
> ---
>
> **[1]. D-flow: Differentiating through flows for controlled generation (ben2024d)**
>
> **[2]. FireFlow: Fast Inversion of Rectified Flow for Image Semantic Editing (deng2024fireflow)**
>
>
> **==============================****==============================****==============================**
> **==============================****==============================****==============================**
>
>
>
>
> **Note**. Due to space constraints in the individual response sections, we provide a brief summary of our main revisions and clarifications here. Reviewer-specific details are included where space allowed.
>
>
> **Summary of Revisions:**
>
>  We would like to provide all reviewers with a concise summary of the steps we have taken to address the comments and suggestions. For each major point, we have incorporated changes as recommended by the specific reviewer(s), as detailed below:
>
> - 1. **As recommended by Reviewer R8py**, we revised and modified the Related Work section and moved it before the Methodology section to clearly highlight prior approaches and better position our own contributions.
>
> - 2. **As suggested by Reviewer H6V9**, we improved the Methodology section—especially our contributions on latent likelihood approximation and posterior covariance estimation—and restructured it to better highlight our key contributions.
>
> - 3. **Following Reviewer R8py**’s suggestion, we added a conceptual figure illustrating the flow of our proposed method. We also moved the original figure (previously on page 25) to appear alongside this new figure, clearly presenting our methodology and highlighting the differences from prior work.
>
> - 4. **In response to Reviewer xFYk and Reviewer H6V9**, we added additional baselines. Results are now presented in two categories: **latent-based methods** (*MPGD*, *Resample*, *PSLD*, *LatentDAPS* on three datasets, and *LatentDMPlug* on FFHQ) and **pixel-based methods** ($\Pi$GDM, *OT-ODE*, and *C-$\Pi$GFM* on CelebA-HQ). Accordingly, we have revised and updated the Implementation Details section for improved readability and completeness.
>
> - 5. **Based on feedback from Reviewer kCk5**, we included an ablation study on the hyperparameter $K$, which controls the number of gradient update steps. This analysis clarifies the impact of $K$ on performance.
>
> We hope these targeted revisions directly address each reviewer's feedback and strengthen the overall clarity and quality of the paper.

---

> ### Comment · Reviewer_kCk5 · 2025-08-06
>
> I thank the authors for answering my questions about notation and for including additional ablations. One issue that I previously missed (and has been raised by another reviewer) is that in some of the results, the backbone model is not the same (as well as the checkpoint). I previously assumed that they were the same for fair comparison as has been done in previous works. While I understand that it is difficult to replicate all the results of all the previous methods, the paper should ideally include the performance of at least 1-2 other SOTA methods on the same model checkpoint. As a result, I would like to retain my score for now.

---

> > ### Author Response · Authors · 2025-08-07
> >
> > We thank the reviewer for their engagement.
> >
> > **1.** While I understand that it is difficult to replicate all the results of all the previous methods, the paper should ideally include the performance of at least 1-2 other SOTA methods on the same model checkpoint.
> >
> >
> > > We believe there may be a **misunderstanding**. As stated in the *Implementation Details* section of our paper and appendix, we have not modified any of the baseline methods. All baselines are adopted exactly as released by their authors, using the same official checkpoints and configurations from their respective GitHub repositories.
> >
> > > The SOTA latent-based methods (e.g., PSLD, LatentDMPlug, Resample, LatentDAPS) are based on **discrete-time latent diffusion models**. In contrast, our method leverages a **continuous-time flow in latent space**, which is novel in its own right, as **no** prior work has explored solving inverse problems using a **latent-based continuous-time flow prior**.
> >
> > > Our method adopts a different **training objective** (e.g., different loss, noise schedule, and hyperparameters), and we learn continuous-time **vector fields** rather than score functions or denoisers as in latent diffusion models. Specifically, our LFlow method requires a **VAE-based autoencoder**, which aligns with the theoretical formulation of our proposed latent flow framework. In contrast, some baselines—such as Resample and DAPS—are built upon vector-quantized LDMs (e.g., **LDM-VQ-4**), which are not compatible with our VAE-based latent space modeling and posterior covariance estimation.
> >
> > > The reviewer notes that “two diffusion models even with **identical architectures** and similar unconditional generation performance could still behave differently in inverse problems if they were trained on different datasets or with **different hyperparameters**.” While this might be true in other contexts, in our case this concern is **theoretically (probabilistically) ruled out**. Once the generative model is trained to learn continuous-time flows in latent space, the resulting distribution and sampling process are fully determined. At that point, the specific architecture used to implement the generative model serves mainly to support this objective, rather than being a source of variation or inconsistency at inference time.
> >
> > > **Nevertheless**, even if the reviewer’s point **holds entirely**, it still supports our argument, because our training objective naturally calls for a different design (as we mentioned in the third paragraph) to enable posterior covariance modeling. Thus, whether or not the architecture matches those of baselines becomes **a secondary concern**.
> >
> > > Lastly, we note that the reviewer expressed **partial agreement**, and we appreciate this. We elaborated further to provide additional clarification.

---

> ### Comment · Reviewer_kCk5 · 2025-08-07
>
> Thank you for your reply to my concerns raised above. I would like to further elucidate upon my concerns.
>
> "As stated in the Implementation Details section of our paper and appendix, we have not modified any of the baseline methods. All baselines are adopted exactly as released by their authors, using the same official checkpoints and configurations from their respective GitHub repositories."
>
> I understand that the authors have not modified any of the baseline methods and i agree that it is useful to include these metrics for comparison. However, 1-2 of these SOTA methods should be replicated with the same architecture and model checkpoint as the one used in LFlow.
>
> "The SOTA latent-based methods (e.g., PSLD, LatentDMPlug, Resample, LatentDAPS) are based on discrete-time latent diffusion models. In contrast, our method leverages a continuous-time flow in latent space, which is novel in its own right, as no prior work has explored solving inverse problems using a latent-based continuous-time flow prior."
>
> I also understand the point that this is a latent-based continuous time flow model. However, we can easily replicate inverse solvers for discrete time models with checkpoint for continuous time models. The vice versa is ofcourse not possible.

---

> > ### Author Response · Authors · 2025-08-08
> >
> > **1.** I also understand the point that this is a latent-based continuous time flow model. However, we can easily replicate inverse solvers for discrete time models with checkpoint for continuous time models. The vice versa is ofcourse not possible.
> >
> > > We believe that **this is not generally possible for all discrete-time methods**, as there are fundamental differences that require careful consideration. For example, **Resample** constructs a Gaussian posterior by combining a (supposedly) Gaussian prior on the unconditional reverse sample with a Gaussian pseudo-likelihood centered at a forward-projected estimate of the conditional posterior mean. Aside from the fact that these Gaussian assumptions are often unrealistic—potentially leading to miscalibrated posteriors and degraded inference—the method **explicitly relies on the forward process of the discrete-time diffusion**, which cannot be directly modeled in the same way in continuous-time flow frameworks. Similarly, LatentDAPS aims to improve posterior sampling via **global correction** through a two-step procedure: drawing $\mathbf{z}_0 \sim p(\mathbf{z}_0 | \mathbf{z}_t, \mathbf{y})$, followed by re-noising  $\mathbf{z} _{t - \Delta t} \sim \mathcal{N}(\mathbf{z}_0, \sigma _{t - \Delta t}^2 \boldsymbol{I})$.  This procedure inherently depends on the discrete-time noise schedule and the notion of discrete re-noising steps, which do not have a straightforward analog in the ODE-based continuous-time flow setting.
> >
> > > Second, many other baselines use SDE-based sampling—using DDPM, DDIM, and Hamiltonian Monte Carlo (HMC)—whereas our method is ODE-based and involves no noise injection during inference. This structural difference means that techniques designed for SDE trajectories cannot be directly ported to our setting without substantial reformulation, as the role of noise, variance scheduling, and stochastic integration is absent in ODE-based flows.
> >
> >
> > > However, we tried to extend PSLD (which is largely based on DPS) and MPGD to the continuous-time setting. The following table presents the results for these methods, denoted as CT-PSLD and CT-MPGD.
> >
> >
> > | Method        |        | **GDB**     |        |        | **MDB**     |        |        | **SR**      |        |        | **BIP**     |        |
> > |---------------|--------|-------------|--------|--------|-------------|--------|--------|-------------|--------|--------|-------------|--------|
> > |               | PSNR↑  | SSIM↑       | LPIPS↓ | PSNR↑  | SSIM↑       | LPIPS↓ | PSNR↑  | SSIM↑       | LPIPS↓ | PSNR↑  | SSIM↑       | LPIPS↓ |
> > | PSLD          | 30.28  | 0.836       | 0.281  | 29.21  | 0.812       | 0.303  | 29.07  | 0.834       | 0.270  | 24.21  | 0.847       | 0.169  |
> > | **CT-PSLD**       | 27.11  | 0.783       | 0.332  | 25.98  | 0.770       | 0.356  | 26.53  | 0.772       | 0.331  | 20.60  | 0.790       | 0.252  |
> > | MPGD          | 29.34  | 0.815       | 0.308  | 27.98  | 0.803       | 0.324  | 27.49  | 0.788       | 0.295  | 20.58  | 0.806       | 0.324  |
> > | **CT-MPGD**       | 24.96  | 0.765       | 0.343  | 24.30  | 0.751       | 0.375  | 24.69  | 0.740       | 0.338  | 20.01  | 0.780       | 0.353  |
> > | **LFlow**     | **29.10** | **0.837** | **0.166** | **30.04** | **0.849** | **0.168** | **29.12**  | **0.841** | **0.176** | **23.85** | **0.867** | **0.132** |
> >
> > **Table 1:** *Quantitative comparison of LFlow results on FFHQ dataset against continuous-time version of CT-MPGD and CT-PSLD across four inverse problems (BIP: Box InPainting, GDB: Gaussian DeBlurring, MDB: Motion DeBlurring, SR: Super-Resolution) on CelebA-HQ datasets.*
> >
> > ---
> >
> > We have made continuous efforts, collectively as all authors, to address the issues raised by the reviewer. We sincerely hope that the clarifications and additional experiments we have provided will assist the reviewer in reassessing their evaluation.

---

> > > ### Author Response · Authors · 2025-08-09
> > > **Follow-up on Clarifications and Additions**
> > >
> > > Dear Reviewer,
> > >
> > > We wish to express our gratitude for your constructive comments and engagement with our work. We have carefully followed your suggestions by adding the requested **ablation studies**, **conducting further experiments**, and providing **refined clarifications** to address your points. We value your perspective in this process, and we hope these updates clarify the aspects you highlighted. It would be great to know if you feel they satisfactorily address them.
> > >
> > > Best regards,
> > >
> > > The Authors

---

### Official Review · Reviewer_R8py · 2025-07-09

**Clarity:** 2
**Significance:** 3
**Originality:** 3
**Rating:** 4
**Confidence:** 4

**Summary:**

In this work, the authors propose LFlow that exploits a pretrained latent flow matching prior for solving linear inverse problems. It mainly exploits flow matching and ODE sampling in a latent space. The work develops a time-varying posterior covariance for latent representation for gradient-based estimation. Results are shown for a few image restoration tasks with comparison to diffusion models.

**Questions:**

1.	The term “training-free inference” is a confusing one for diffusion models and the like. Given many methods do use pre-trained models, so there is typically expensive training on datasets. Some recent diffusion methods even do test-time adaptation of the pre-trained model when training and test data are significantly mismatched. There are other classes of image restoration methods like deep image prior, which do not involve any pre-training but still involve test-time adaptation. They could be better thought of as training-free but test-time adaptive.

2.	Can the authors include a derivation for (3)?

3.	Below (14), Cov[z0|zt]  = (t^2/(1-t))E[z0|zt] is missing a gradient w.r.t. zt per (13)?

4.	Equation (20) uses E_phi(y). Is this an error since y (measurement) and x (image) have different dimensions? The following sentence says “ensures z_ts is closer to the posterior mode z0|y”, which is unclear. Is the y assumed to have noise?

5.	The Algorithm pseudocode could be better stated in the paper rather than the appendix.

6.	The Related Work section is misplaced before the final discussion in my opinion.

7.	There need to be more insights provided on how the flow is happening in the proposed model. Could authors add more results and insights?

**Ethical Concerns:**

["NO or VERY MINOR ethics concerns only"]

**Final Justification:**

The authors responded to my comments in detail. While I do have some remaining concerns such as comparisons/baselines and what authors claimed they included in revised manuscript, I find value in the work. Hence, I increased my score to borderline accept.

**Limitations:**

Yes

**Quality:**

3

**Strengths And Weaknesses:**

Strengths

1.	The work proposes latent space based flow matching for image restoration.

2.	The algorithms are theoretically motivated, with proofs included in the Appendix, which is a strength.

Weaknesses

1.	Despite rigor, there are numerous over-simplifying assumptions and approximations made throughout the paper that appear to make the work less complete and reduce interest, such as ones noted below.

   a.	For example, Proposition 3.1 assumes the reverse-time Fokker-Planck model.

   b.	Below (8), the dimensionality of zt and y are assumed matched, which is ad hoc. The authors say this can later be relaxed via an appropriate mapping. Please discuss more what you mean?

   c.	The linear approximation to the decoder in (10) seemed overly simplistic.

   d.	The Gaussian assumption on z0|zt wasn’t motivated.

   e.	In (11), it seems incorrect that the distribution of images x0 depends on zt for variable t. Approximations are also made in (11) and (12).

   f.	Assumption 3.2 could hold for cases like Gaussian priors with well-trained VAEs. But more needs to be said for other architectures or more complex priors.

   g.	Below (18),  the decoder Jacobian is assumed to be orthonormal, which is a very strong assumption.

2.	The comparisons to diffusion models in the experiments use only a few sample methods. There have been other diffusion works significantly outperforming DAPS etc., across tasks recently.

3.	Comparison to state of the art supervised image restoration methods is lacking.

4.	From Table 2, reconstruction accuracy metrics like PSNR are typically worse for LFlow although LPIPS seems slightly better. This makes me concerned the method may recover spurious realistic features. More discussion is needed.

5.	For results in Figure 2, the quantitative metrics could be indicated. For example, the dog on the bottom right for LFlow shows spurious features that DAPS doesn’t.

---

> ### Author Rebuttal · Authors · 2025-07-27
>
> We thank the reviewer for their time and constructive comments.
> ### **1. On the Necessity and Scope of Our Assumptions**
>
> > **a. Proposition 3.1.** This proposition is based on the **reverse-time continuity equation** (**Liouville** equation) for deterministic ODE flows, rather than the reverse-time **Fokker-Planck** equation, which describes stochastic diffusion processes (SDEs). Thus, our theoretical results apply directly to deterministic ODEs, not SDEs. This follows a standard setup used in prior works (e.g., Lipman et al., 2023) to justify sampling in continuous-time generative models. We revised the text to reflect this correction and avoid confusion.
>
> > **b. Dimension relaxation.** We thank the reviewer for this question. While Eq. (8) assumes matched dimensions for notational simplicity, our method is fully general and does not require $\mathbf{z}_t \in \mathbb{R}^k$ and $\mathbf{y} \in \mathbb{R}^m$ to have the same dimension. The likelihood is defined in image space as $p(\mathbf{y} \mid \mathbf{x}_0)$, with $\mathbf{x}_0 = \mathcal{D} _\varphi(\bar{\mathbf{z}}_0)$ and $\bar{\mathbf{z}}_0 = \mathbb{E}[\mathbf{z}_0 \mid \mathbf{z}_t]$. The guidance gradient is computed using the chain rule: $\nabla _{\mathbf{z}_t} \log p(\mathbf{y} \mid \mathbf{z}_t) =(\nabla _{\mathbf{z}_t} \bar{\mathbf{z}}_0)^\top J _\mathcal{D}^\top \mathcal{A}^\top \nabla _{\mathbf{x}_0} \log p(\mathbf{y} \mid \mathbf{x}_0)$. For a Gaussian likelihood, this yields the explicit formula given in Eq. (12) of the paper. We have clarified this mapping and removed any ambiguity in the revised manuscript.
>
> > **c. Decoder linearity.** We thank the reviewer for this point. Most latent-based inverse solvers ignore decoder nonlinearity during inference and effectively assume local linearity, but often *without formal justification*—see, e.g., MPGD, Resample, LatentDMplug, and LatentDAPS. While PSLD [31] attempts to explicitly address the decoder’s nonlinearity, recent work **[1]** has shown this approach to be ineffective and prone to artifacts. In our method, we use a first-order Taylor expansion around $\bar{\mathbf{z}}_0$, which provides a tractable and theoretically grounded way to propagate uncertainty. This linearization is empirically justified by the observed smoothness of pretrained decoders and the fact that posterior samples typically remain in high-density regions of the latent prior, where the decoder behaves approximately linearly.
>
> > **d. Motivation for the Gaussian assumption on $p(\mathbf{z}_0 | \mathbf{z}_t)$.** We thank the reviewer for this question. The Gaussian assumption for $p(\mathbf{x}_0 | \mathbf{x}_t)$ is widely used in training-free (zero-shot) guidance methods for pixel-space inverse solvers (see, e.g., **[2]**). This approximation is motivated by tractability: computing $\nabla{\mathbf{x}_t} \log \mathbb{E} _{\mathbf{x}_0 \sim p(\mathbf{x}_0 \mid \mathbf{x}_t)} [p(\mathbf{y} | \mathbf{x}_0)]$ is otherwise intractable, as it requires sampling from the full conditional at each step. In our case, the Gaussian assumption is further justified by the Gaussian latent prior and the smoother, more linear dynamics in latent space, which make the conditional distribution $p(\mathbf{z}_0 | \mathbf{z}_t)$ closely approximate Gaussianity—especially near high-density regions where the posterior concentrates. This is also supported by standard results for linear Gaussian systems in diffusion and flow models. We have clarified this motivation in the revised manuscript.
>
> > **f. Latent Prior $p(\mathbf{z}_0)$.**  Our assumption is grounded in a bound on Jacobian of vector field $\nabla_{\mathbf{z}_t} \mathbf{v} _\theta(\mathbf{z}_t, t)$ (Revised Proposition 3.3, which holds for any strongly log-concave prior, including  Smoothed Laplace,  Logistic Distribution, Generalized Gaussian with hight $\beta$,  and Sub-Gaussian priors with strong tail curvature, ensuring theoretical robustness even when the prior deviates mildly from exact Gaussianity. In Remark 3.4 (Please see our response to Reviewer xFYk), we show that the Gaussian prior is a tight instance of this bound, justifying the closed-form posterior covariance in Eq. 17.
>
> > **g. Local isometry.** While assuming $J_{\mathcal{D}}^\top J_{\mathcal{D}} \approx \mathbf{I}$ is indeed a simplification, we emphasize that: first, many decoder networks used in generative models (e.g., VAEs, diffusion decoders) are designed to approximately preserve local structure in the latent space, especially when trained with reconstruction loss and smooth latent priors. This implies that the decoder acts locally as a near-isometry around well-behaved regions of the latent distribution **[3]**, **[4]**; second, our assumption is only invoked locally around the posterior mean $\bar{\mathbf{z}}_0$, not globally across the entire latent space. In this regime, the decoder Jacobian typically varies smoothly, and near-orthonormal behavior has been observed empirically in prior work (e.g., VAEs, latent diffusion); third, importantly, the assumption allows us to arrive at a tractable and interpretable approximation of the likelihood gradient, which facilitates training-free inference.
>
> ---
> ### **2. Baselines**
>
> > We initially compared against five strong baselines. In response to all reviewer feedback, we have added additional baselines. Results are now presented in two categories: **latent-based methods** (*MPGD*, *Resample*, *PSLD*, *LatentDAPS* on three datasets, and *LatentDMPlug* on FFHQ) and **pixel-based methods** ($\Pi$GDM, *OT-ODE*, and *C-$\Pi$GFM* on CelebA-HQ). We kindly refer the reveiwer to **Table 2** (Reviewer **xFYk**) and **Table 3** (Reviewer **H6V9**).
>
> ---
> ### **3. Supervised baseline**
>
> >  Our focus was on training-free (zero-shot) methods, and none of the baselines we compare against—including recent state-of-the-art approaches—assume access to paired training data or require supervised fine-tuning. This aligns with the standard protocol adopted in many zero-shot studies. Supervised methods typically require task-specific training and do not generalize across different degradation operator settings.
>
> ---
> ### **4. Table 2: PSNR-LPIPS tradeoff and spurious features**
>
> > We thank the reviewer for this observation. As shown in Table 2, LFlow achieves consistently superior perceptual similarity (lower LPIPS) and SSIM across all tasks, while maintaining PSNR values comparable to the strongest baselines. This is in line with widely observed trends in generative models: methods optimized for perceptual quality may slightly sacrifice pixel-wise metrics like PSNR to better capture semantic structure and fine details. Occasionally, this can result in enhanced textures or subtle features that, while visually plausible, do not perfectly align with the ground truth—referred to as “spurious features.” Importantly, our qualitative results (Fig. 2) show that LFlow faithfully reconstructs key structures without introducing unnatural or fabricated artifacts. We have added further discussion of this well-known tradeoff and its implications for reconstruction quality in the revised manuscript.
>
> ---
> ### **5. Figure 2: dog case**
>
> > We thank the reviewer for the observation. Upon closer inspection, we note that LFlow’s reconstruction of the dog preserves the pose and structure consistent with the ground truth, while enhancing texture and fur detail. We believe the perceived "spurious features" may stem from higher-frequency detail restoration, not from hallucinated or incorrect content. In the revised manuscript, we have now annotated the visual examples in Figure 2 with their corresponding PSNR and LPIPS values for each method, including the dog example in the last row. This allows a direct quantitative comparison between perceptual and distortion-based metrics for the highlighted regions. We hope this clarifies how each method balances detail restoration and faithfulness to the reference image.
>
> ---
> ### **6. Questions**
>
> > **1.** We would like to clarify that the phrase “training-free inference” appears only once in our paper (Line 28), and we are happy to rephrase it if the wording is unclear. First, we acknowledge that diffusion models have proved to be effective for both **training-free inference** and **test-time adaptation**, the latter of which can involve updating the model during inference. Second, we recognize that the term **test-time adaptation** is broad and applies to various tasks—including classification, object detection, and inverse problems—whenever one has access to observations from a target domain and a pre-trained model from a source domain. In our case, however, we do not adapt the model at inference time; thus, no additional training is involved.
>
> > **2.** We have now included a derivation for Equation 3.
>
> > **3.** It is fixed now. Thank you.
>
> > **4.** In practice, $\mathbf{y}$ is a *degraded version* of $\mathbf{x}_0$, and we resize or zero-fill it to match the input size of the encoder $\mathcal{E} _\phi$, ensuring dimensional compatibility.  Also, as clarified in the experimental section, $\mathbf{y}$ indeed contains noise. Yes — we directly pass the measurement to the encoder rather than a reconstructed image ($\mathcal{A}^\dagger \mathbf{y}$).
>
> > The phrase *closer to the posterior mode* $\mathbf{z}_0 | \mathbf{y}$'' refers to initializing in a *plausible latent region* guided by $\mathbf{y}$, which improves the stability of reverse-time sampling.
>
> > **5.** We did this now.
>
> > **6.** We have now revised the Related Work section and moved it to Section 2.
>
> > **7.** We added a trajectory of image flow to demonstrate the generation process.
> ---
> **[1]. Prompt-tuning latent diffusion models for inverse problems (chung2023prompt)**
>
> **[2]. A Survey on Diffusion Models for Inverse Problem (daras2024survey)**
>
> **[3]. When is Unsupervised Disentanglement Possible? (horan2021unsupervised)**
>
> **[4]. Variational Autoencoders Pursue PCA Directions (by Accident) (2019Rolınek)**

---

> > ### Author Response · Authors · 2025-08-05
> > **Kindly Reviewing and Sharing Your Feedback on Our Response**
> >
> > Dear Reviewer,
> >
> > We sincerely appreciate the time and effort you have already devoted to reviewing our work. We have carefully prepared a detailed response to address the concerns you raised. When convenient, we would be grateful if you could kindly review it and let us know whether it resolves the points in question. We would, of course, be happy to address any further questions or suggestions you may have.
> >
> > Thank you once again for your valuable time and consideration.
> >
> > Best regards,
> >
> > The Authors

---

> > ### Comment · Reviewer_R8py · 2025-08-06
> >
> > Thanks to the authors for their thoughtful replies to comments. Their replies to my comments on the various assumptions and approximations is reasonable. Regarding comparisons to baselines, the authors mention LatentDAPS but I didn’t find results for it which they say are now presented. I think other methods showing improved performance over DAPS (pixel or latent space based) could be included. In response to comment on comparing to other supervised baselines, the authors say their focus is training-free methods – however, most diffusion models require pre-trained models learned on large datasets similar (whether paired or unpaired)  to supervised methods and can struggle when training-test distributions are significantly mismatched, requiring some form of OOD adaptation. Truly training-free methods would be ones that don’t need pre-trained models like deep image prior, etc. I still think expanding comparisons/baselines rather than making them too narrow in scope is important for a work to have broader impact.
> >
> > The authors keep referring to what they have done in the revised manuscript. However, I can’t access that information especially if new things are added there but not mentioned in previous submission or in rebuttal/response. The authors can expand these in the response to provide pertinent information there. Regarding the dog case, I still wonder about the artifacts – it would be good for the authors to look into more such cases or failure cases to build more understanding. I think this work is valuable but the points above could use more clarity for moving the work forward.

---

> ### Author Response · Authors · 2025-08-07
> **Clarifying Terminologies: Zero-Shot (Training-free) vs Supervised vs OOD adaptation**
>
> We sincerely appreciate your positive feedback and are glad that you found our clarifications on the assumptions reasonable. Below, we provide further clarification on the remaining points. Due to space constraints, we address your points in two parts.
>
> **1.** Clarifying the term **Training-Free**
>
> > We fully understand the reviewer’s concern, and we agree that the term **"training-free"** can sometimes be confusing. Here, we would like to clarify how this term is commonly used in the context of diffusion/flow-based inverse problem solvers.
> Most diffusion/flow based approaches to inverse problems fall into **two distinct paradigms** based on how they handle the measurement data during training:
>
> > + 1. Supervised (**Task-Specific**) Methods: These methods train a conditional diffusion model $p(\mathbf{x} | \mathbf{y})$ that is tailored to a specific degradation operator (e.g., super-resolution, deblurring). This setup requires paired datasets of measurements (degraded images) and clean images and is analogous to supervised learning. For example **[1]**.
>
> > + 2. Zero-Shot (**problem-agnostic**) Methods: In contrast, zero-shot methods (also referred to as **training-free** **[2, 3, 4]** or plug-and-play **[5]**) leverage a pretrained unconditional diffusion model $\nabla_{\mathbf{x}_t} \log p(\mathbf{x}_t)$ **trained only once** for generative modeling, without conditioning on specific measurements or inverse problems. **At inference time**, these methods combine the pretrained prior with any degradation operator (e.g., blur, mask, undersampling) to produce conditional samples. Crucially, no **retraining** or **finetuning** of the generative model is performed for each new inverse problem.
>
> > + Why Zero-shot **[6]** ? Zero-shot learning is one of the several learning paradigms aimed at out-of-distribution generalization, where the algorithm is trained to categorize objects or concepts that it has not been exposed to during training. For inverse problems, where the distribution of measurements $\mathbf{y}$ can change based on the undersampling pattern, the term *zero-shot solver* is sometimes applied. This usage is intended to reflect the solver’s ability to adapt to different undersampling patterns without retraining, similar to how zero-shot learning generalizes to new classes without prior exposure.
>
> > Therefore, when we refer to a method as **training-free**, we mean that the model is trained once on a dataset and can be directly applied for inference across various tasks and degradation operators **without any retraining**. Such zero-shot methods offer clear practical advantages, especially in scenarios where retraining for each task is infeasible. In this work, we specifically focus on zero-shot solvers for inverse problems.
>
> ---
> **2.** Training-free methods would be ones that don’t need pre-trained models like DIP, etc.
> > We totally agree with the reviewer about DIP and its being training-free. But we also note that the term *training-free* is commonly used in the context of diffusion/flow model-based solvers. This is **not** something we introduced, but follows from prior works, for example **[2 , 3, 4]**. **If the reviewer is still concerned, we are happy to adopt the term *zero-shot* or *plug-and-play* instead**.
>
> ---
> **3.** Test-Time Adaptation (TTA)
>
> > **TTA** refers to adapting (training) a model at inference time, without access to training data, using only the test sample(s) or their measurements, which is distributionally different from the training data. This allows the model to handle distribution shifts or unknown degradations not seen during training. In the context of **zero-shot inverse problem solvers**, TTA has been explicitly introduced in **[7, 8]**, where the titles themselves indicate that the methods are designed to address or challenge this task. For example, in **[8]**, the authors treat the already pretrained generative model as a DIP model and introduce additional LoRA parameters, which are then optimized during inference. OOD adaptation is beyond the scope of this work, and we believe it warrants dedicated investigation in a separate study, as we outlined in our discussion of future work.
>
> ---
>  **[1].** SUD $\^2$: Supervision by Denoising Diffusion Models for Image Reconstruction (chan2023sud)
>
> **[2].** **Training-free** linear image inverses via flows (pokle2024training)
>
> **[3].** **Understanding... **Training-free** Loss-based Diffusion Guidance (shen2024understanding)
>
> **[4].** TFG-Flow: **Training-free** Guidance in Multimodal Generative Flow (lin2025tfg)
>
> **[5].** Loss-guided diffusion models for **plug-and-play** controllable generation (song2023loss)
>
> **[6].** **Zero-shot** image restoration using ... null-space model (wang2023zero)
>
> **[7].** Steerable Conditional Diffusion for **Out-of-Distribution Adaptation** in ...Reconstruction (barbano2025steerable)
>
> **[8].** Deep diffusion image prior for efficient **ood adaptation** in 3d inverse problems (chung2024deep)

---

> > ### Comment · Reviewer_R8py · 2025-08-08
> >
> > Thanks to the authors for sharing their nuanced understanding of the training-free phrase. I think pre-trained zero-shot perhaps makes better sense as the phrase. This is similar to approaches like dictionary learning where the learning can be done on images in a task-agnostic matter and the learned model can be used in specific inverse problems. However, unlike methods like DIP, for diffusion, test-time OOD adaptation of pre-trained models might be necessary when there is distribution mismatch.

---

> ### Author Response · Authors · 2025-08-08
>
> Dear Reviewer,
>
> We have made continuous efforts, collectively as all authors, to address the issues raised by the reviewer. We sincerely hope that the clarifications and additional experiments we have provided will assist the reviewer in reassessing their evaluation.
>
> We would greatly appreciate it if the reviewer could let us know whether our clarifications and additional results have satisfactorily addressed their concerns. If not, we would be happy to know where and how our response may have fallen short, so that we can further clarify.
>
> Best regards,
>
> The Authors

---

### Note · Authors · 2025-08-13

We sincerely thank the Reviewers and Area Chair for their time, constructive feedback, and thoughtful engagement. We especially appreciate the patience and sustained effort these roles require.

We are encouraged by the recognition that the paper is well-written, theoretically rigorous, and delivers practical efficiency with promising results. Reviewer **R8py** noted it is “theoretically motivated, with proofs included in the Appendix.” Reviewer **kCk5** remarked that “the paper explains the main steps of the method and assumptions well.” Reviewer **xFYk** observed the methodology is “supported by rigorous theory,” and Reviewer **H6V9** described it as “well-written, clearly structured, and easy to follow,” with a reproducible appendix.

**During the rebuttal period**, we addressed all feedback with targeted revisions:

1. **Introduction and Related Work** (Following R8py)

   a. Replaced “training-free inference” with “zero-shot inference”.

   b. Moved Related Work section to precede the Methodology, *revising* it to clarify prior approaches and better position our contributions.

2. **Methodology**

   a. Following H6V9 and xFYk, we *revised* and *restructured* this section, especially on latent likelihood approximation and posterior covariance estimation, making propositions clearer with a remark linking them.

   b. Following R8py, we added a conceptual figure and relocated the comparison from page 25 to Methodology to highlight differences from prior work.

3. **Experiments**

   a. In response to xFYk, H6V9, and R8py, we added four new baselines including OT-ODE, C-$\Pi$GFM, LatentDMPlug, and SITCOM.
   b. Following R8py, we added an intermediate trajectory visualization for Gaussian deblurring, showing that early outputs are noisy or blurry, progressively refining toward the ground truth.

   c. Following kCk5, we extended PSLD and MPGD to the continuous-time setting with additional results. Note. Conversion from latent discrete- to continuous-time is non-trivial for pretrained models (see our response to xFYk) and for algorithms—some explicitly designed for discrete time (e.g., DAPS).

4. **Ablations**

   a. Based on kCk5’s feedback, we added an ablation on the effect of the number of gradient update steps.

   b. Per R8py’s suggestion, we compared LFlow in SR and box-inpainting with supervised methods.

These revisions strengthen the paper’s clarity, rigor, and impact, directly addressing all reviewer comments.

---

### Decision · Program_Chairs · 2025-09-17

**Decision:**

Accept (poster)

**Comment:**

Three reviewers recommend acceptance (two borderline), and one is a borderline reject. The paper is well-written and theoretically rigorous. The main concerns raised by the reviewers are the number of simplifying assumptions made and the limited comparison to state-of-the-art methods. The authors' detailed rebuttal, which included additional experiments, successfully addressed many of these concerns and led one reviewer to raise their score.